# A Novel Objective Function DYNO for Automatic Multi-variable Calibration of 3D Lake Models

Wei Xia[1,2,3], Taimoor Akhtar[4], Christine A. Shoemaker[1,2,3]

[1]Department of Civil and Environmental Engineering, National University of Singapore, 117576, Singapore
[2]Department of Industrial Systems Engineering and Management, National University of Singapore, 117576, Singapore
[3]Energy and Environmental Sustainability for Megacities (E2S2) Phase II, Campus for Research Excellence and Technological Enterprise (CREATE), 138602, Singapore
[4]RWDI Consulting Engineers and Scientists, N1G 4P6, ON, Canada

*Correspondence to:* Wei Xia (xiawei@u.nus.edu)

**Abstract.** This study introduced a novel Dynamically Normalized Objective Function (DYNO) for multi-variable (i.e., temperature and velocity) model calibration problems. DYNO combines the error metrics of multiple variables into a single objective function by dynamically normalizing each variable's error terms using information available during the search. DYNO is proposed to dynamically adjust the weight of the error of each variable hence balancing the calibration to each variable during optimization search. The DYNO is applied to calibrate a tropical hydrodynamic model where temperature and velocity observation data are used for model calibration simultaneously. We also investigated the efficiency of DYNO by comparing the calibration result obtained with DYNO to the result obtained through calibrating to only temperature and to the result obtained through calibrating to only velocity. The result indicates that DYNO can balance the calibration in terms of water temperature and velocity and that calibrating to only one variable (e.g., temperature or velocity) cannot guarantee the goodness-of-fit of another variable (e.g., velocity or temperature) in our case. Our study implies that in practical application, for an accurate spatially distributed hydrodynamic quantification, including direct velocity measurements are likely to be more effective than using only temperature measurements for calibrating a 3D hydrodynamic model. Our example problems were computed with a parallel optimization method PODS but DYNO can also be easily used in serial applications.

## 1. Introduction

Lake hydrodynamic models simulate the hydrodynamic or thermodynamic processes in lakes and reservoirs that are important for simulating water quality in aquatic eco-systems (Chanudet et al., 2012). These simulation models (e.g., hydrodynamic modelling) play a critical role in managing water bodies, as they are built to support the simulation of the spatial and temporal distributions of specific water quality variables (e.g., nutrients, chlorophyll-a), and to study the response of a water body to different future management scenarios. The parameters of these models usually need to be calibrated to measured data to adequately represent local effects and hydrodynamic processes. Model calibration is a vital step in complex hydrodynamic modelling of lakes and other aquatic systems.

Model calibration of lake hydrodynamic models is mainly done manually (also called trial and error), where experts tune the parameters and simultaneously evaluate the goodness-of-fit between the simulation output and observations. This process is subjective, time-intensive and requires extensive expert knowledge (Afshar et

al., 2011; Xia et al., 2021; Solomatine et al., 1999; Fabio et al., 2010; Baracchini et al., 2020). The challenges
associated with manual calibration have encouraged the application of auto-calibration to lake hydrodynamic
models, where the calibration is set up as an inverse problem to minimize the error between the simulation and
observations. Some studies (e.g., Gaudard et al. (2017), Luo et al. (2018), Ayala et al. (2020) and Wilson et al.
(2020)) have applied automatic calibration to one-dimensional hydrodynamic lake models where water
temperature is the variable that is simulated and calibrated. These one-dimensional models are relatively cheap to
run, allowing the use of automatic calibration methods that typically require many simulation evaluations to
determine suitable parameter sets (e.g., differential evolution used in Luo et al. (2018) and Monte Carlo sampling
used in Ayala et al. (2020)). However, one-dimensional models are unable to simulate the horizontal spatial
distribution and cannot capture the 3D processes, and thus may not be suitable for certain studies. Consequently,
2-dimensional or 3-dimensional models are preferred for studying the spatial-temporal distribution of water
variables and are increasingly used to study lakes around the world (Chanudet et al., 2012; Galelli et al., 2015;
Hui et al., 2018; Soulignac et al., 2017; Wahl and Peeters, 2014; Xu et al., 2017; Baracchini et al., 2020) . The
calibration of 3-dimensional models, though, is considerably more challenging than calibration of one-
dimensional models, since 3-dimensional models are significantly more computationally expensive and also
involve more complicated physical processes (such as advection of flows).
The computationally expensive character of 3-dimensional lake models makes traditional optimization
methods, such as differential evolution and Monte Carlo sampling, unsuitable for automatic calibration because
these methods usually require many evaluations to get an acceptable solution. Surrogate-based optimization is
highly suitable for such problems (Bartz-Beielstein and Zaefferer, 2017; Lu et al., 2018; Razavi et al., 2012) and
recent studies have applied surrogate-based optimization methods to parameter estimation of hydrodynamics
models. Surrogate-based optimization methods use a cheap-to-run surrogate approximation model (of the
calibration objective) fitted with all known (i.e., already evaluated) values of the original expensive objective
function, to guide the optimization search and reduce the number of evaluations required on the expensive
simulations. For example, Xia et al. (2021) proposed a new optimization method called PODS (parallel
optimization with dynamic coordinate search using surrogates) suitable for computationally expensive problems,
and applied it to automatic calibration of a three-dimensional lake hydrodynamic models. More elaborate
discussions on surrogate-based optimization algorithms can be found in Xia et al. (2021), Xia and Shoemaker
(2021), Razavi et al. (2012), Bartz-Beielstein and Zaefferer (2017) and Haftka et al. (2016).
Computational intensity is not the only critical challenge associated with parameter estimation of 3-
dimensional lake hydrodynamic models. Parameter estimation of these models is also a multi-site & multi-variable
calibration problem, i.e., observation data is usually available at multiple locations and the underlying models
simulate multiple variables (e.g., temperature and velocity). Moreover, simultaneous calibration of multiple
variables is desired due to complex interactions between the different variables. For instance, temperature and
velocity are inter-dependent variables of a lake hydrodynamic model, since water temperature affects the
movement of water, and water velocity affects the distribution of water temperature. However, most prior research
studies have calibrated hydrodynamic models to only temperature. This might be because temperature
measurements are relatively less expensive to get compared with velocity measurements and often temperature
measurements are available to help predict water quality phenomena. Wahl and Peeters (2014) use the measured
water temperatures to calibrate a 3-dimensional hydrodynamic model of Lake Constance. Kaçıkoç and Beyhan
(2014) calibrate the temperature of Lake Egirdir hydrodynamic model, the flow simulation of which is used for
the lake water quality modeling. Marti et al. (2011) and Xue et al. (2015) also only used temperature data for lake
hydrodynamic model calibration. Moreover, these studies use manual calibration for parameter estimation. Xia et
al. (2021) use automatic calibration for parameter estimation, but only use water temperature observations in the
calibration process. Reproducing water level is also a parameter estimation approach that pseudo-considers flow
dynamics in calibration; however, a calibrated model that correctly simulates observed water level does not
necessarily reproduce the observed 3D flow field accurately (Wagner and Mueller, 2002; Parsapour-Moghaddam
and Rennie, 2018). Amadori et al. (2021) investigated the use of different sources of temperature data (from in-
suite observations, multi-site high-resolution profiles and remote sensing data) to compensate for the scarcity of
velocity measurements. This is a practicable approach when there is no velocity data available and there are such
different sources of temperature data available. However, when there is no high-quality remote sensing data (for
example, because of cloud) or a large amount of high-resolution profiles of temperature measurement it is still
challenging to verify the spatial simulation of hydrodynamic quantities.

Lake hydrodynamic models predict the velocities throughout a water body. Accurate velocity simulations

are thus important to understand the spatial distribution of water quality problems (e.g., algal blooms) in sizeable
lakes. Hence, during the calibration of these models, it is useful to know whether efforts to measure velocity
directly are justifiable even if temperature data is already available. We will examine the extent to which direct
measurement of velocities justifies the extra effort by giving more accurate results for hydrodynamics models.
We will also look at the error of the spatial distribution of hydrodynamics associated with calibrating to
temperature only, which is rarely studied in the literature.

There are a few studies that attempt to calibrate lake hydrodynamic models to both temperature and

velocity. Chanudet et al. (2012) attempt to calibrate both temperature and velocity sequentially (using manual
calibration), i.e., they calibrate water temperature first and then the current velocities. Baracchini et al. (2020)
performed two sequential steps in the automatic calibration of temperature and velocity, and the velocity
calibration is based on the results obtained from temperature calibration. However, one problem with such two-
step sequential approaches, either by manual or auto-calibration, is that the calibration of the second variable
might significantly alter the calibration quality of the first variable. This is especially true for multi-variable
calibration problems, where the multiple variables being calibrated are sensitive to the parameters being
calibrated. Other examples of such multi-variable calibration problems include watershed model calibration
(Franco et al., 2020) and seawater intrusion model calibration (Coulon et al., 2021), among others. These multi-
variable problems require calibration frameworks that allow simultaneous calibration of all variables rather than
calibrating one and then the second.

There are prior studies that simultaneously calibrate both temperature and velocity variables of

hydrodynamic models. However, these use a trial and error (manual) mechanism for calibration (Råman Vinnå et
al., 2017; Soulignac et al., 2017; Jin et al., 2000; Paturi et al., 2014). Manual calibration of multiple hydrodynamic
variables simultaneously, is even harder than calibration of a single variable. A key challenge for automatic
calibration of multi-variable calibration problems is in defining a suitable objective function. Traditional
approaches typically formulate the goodness-of-fit of multiple variables into a single objective function by adding
weights between the goodness-of-fit of multiple variables and solve the problem with single objective
optimization (SOO) techniques (Afshar et al., 2011; Pelletier et al., 2006). However, a drawback of this approach
is that the relative error magnitude of each variable of the new solutions found will probably vary during the
search making it difficult to determine appropriate weights since they need to be determined / defined *a prior*, i.e.,
before optimization.

Another approach for calibration of multi-variables is using multi-objective optimization (MOO)

techniques (Afshar et al., 2013). However, multi-objective techniques are commonly used to optimize multiple
sub-objectives that have a trade-off between each sub-objective (Akhtar and Shoemaker, 2016; Reed et al., 2013;
Alfonso et al., 2010; Giuliani et al., 2016; Herman et al., 2014). While for the multi-variable hydrodynamic
calibration problems, it is not apparent that there is usually a trade-off between the fit of multiple variables.
Moreover, MOO is considerably more computationally difficult than SOO and typically requires many more
objective function evaluations. Thus, MOO may not be desired for computationally expensive calibration
problems, especially when a significant trade-off between the objectives may not be present. Consequently, multi-
variable calibration utilizing efficient SOO algorithms, while balancing the calibration to each variable equally
during calibration, is a research area of significant value.

We introduce a new Dynamically Normalized Objective Function (DYNO) for automatic multi-variable

calibration. The error of each variable (e.g., temperature and velocity of hydrodynamic models) is dynamically
normalized by using the information about variable error of the evaluations found during the optimization search
process. In this way, the balance between the calibration of each variable is dynamically adjusted. We tested the
efficiency of DYNO on a computationally expensive hydrodynamic lake model of a tropical reservoir, which
takes 5 hours to run per simulation. DYNO is coupled into a recent parallel surrogate optimization algorithm,
PODS (Xia et al., 2021), and successfully applied for the calibration of multiple variables of the hydrodynamic
model. Using DYNO, we investigate the impact of using temperature and/or velocity observations on model
accuracy. Since velocity measurements are usually not included in standard lake monitoring systems (whereas
temperature measurements are included), real velocity observations are seldom available (Amadori, et al, 2021).
Real observations for velocity are not available in our case as well. Hence, we conducted our investigation based
on synthetic observations generated from a calibrated model. It is worthwhile to revisit and validate this analysis
with real velocity measurements if they are available in the future.
**2. Methodology**
**2.1 Multi-variable Calibration Problems Description**
The calibration problems investigated in this study are multi-site (i.e., observations are available from multiple
locations), multi-variable (e.g., temperature and velocity for hydrodynamics) problems, and are defined
mathematically as follows (the variable and function definition are given in Table 1):
$$\min_{X \in \Theta} F(X|K) = F(\{f_k(X)|k \in K\}) \tag{1}$$
$$f_k(X) = f_k(\{g_j(Sim_j^k(X), Obs_j^k)|j = 1, \cdots, M\}) \tag{2}$$
Note that the notation $\{z_i\}$ in Eq. (2) is simply meant to imply the function on the left depends on the finite series
of quantities inside the braces $\{\bullet\}$.
**Table 1**. Notation and definitions of variables and functions in Eq. (1) and (2).

| Variable | Description |
|----------|-------------|

| | |
|---|---|
| $K$ | The set of variables whose observation data is used in calibration. For example, $K = [Tem]$ means that water temperature observation is used for calibration, i.e., water temperature is the variable that is being calibrated; $K = [\overrightarrow{Vel}]$ means velocity observation is used for calibration; $K = [Tem, \overrightarrow{Vel}]$ means that both temperature and velocity observations are used for model calibration |
| $k$ | The symbol for elements in $K$ variable (e.g., water temperature or velocity, $k = Tem$ or $k = \overrightarrow{Vel}$). $k \in K$ |
| $X$ | A $d$ dimensional parameter vector restricted to parameter space $\Theta$, where $d$ is the number of parameters to be optimized. $X = (x_1, x_2, \ldots, x_d)$. |
| $\Theta$ | The parameter space is defined by the upper and lower limits on each parameter ($X^{max}$ and $X^{min}$, respectively) |
| $M$ | The total number of observation locations (or sites). |
| $j$ | The index for observation location. $j = 1, \ldots, M$ |
| $Sim_{t,j}^k(X)$ | The simulation output of variable $k$ at location $j$ at time step $t$ given the parameter vector $X$ |
| $Obs_{t,j}^k$ | The observation (data) of variable $k$ at location $j$ at time step $t$. |
| $Sim_j^k(X)$ | The simulation time series output of variable $k$ at location $j$ at times $t = 1, \ldots, N$ given the parameter vector $X$. $Sim_j^k(X) = (Sim_{1,j}^k(X), \ldots, Sim_{N,j}^k(X))$. |
| $Obs_j^k$ | The observation (data) time series of variable $k$ at location $j$ at times $t = 1, \ldots, N$. $Obs_j^k = (Obs_{1,j}^k, \ldots, Obs_{N,j}^k)$. |
| $N$ | The total time steps of the observation data |
| $t$ | The index for time steps. $t = 1, \ldots, N$ |
| **Function** | **Description** |
| $F(X\|K)$ | The calibration objective function given the observation data of variables in $K$ for calibration. $F(X\|K)$ is a composite function of $f_k(X)$ |
| $f_k(X)$ | The error function of variable $k$ over multiple site. $f_k(X)$ is a composite function of $g_j(Sim_j^k(X), Obs_j^k)$ for sites $j = 1, \ldots, M$ |
| $g_j(Sim_j^k(X), Obs_j^k)$ | Goodness of fit between time series simulation output $Sim_j^k(X)$ and observation $Obs_j^k$ of variable $k$ at location $j$. When $k = Tem$, Normalized Root Mean Square Error (NRMSE) is utilized for $g_j(\bullet)$. When $k = \overrightarrow{Vel}$, normalized Fourier Norms of Root Mean Square Error (FNs) is used for $g_j(\bullet)$. |


The set of parameters $X$ being calibrated in this study includes nine parameters ($d = 9$). Details of these
parameters are provided in Table 2 in section 2.4. The two variables calibrated in this study are velocity and
temperature, for which data exists for different spatial locations and time points.
We investigate different calibration formulations, where either one or both of these variables are
calibrated. Consequently, $K = [Tem]$ means that water temperature observation is used for calibration, i.e., water
temperature is the variable that is being calibrated; $K = [\overrightarrow{Vel}]$ means velocity observation is used for calibration;
$K = [Tem, \overrightarrow{Vel}]$ means that both temperature and velocity observations are used for model calibration, i.e., both
variables are being calibrated simultaneously. The objective function in each scenario is discussed in section 2.5.
**2.2 DYNO for Model Calibration with Multiple Variables**
One major issue for model calibration with multiple variables is how to formulate the error of multiple variables
with a single objective function. In practice, different variables (e.g., temperature and velocity) usually have
different physical units and magnitudes of error. Their error functions cannot be summed up directly into a single
objective function if we wish to give the error of each variable an equal weight in the overall objective function.
The respective error functions have to be normalized. There are goodness-of-fit metrics that can normalize the
error of different variables (for example, Normalized Root Mean Square Error (NRMSE) and Kling-Gupta
Efficiency (KGE, (Gupta et al., 2009))). However, it is still possible that the highest attainable value (or
distribution) of NRMSE (or KGE) across the parameter space for one variable maybe be much higher than the
highest attainable value (or distribution) of NRMSE (or KGE) of another variable. Hence, how to balance such
differences among multiple variables is still important even when the normalized goodness-of-fit metrics are used.
We propose a new general objective function, DYNO, for the multi-variable calibration problem. Let $\psi$
be the set of evaluations found so far by the optimization, DYNO (as shown in Eq. (3)) normalizes the error of
each variable $f_k(X)$ with its upper and lower bound, $f_k^{max}$ and $f_k^{min}$ of all evaluations in $\psi$. Since true values of
bounds are not known, $f_k^{max}$ and $f_k^{min}$ are dynamically updated during the optimization search after each iteration.
The Mathematical formulation of the multi-variable calibration problem, with DYNO, is as follows:
$$min\ F\ (X|K) = \sum_{k \in K} \frac{f_k(X) - f_k^{min}(X)}{f_k^{max}(X) - f_k^{min}(X)} \tag{3}$$
$$f_k^{max}(X) = \ max\{f_k(X)\ for\ all\ X \in \psi\} \tag{4}$$
$$f_k^{min}(X) = \ min\{f_k(X)\ for\ all\ X \in \psi\} \tag{5}$$
where $f_k^{max}(X)$ and $f_k^{min}(X)$ are the maximum and minimum values of $f_k(X)$ for all evaluations in $\psi$ . $f_k^{max}(X)$
and $f_k^{min}(X)$ have to be updated dynamically in each iteration during optimization. The detailed description of
the implementation of Eq. (3) in the algorithm (i.e., PODS) tested in this study is given in Section 2.6 (the
Algorithm Description section).

## 2.3 Study Site and Data

We use a 3-dimensional model of a tropical reservoir as an example to test the efficiency of DYNO for multi-
variable calibration problems and to study the impact of using temperature and/or velocity data for model
calibration. The horizontal boundary of the studied reservoir is given in Fig. 1 (a) and (b). The reservoir has over
250 ha of water surface with a maximum depth of about 22 meters. One online water quality profiler station (STN.
A1) was installed in the middle of the reservoir. The water temperature data at the station are available at various
depths. The model is built for the simulation of the year 2013. One-year measured temperature data was used for
model calibration in a previous study (Xia et al., 2021). We use this calibrated model to create synthetic
observation data since the real velocity measurements are not available. We first assume a set of "true" model
parameters $X^R$. The value of $X^R$ is based on manual calibration by experts and is listed in Table 2. The spatial
and temporal observation data for the hypothetical lake is synthetically generated based on the "true" model
parameters $X^R$. The synthetic observation data for the hypothetical temperate lake is generated by running the
simulation model for one year with the vector of model parameters $X^R$. The simulation output is then saved hourly
in $N$ time steps for multiple variables, i.e., temperature and velocity ($K = [Tem, \overrightarrow{Vel}]$) at $M$ locations (specified
in Fig. 1). In our study case, $N = 8761$ and $M = 12$ with different depths of five hypothetical sensor stations
(STN. A1 and STN. B1-4 as shown in Fig. 1 (a) and (b)).
The saved hourly simulated output time series is denoted as $\Gamma = \{Sim_j^k(X^R), k \in K = [Tem, \overrightarrow{Vel}], j = $
$1, ..., M\}$, which as defined (in Table 1) contains information for each time step, $t = 1, ..., N$. So $\Gamma$ is used as

observation data for model calibration, i.e., $Obs^k, k \in \mathbf{K} = [Tem, \overrightarrow{Vel}]$ in Eq. (1). In the test of optimization for calibration, the true values of the parameter vector $\mathbf{X}^R$ are not provided to the optimization. The optimization will, instead, search for the best set of $\mathbf{X}$ that will minimize the objective function $F(\mathbf{X}|\mathbf{K})$, where $\mathbf{K} = [Tem], [\overrightarrow{Vel}], or [Tem, \overrightarrow{Vel}]$. So the goal of automatic calibration via optimization is to obtain an optimum calibration $\mathbf{X}^*$ that results in simulation model output, $\mathbf{Sim}_j^k(\mathbf{X}), k \in \mathbf{K}, j = 1, ..., M$, (see Eq. (1) and Eq. (2)) that is close to the synthetic observation time series data in $\mathbf{\Gamma}$.

The temperature and velocity simulation results at the year 2013 based on the "true" model parameters (shown in Table 2) show temporal and spatial variation, as shown in Fig. 1 (a)-(d). Figure 1 (a) and (b) show the temperature and horizontal velocity distribution at the surface layer. Figure 1 (c) and (d) show the distribution of temperature and velocity magnitude at STN. A1. There is obvious temperature stratification in the vertical direction (as shown in Fig. 1(c)). We have five sampling locations across the reservoir. The observation data at these five locations are used to calibrate the model parameters.

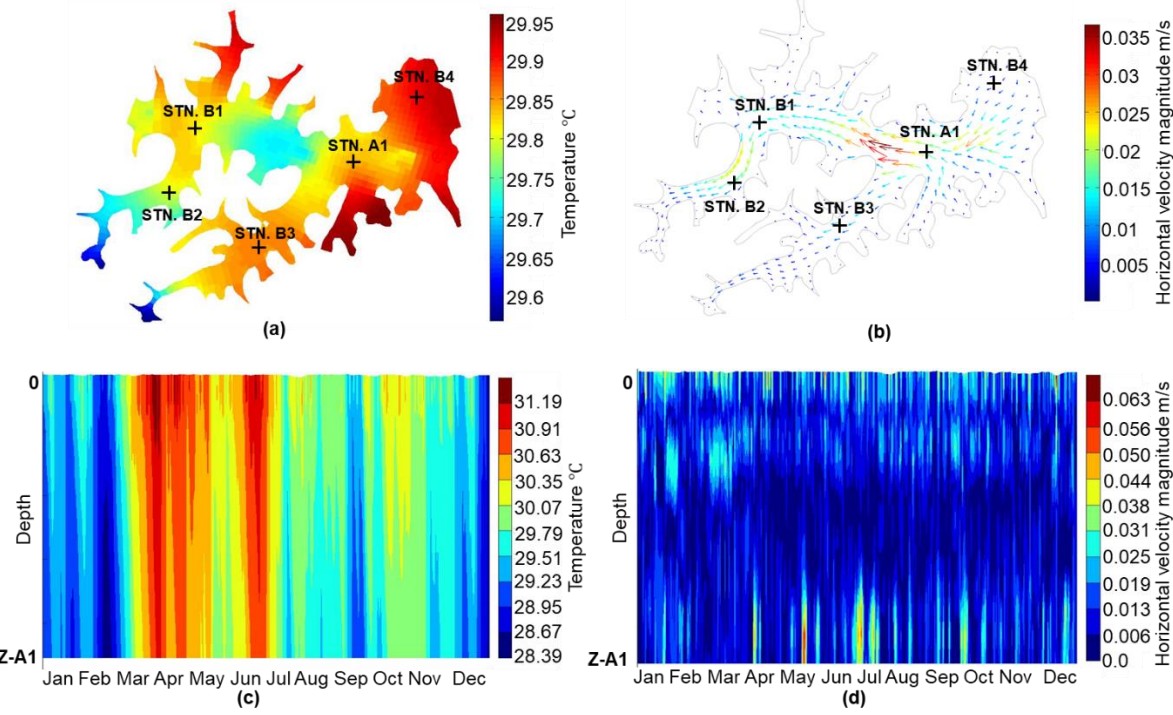

**Figure 1.** Hydrodyanmic model simulation result (with "true" model parameters) at the year 2013. (a) Simulated temperature spatial distribution with sampling locations. (b) Simulated velocity spatial distribution with sampling locations. (c) Time-depth plot of simulated temperature at STN. A1. (d) Time-depth plot of velocity magnitude at STN. A1. Z-A1 is the maximum water depth at station A1.

**2.4 Hydrodynamic Model and Calibration Parameters**

The description of the hydrodynamic model is given in (Xia et al., 2021). The hydrodynamic model is built with Delft3D-FLOW (Hydraulics, 2006). The Delft3D-Flow hydrodynamic model used was set up by the water utilities' employees and consultants, including the domain construction, input data preparation, and model configuration. The grid coordinate system is based on Cartesian coordinates (Z-grid), which havs horizontal coordinate lines that are almost parallel with density interfaces to reduce the artificial mixing of scalar properties such as temperature. The number of grid points in the x-direction is 65, the number of grid points in the y-direction

is 67, and the number of layers in vertical is 19. A single 1-year simulation takes about 5 hours to run in serial on
a windows desktop with CPU Intel Core i7-4790.
There are nine tunable model parameters (listed in Table 2) in the model. The first five parameters in
Table 2 are related to the turbulence calculation. The k-ε closure model (Uittenbogaard et al., 1992) was chosen
as the turbulence closure model to calculate the viscosity and diffusivity of the water. The calculation of the
viscosity and diffusivity involves five parameters: 1) background viscosity in horizontal $v_H^{back}$ , 2) vertical $v_V^{back}$,
3) the background eddy diffusivity in horizontal $D_H^{back}$ , 4) vertical $D_V^{back}$ and 5) the Ozmidov length $L_{oz}$. These
parameters affect both the velocity and the temperature. The vertical exchange of horizontal momentum and mass
is affected by vertical eddy viscosity and eddy diffusivity coefficients (Elhakeem et al., 2015). The horizontal
velocities are affected by the horizontal eddy viscosity and diffusivity coefficients (Chanudet et al., 2012).
Chanudet et al. (2012) highlighted that the most impactful parameter for temperature is the background vertical
eddy viscosity and the Ozmidov length $L_{oz}$ also has a significant effect on the thermal stratification by affecting
the vertical temperature mixing.
The next three parameters in Table 2 are related to the simulation of surface heat flux. In the heat flux
model, the evaporative heat flux and heat convection by forced convection are parameterized by the Dalton
number $c_e$ and Stanton number $c_H$ , respectively, which are also in the list of calibration parameters. The Secchi
depth $H_{Secchi}$ (also included in Table 2) is another parameter required by the Ocean heat flux model. Secchi depth
is related to the transmission of radiation in deeper water and thus affects the vertical distribution of heat in the
water column (Chanudet et al., 2012). Heat fluxes through the reservoir bottom were not simulated in the current
model. The last parameter is the Manning coefficient, which affects the roughness of the bottom of the lake and
has a direct impact on velocity.
All these nine parameters affect (either directly or indirectly) the thermal and current activity in the water
body. These are also the parameters included in the routine model calibration by local experts and thus, are
included in the calibration process in our study. The calibration range for these parameters (given in Table 2) is
suggested by Singapore water utilities employees and consultants. Some of these parameters might be
spatiotemporally variant (such as Secchi depth, Ozmidov length scale, Dalton number, and Stanton number).
Considering these parameters as time or space-varying parameters will substantially increase the number of
decision variables in optimization. Considering that the reservoir in our study is relatively small and is located in
a tropical region where there is no significant seasonal variation, we consider these parameters to be constant
across space and time.
**Table 2.** Model parameters used in calibration. $X^R$ denotes the true solution used to generate synthetical
temperature and velocity observations at multi-sites.

| Parameter vector $X$ | Parameter | Description (unit) | Physical process | Range | $X^R$ |
|---|---|---|---|---|---|
| $x_1$ | $v_H^{back}$ | Background viscosity in horizontal (m²/s) | | 0.1-1.0[a,b,d,e] | 0.5 |
| $x_2$ | $D_H^{back}$ | Background eddy diffusivity in horizontal (m²/s) | 3D turbulence | 0.1-1.0[a,b,d,e] | 0.5 |
| $x_3$ | $v_V^{back}$ | Background viscosity in vertical (m²/s) | | 0-0.005[a,b,c,e] | 5.00E-05 |

| | | | | | |
|---|---|---|---|---|---|
| $x_4$ | $D_V^{back}$ | Background eddy diffusivity in vertical (m²/s) | | 0-0.005 [a,b,c,e] | 5.00E-05 |
| $x_5$ | $L_{oz}$ | Ozmidov length scale (m) | | 0-0.05 [a,b,e] | 0.015 |
| $x_6$ | $H_{Secchi}$ | Secchi depth (m) | | 0.1-2.0 [a,e,f] | 1 |
| $x_7$ | $c_e$ | Dalton number (-) | Heat flux | 0.001-0.002 [a,b,c,e] | 0.0013 |
| $x_8$ | $c_H$ | Stanton number (-) | | 0.001-0.002 [a,b,c,e] | 0.0013 |
| $x_9$ | $n$ | Manning coefficient (m$^{-1/3}$s) | Roughness | 0.02-0.03 [a,b,e] | 0.022 |

[a]Deltares (2014); [b]Chanudet et al. (2012); [c]Wahl and Peeters (2014); [d]Råman Vinnå et al. (2017); [e]Soulignac et al. (2017); [d]Pijcke (2014)

## 2.5 Calibration Problem Formulation

Three scenarios are considered to investigate the impact of model calibration against temperature and/or velocity observations (as discussed in section 2.1). The first two scenarios calibrate to only one variable, and the last scenario calibrates both variables simultaneously. This section gives the detailed calibration formulations of these three scenarios.

## 2.5.1 Model Calibration with One Variable

The objective functions for Cali-Tem and Cali-Vel scenarios are summarized in Eq. (6)-(8) and Eq. (9)-(11), respectively, where only observations of one variable are included in the calibration.

$$F(\boldsymbol{X}|\boldsymbol{K} = [Tem]) = f_{Tem}(\boldsymbol{X}) \tag{6}$$

$$f_{Tem}(\boldsymbol{X}) = \sum_{j=1}^{M} NRMSE_j^{Tem}(\boldsymbol{X}) \tag{7}$$

$$NRMSE_j^{Tem}(\boldsymbol{X}) = \frac{\sqrt{\frac{1}{N}\sum_{t=1}^{N}\left[Sim_{t,j}^{Tem}(\boldsymbol{X}) - Obs_{t,j}^{Tem}\right]^2}}{\frac{1}{N}\sum_{t=1}^{N} Obs_{t,j}^{Tem}} \tag{8}$$

$$F(\boldsymbol{X}|\boldsymbol{K} = [\overrightarrow{Vel}]) = f_{\overrightarrow{Vel}}(\boldsymbol{X}) \tag{9}$$

$$f_{\overrightarrow{Vel}}(\boldsymbol{X}) = \sum_{j=1}^{M} FNs_j^{\overrightarrow{Vel}}(\boldsymbol{X}) \tag{10}$$

$$FNs_j^{\overrightarrow{Vel}}(\boldsymbol{X}) = \frac{\sqrt{\frac{1}{N}\sum_{t=1}^{N}\left\|Sim_{t,j}^{\overrightarrow{Vel}}(\boldsymbol{X}) - Obs_{t,j}^{\overrightarrow{Vel}}\right\|_2^2}}{\sqrt{\frac{1}{N}\sum_{t=1}^{N}\left\|Obs_{t,j}^{\overrightarrow{Vel}}\right\|_2^2}} \tag{11}$$

where, $NRMSE_j^{Tem}(\boldsymbol{X})$ and $FNs_j^{\overrightarrow{Vel}}(\boldsymbol{X})$ denote the Normalized Root Mean Square Error (NRMSE) of temperature (described in Eq. (8)), and normalized Fourier Norms (FNs) of velocity vectors (described in Eq. (11)) at locations $j$. $Sim_{t,j}^{\overrightarrow{Vel}}(\boldsymbol{X})$ and $Obs_{t,j}^{\overrightarrow{Vel}}$ denote the simulated velocity given a parameter vector $\boldsymbol{X}$ and observed velocity, respectively, at time step $t$ and location $j$. $Sim_{t,j}^{\overrightarrow{Vel}}(\boldsymbol{X})$ and $Obs_{t,j}^{\overrightarrow{Vel}}$ are 3-dimensional vector. $\|-\|_2$ in Eq. (11) is the Euclidean norm used to quantify the size of a vector.

The temperature and velocity data are taken at different depths of multiple stations, and their magnitude at different locations might be different due to spatial variation. Hence, the fitness at each location should be normalized before being summed into the objective function. For water temperature, Normalized Root Mean Square Error (NRMSE, as described in Eq. (8)) is used to quantify and normalize the error between the simulated and observed data. For velocity, normalized Fourier Norms of RMSE (FNs, as described in Eq. (11)) are used to

measure the error between the model-simulated and observed data (corresponding simulated and observed
velocity data points are three-dimensional vectors). The calculation of the Fourier Norm follows the description
in Beletsky et al. (2006), Huang et al. (2010), Paturi et al. (2014), and Råman Vinnå et al. (2017).

**2.5.2 DYNO for Model Calibration with Multiple Variables**

In the Cali-Both scenario, both temperature and velocity are calibrated simultaneously, which can be treated as a
bi-objective function problem. The objective function in the Cali-Both scenario (as shown in Eq. (12)) applies the
DYNO proposed in Eq. (3). The error functions for water temperature, i.e., $f_{Tem}(X)$, and velocity, i.e., $f_{\overrightarrow{Vel}}(X)$,
are the objective functions of the Cali-Tem scenario (Eq. (7)) and the Cali-Vel scenario (Eq. (10)), respectively.
The temperature and velocity errors are dynamically normalized with their upper and lower bounds during the
search of the optimization algorithm before being summed into a single objective function. The mathematical
formulation of the objective function in the Cali-Both Scenario (based on Eq. (3)) is as follows:

$$F(X|K = [Tem, \overrightarrow{Vel}]) = \frac{f_{Tem}(X) - f_{Tem}^{min}(X)}{f_{Tem}^{max}(X) - f_{Tem}^{min}(X)} + \frac{f_{\overrightarrow{Vel}}(X) - f_{\overrightarrow{Vel}}^{min}(X)}{f_{\overrightarrow{Vel}}^{max}(X) - f_{\overrightarrow{Vel}}^{min}(X)} \tag{12}$$

where the maximum and minimum of $f_{Tem}(X)$ and $f_{\overrightarrow{Vel}}(X)$ are updated after each optimization iteration (since
new parameter sets are sampled in each optimization iteration). As the number of iterations increases, the
denominators in Eq. (12) also increase since the optimization method finds better minimum objective function
values. Hence the individual objective function components (for each variable) scale dynamically to maintain an
approximately equal weight of the terms related to temperature and velocity.
As defined in Eq. (6) to Eq. (12), three calibration formulations are investigated in this study. Table 3
gives a summary of these calibration formulations.
**Table 3**. Summary of Objective function formulations for different calibration scenarios.

| Scenario Name | Variables used for calibration | Objective Function | Objective Function Formula |
|---|---|---|---|
| Cali-Tem | Temperature | $F(X|K = [Tem])$ | Eq. (6)~(8) |
| Cali-Vel | Velocity | $F(X|K = [\overrightarrow{Vel}])$ | Eq. (9)~(11) |
| Cali-Both | Temperature and Velocity | $F(X|K = [Tem, \overrightarrow{Vel}])$ | Eq. (12) |

**2.6 Implementation of DYNO with PODS**

In this section, we describe the implementation details for incorporating DYNO into a new efficient parallel
surrogate optimization algorithm, PODS (described in Fig. 2). PODS (Xia et al., 2021) is a parallel version of the
serial DYCORS (DYnamic COordinate search using Response Surface models) algorithm introduced by (Regis
and Shoemaker, 2013). DYCORS is an iterative surrogate method (such methods are sometimes also called
Response Surface Optimization methods, where cheap surrogates of the expensive objective are built to improve
optimization efficiency), designed for optimization of computationally expensive black-box functions within a
limited number of evaluations. DYCORS uses RBF (Radial Basis Function) as surrogates to efficiently explore
the parameter space and propose promising new solutions for expensive evaluation in each algorithm iteration.
The RBF-guided search methodology of DYCORS is designed for high-dimensional black-box optimization
within a limited number of evaluations of computationally expensive real objective functions. PODS, like
DYCORS, is designed for black-box optimization problems that are high-dimensional and computationally
expensive and have multiple local minima. Xia et al. (2021) show that PODS is considerably more efficient than
other parallel global optimization methods in obtaining good solutions with fewer objective function evaluations,
which is very important for expensive objective functions like hydrodynamics models. PODS parallelized the
serial DYCORS algorithm by following the Mater-worker framework (as shown in Fig. 2). This parallelization
strategy of the algorithm allows simultaneous function evaluations on multiple processors (cores) in batch mode,
which reduces the wall-clock time of the optimization process. This can greatly speedup the calibration of
computationally expensive models and make the calibration of some extremely expensive models computationally
tractable.

The PODS algorithm begins the optimization from an initial experiment design where a random initial

set of evaluation points are generated with the Latin Hypercube Design (LHD). These evaluation points are
distributed randomly to $P$ workers for simulation evaluations. Each worker will calculate the error/ objective
function of each variable $\{f_k(X_i)|k \in K\}$ based on Eq. (2) and return them back to the master. This step (W3 in
Fig. 2) for DYNO-based PODS is different from the original PODS. In the original PODS, only the final objective
function value (instead of the error of each variable) is returned to the master.

After the master collects the result of all the $P$ evaluations, it will add these new results into the history

list $\psi$ that saves all evaluation results found in previous iterations. The history list of $\psi$ is not in the original
PODS and it is necessary for the calculation of the DYNO objective function ($F(X|K)$, $X$) (as shown in Eq. (3)).
For instance, the maximum and minimum value of the error of each variable $f_k^{max}(X)$ and $f_k^{min}(X)$ are
dynamically changing with the increase of the history list $\psi$. The objective function value $F(X|K)$ for all
evaluations found in current and previous iterations need to be recalculated because of the update of $f_k^{max}(X)$ and
$f_k^{min}(X)$. And the best solution found so far is identified based on the newly calculated $F(X|K)$. When only one
variable (e.g., temperature or velocity) is considered in the objective function, the best solution is the evaluation
with the lowest error between the simulation output and observations of the variable considered. In cases where
multiple variables are considered in calibration, the best solution should be the evaluation with the smallest value
of $F(X|K)$ after considering the error of multiple variables (as shown in Eq. (3)). Because the objective function
value for all evaluations in $\psi$ changed after each iteration, the RBF is also rebuilt with these new objective
function values of all evaluated solutions ($F(X|K)$, $X$). The rebuilt RBF surrogate is used for the generation of the
evaluation points for the next iteration.

PODS with DYNO implementation uses the RBF surrogate in the same way as the original PODS does.

PODS first generates a large number of candidate points around the best solution found so far (refer to Section
2.2 in Xia et al (2021)). The algorithm then selects $P$ evaluation points from these candidate points based on their
estimated objective function $\hat{F}(X|K)$ based on the RBF surrogate and the minimum distance from all previous
evaluation solutions in $\psi$. A lower estimated objective function $\hat{F}(X|K)$ is better since it is more likely to lead to
solutions with lower objective function value. Meanwhile, candidate points that are far from previous solutions
are also preferred since they help the algorithm to explore regions of the solution domain that were not explored
in previous iterations. These unexplored regions could possibly be regions where better solutions are located. The
consideration of estimated objective function $\hat{F}(X|K)$ and distance information are both considered when
selecting the candidate points through a weighted score based on these two aspects. For detailed information on
the implementation of evaluation point selection criteria, one can refer to Section 2.3 in Xia et al (2021). The
selected $P$ evaluation points are then distributed to $P$ workers for evaluations, and the iteration loop continues
until the stopping criteria are met (e.g., the computing budget is finished.)
In summary, the implementation of DYNO affects the selection of the best solution found so far and also
the surrogate model (These steps are W3 and M1-5, as highlighted in green color in Fig. 2). We should highlight
that the fitting of the surrogate model is computationally inexpensive compared with the runtime of the expensive
objective function. Hence it does not affect the overall algorithm runtime.

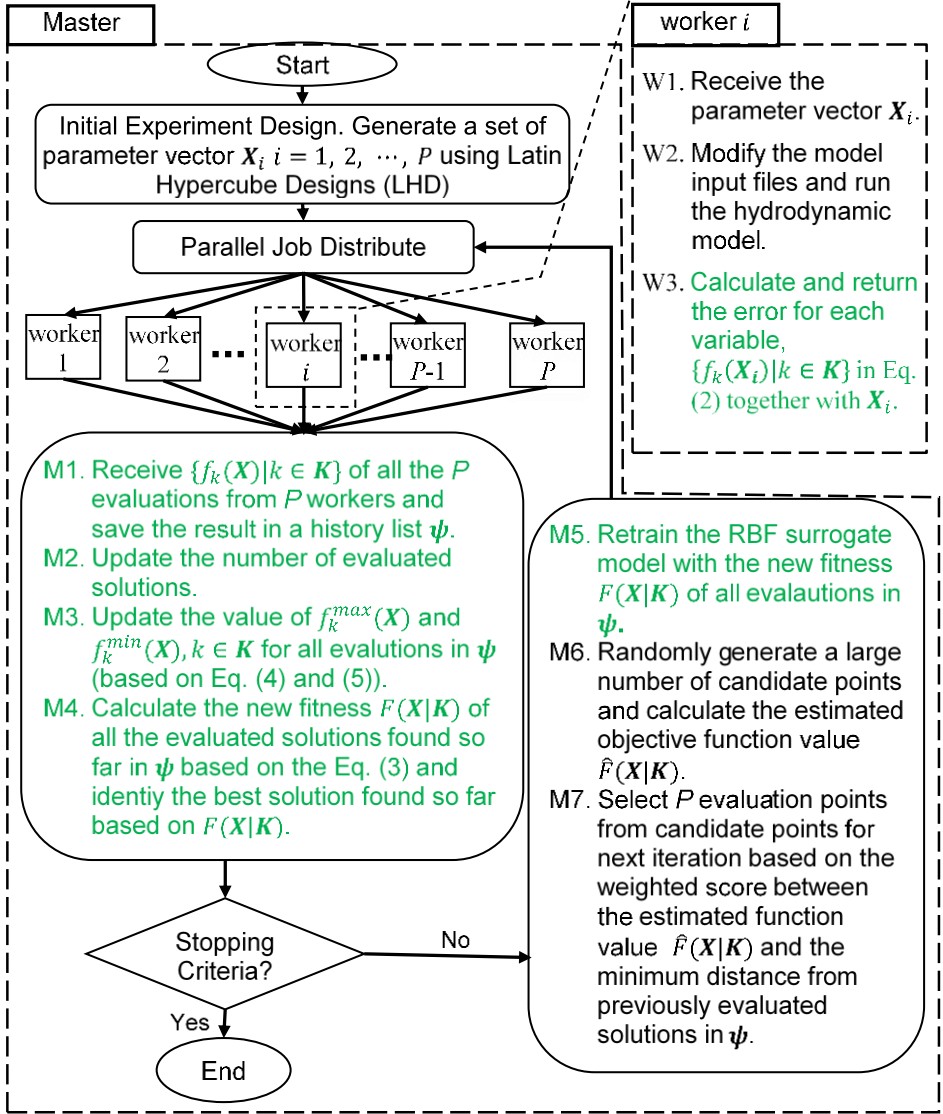

**Figure 2.** Diagram of the implementation of DYNO with the parallel algorithm PODS. $P$ is the number of
processors available. The green texts (i.e., steps W3, M1-5) are changes made on PODS to incorporate DYNO.
The rest part follows the original PODS method.
**2.7 Experiments Setup**
All computational experiments in this study are implemented on a single node on the National Supercomputer
Center (NSCC) of Singapore, which is a Linux-based platform with dual Intel Xeon E5-2690 v3 Processors, with
each node having 24 cores. Hence, we set the number of processors $P$ to be 24. Due to the stochastic nature of the
optimization algorithm (i.e., PODS) used in this study, multiple optimization runs are executed for each calibration
experiment in Table 3. Considering that the calibrated hydrodynamic model in this study is extremely expensive,
we perform three optimization trials for each calibration experiment (see Table 3 for a list of experiments).
Furthermore, to remove any initial sampling bias, each concurrent optimization trial for the three calibration
experiments is initialized with the same Latin Hypercube experimental design (so the calibration in each scenario
starts from the same initial solutions). We also investigated the performance of different forms of DYNO on the
Cali-Both scenario.
We set the same evaluation budget (i.e., the maximum number of hydrodynamic model runs) for each
trial and calibration scenario (i.e., Cali-Tem, Cali-Vel, and Cali-Both). The maximum number of hydrodynamic
model runs in each trial is 192, which is 8 iterations with 24 evaluations in each iteration. Our result indicates that
8 iterations are a sufficient calibration budget, as the calibration progress plot in Figure S1 shows that the
optimization experiments almost converged in the last few iterations.
The computational time of one simulation is approximately 5 hours on a windows desktop with a CPU
Intel Core i7-4790 processing unit. However, when running 24 simulations simultaneously on the multi-core
platform, the computational time gets longer because of the limited cache memory resources (as discussed in Xia
and Shoemaker (2022a)). Cache memory is a small amount of much faster memory than main memory. The wall-
clock time for one iteration with 24 cores simultaneously running is about 12 hours if using the default process
scheduling of the nonuniform memory access (NUMA) multi-core system. We used the mixed affinity scheduling
proposed by Xia and Shoemaker (2022a), and the wall-clock time is reduced to about 8 hours per iteration. The
mixed affinity scheduling changed the default affinity setting by setting a hard affinity on the simulation of each
PDE model (i.e., fixing the process of each PDE simulation to one core). This approach proved to be efficient for
memory usage and reduced the simulation time. More details about the mixed affinity scheduling and the NUMA
system can be found in the study of Xia and Shoemaker (2022a). Hence, the wall-clock time of each trial takes
about 64 hours (8 iteration×8 hours/iteration).
## 3. Numerical Results and Discussion
### 3.1 Comparison of Calibrating to Temperature and/or Velocity
### 3.1.1 Final Solutions in Goodness-of-fit Metrics
We first compare the three calibration formulations in terms of goodness-of-fit metrics for both temperature and
velocity. Table 4 summarizes this comparison for the three formulations, i.e., i) Cali-Tem, ii) Cali-Vel and iii)
Cali-Both (see definition in Table 3), with PODS used as the optimization algorithm and with a budget of 192
simulations.
The mean as well as the standard deviation of both temperature error $f_{Tem}(\boldsymbol{X}^*|\boldsymbol{K})$ (calculated as Eq. (7))
and velocity error $f_{\overline{Vel}}(\boldsymbol{X}^*|\boldsymbol{K})$ (Calculated as in Eq. (10)) over three trials are reported in Table 4, for all three
calibration scenarios. $\boldsymbol{X}^*$ in Table 4 denotes the optimal calibration solution obtained by PODS in each trial for a
given scenario (defined by the set of variables $\boldsymbol{K}$). The solution with the lowest variable error ($f_{Tem}(\boldsymbol{X}^*)$
or $f_{\overline{Vel}}(\boldsymbol{X}^*)$) is highlighted in bold in Table 4. Table 4 reports the variable errors of both temperature and velocity
for all formulations to understand the impact of ignoring or including a variable in the calibration formulation.
Please note that the temperature error, $f_{Tem}(X|K = [Tem])$, reported in Table 4, is exactly the calibration
objective function in the Cali-Tem scenario ($F(X|K = [Tem])$ as shown in Eq. (7)). Similarly, the velocity error
$f_{\overrightarrow{Vel}}(X|K = [Tem])$ is exactly the calibration objective function in the Cali-Vel scenario (i.e., $F(X|K = [\overrightarrow{Vel}])$
as shown in Eq. (10)). We use the word variable error instead of objective function value when referring to the
values in Table 4 in subsequent discussions since we are in part looking at the impact of using data from one
variable to predict another variable for which we don't have data. It is also worth mentioning that the value in
Table 4 is a sum of temperature or velocity error at multiple (in total 12) locations. Hence the error at each location
is smaller than the value in the table.

Table 4 shows that the solution obtained when calibrating to temperature observation only (Cali-Tem)

has smaller temperature errors but larger velocity errors than that if calibrating to velocity observation data only
(Cali-Vel). However, it is surprising that when calibrating to both temperature and velocity (Cali-Both), the
solution obtained by PODS has the lowest temperature and lowest velocity error compared with calibrating to
either temperature observation or velocity observation only. This might be because calibrating to temperature will
help to improve the fit of velocity and vice versa. This makes sense because water temperature and velocity are
two related variables in hydrodynamic modeling, and they affect each other. Velocity is the fundamental variable
of hydrodynamics with directional information not provided by temperature; temperature (via the heat flux model)
may also affect the velocity field since it affects water density. Our analyses here are based on physical models,
which are built based on physics laws and knowledge human have learned over hundreds of years. Our findings
here are in line with the study of Baracchini et al (2020), where they also suggested have both temperature and
velocity for a complete system calibration.
**Table 4.** Summary table of the solution obtained by PODS for each scenario (Cali-Both, Cali-Vel, and Cali-Tem).
$f_{Tem}(X^*|K)$ and $f_{\overrightarrow{Vel}}(X^*|K)$ are the temperature error $f_{Tem}(X^*)$ and velocity error $f_{\overrightarrow{Vel}}(X^*)$ (calculated in Eq. (7)
and Eq. (10), respectively, with the optimal solution $X^*$ obtained in each trial). The mean and standard deviation
of $f_{Tem}(X^*|K)$ and $f_{\overrightarrow{Vel}}(X^*|K)$ among three trials are reported. The variable error is bolded in each scenario when
the observation of the variable is included in the calibration in each scenario. (Some terms defined in Table 1)

| Scenarios | The composite error of each variable (Temperature or Velocity) | |
|---|---|---|
| | $f_{Tem}(X^*|K)$ Mean (Std.) | $f_{\overrightarrow{Vel}}(X^*|K)$ Mean (Std.) |
| Cali-Both $K = [Tem, \overrightarrow{Vel}]$ | **0.014 (0.003)** | **1.939 (0.165)** |
| Cali-Vel $K = [\overrightarrow{Vel}]$ | 0.087 (0.023) | 2.809 (0.319) |
| Cali-Tem $K = [Tem]$ | 0.024 (0.005) | 5.888 (1.435) |

**3.1.2 Visual Comparison of Calibration Errors**
The above analysis is based on the average variable error statistics only (i.e., $f_{Tem}(X^*|K)$ and $f_{\overrightarrow{Vel}}(X^*|K)$), of the
best results obtained from PODS (over multiple trials) for all calibration scenarios. In order to further analyze the
difference between calibration formulations (in terms of their effectiveness in calibrating both temperature and
velocity), we visually compare the best calibration solutions ($X^*$) obtained by PODS for each scenario, i.e., Cali-
Tem, Cali-Vel and Cali-Both. We select one representative optimal solution ($X^*$) from 3 trials in each scenario
for this comparison. An initial uncalibrated solution is included in the comparison. The parameter value of the
uncalibrated solution (in Table S1) uses the mean of the calibration range in Table 2.

The objective function value in terms of temperature and velocity composite error (over multiple

locations) ($f_{Tem}(X)$ and $f_{\overrightarrow{Vel}}(X)$, as formulated in Eq. (7) and (10), respectively) and the corresponding parameter
configuration ($X^*$) of the selected solution (among three trials) are plotted in Fig. 5 and reported in Table S1. In
general, the solution in the Cali-Both scenario is closer to the True solution than solutions in the other two
scenarios in terms of parameter values. Calibrated values proposed by the Cali-Both scenario are closest (relative
to other scenarios) to the True values for four parameters (i.e., $D_H^{back}$, $D_V^{back}$, $H_{Secchi}$, $c_H$). Moreover, besides the
Manning coefficients, the calibrated values proposed by the Cali-Both scenario are not the worst (relative to the
other two scenarios) for any other parameter. In contrast, calibrated values proposed by the Cali-Tem scenario are
worst (i.e., the parameter values are farthest from the true solution, relative to other scenarios) for five parameters
(i.e., $v_H^{back}$, $D_H^{back}$ $v_V^{back}$, $D_V^{back}$, $c_e$) and calibrated parameter values for the Cali-Vel scenarios are worst for
$L_{oz}$, $H_{Secchi}$, $c_H$. This indicates that calibrating to both temperature and velocity can help to prevent the value of
the 9 calibration parameters from being very far from the corresponding value for the True solution.

The horizontal velocity error $\Delta \overrightarrow{Vel}$ (2-dimensional) between simulated velocity $Sim_{t,j}^{\overline{Vel}}(X^*)$ and

observed velocity $Obs_{t,j}^{\overline{Vel}}$ (in the horizontal plane) is plotted as scatter plots of time-series in Fig. 3 (for all
calibration scenarios). The temperature error $\Delta Tem$ between simulation temperature $Sim_{t,j}^{Tem}(X^*)$ and observed
temperature $Obs_{t,j}^{Tem}$ is plotted as a time series (for each calibration scenario) in Fig. 4.

The error plots for the two sampling locations at multiple depths (i.e., surface layers of station STN. A1

and STN. B1 as shown in Fig. 1 (a)) are visualized in Fig. 3 and 4 (for one year). Since the velocity error $\Delta \overrightarrow{Vel}$ at
a particular time and location is a vector (and not a scalar like temperature) and velocity error in 3 dimensions (for
a time-series) is hard to represent visually, Fig. 3 only plots the velocity error (for one year) $\Delta \overrightarrow{Vel}$ in the horizontal
plane (i.e., X and Y directions only). Moreover, each dot represents the error at one point in time within the study
period.

Figure 3 plots the difference between the simulated velocity (for the optimized parameter values obtained

from Cali-Tem (red scatter points), Cali-Vel (black scatter points), and Cali-Both (blue scatter points) scenarios)
and observed velocity.  Ideally, the error for each scatter point should be zero, i.e., at the intersection of the two
lines.  Figure 3 illustrates that calibrating to temperature data only (red scatter plot) results in a larger velocity
error $\Delta \overrightarrow{Vel}$, relative to velocity error when calibrating to velocity data only (Cali-Vel scenario, i.e., black scatter
plot) or to both velocity and temperature data (Cali-Both scenario, i.e., blue scatter plot). Figure 3 also shows that
solutions of all the three scenarios improved the temperature fit compared with the initial solution, which
demonstrates the effectiveness of the optimization calibration.

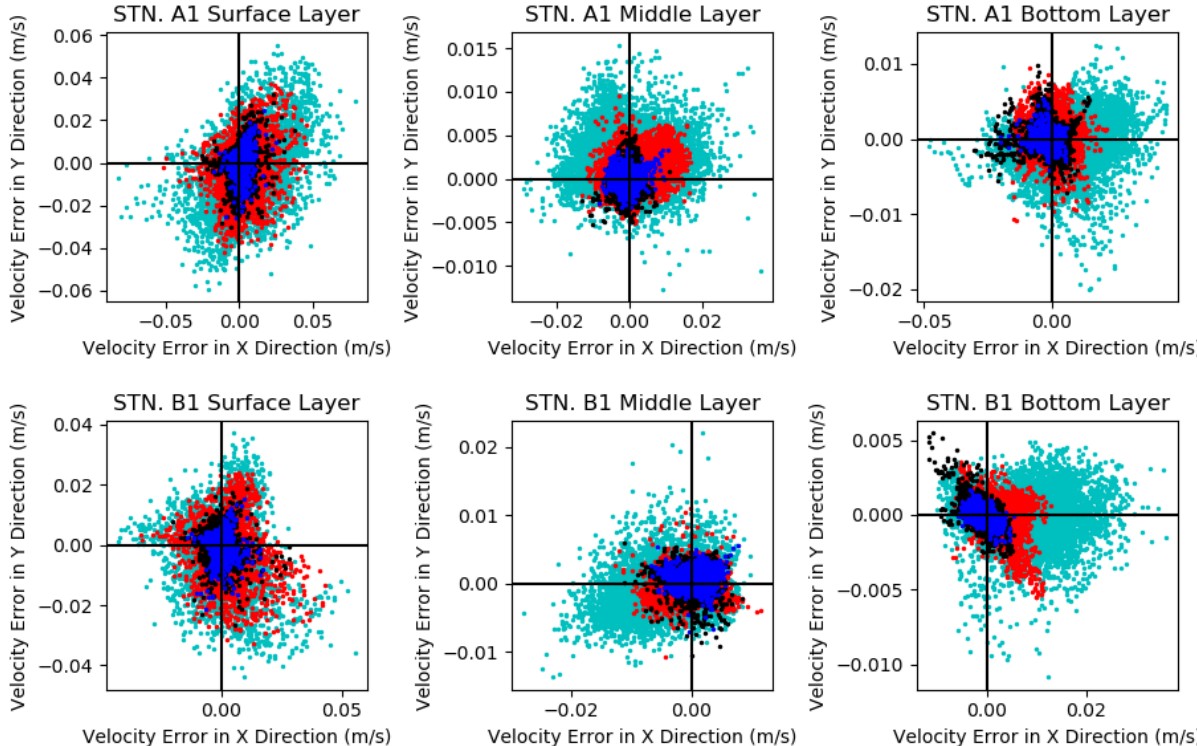


**Figure 3.** Scatter plot of velocity error $\Delta\overrightarrow{Vel}$ in horizontal (X and Y direction) between simulated velocity $Sim_{t,j}^{\overrightarrow{Vel}}(X^*)$ and observed velocity $Obs_{t,j}^{\overrightarrow{Vel}}$ at location $j$. Each dot denotes the velocity error $\Delta\overrightarrow{Vel}$ of location $j$ at one time step. $j$ = surface layer of STN. A1 for upper panel and $j$ = STN. B1 for lower panel. $X^*$ is the optimal solution found by PODS in each scenario: Cali-Tem (red dots); Cali-Vel (black dots) and Cali-Both (blue dots) as listed in Table S1. The "True" solution is on or near the intersection of the two perpendicular black lines. An initial uncalibrated solution (cyan dots) is plotted for reference.

Figure 4 shows the temperature error of solutions from three different calibration scenarios: Cali-Tem (red time-series), Cali-Vel (black time-series) and Cali-Both scenarios (blue time-series). The errors between simulated and observed water temperature at the surface, middle and bottom layers of two stations (STN. A1 and STN B1) are plotted. In general, the temperature error of the solution in Cali-Both scenario is generally close to zero °C for all the layers and stations shown. The solution in Cali-Tem scenario also got temperature errors close to zero °C at the middle and bottom layer at STN. A1, but it has larger temperature errors than the solution in the Cali-Both at the surface layer of STN. A1 and all layers of STN. B1. The solution in the Cali-Vel scenario generally overestimated the water temperature in all locations (i.e., all the surface, middle and bottom layers at both stations). The temperature error of the solution in the Cali-Vel scenario is much larger than that of the solution in Cali-Tem and Cali-Both scenarios in the middle and bottom layers of both stations. The temperature error at most times, for the Cali-Vel scenario, is greater than 0.1 °C. This might be because both the Stanton and Dalton numbers are underestimated in the Cali-Vel scenario when compared with the True solution ($X^R$) (As shown in Fig. 5). The Dalton number $C_e$ affects the evaporative heat flux modeling and the Stanton number $C_H$ influences the convective heat flux modeling in the Delft3D-FLOW model (Hydraulics, 2006). For the solution in Cali-Vel, a smaller Stanton number $C_H$ (shown in Fig. 5) might lead to underestimated convective heat flux, which will lead to the overestimation of the water temperature. In summary, calibrating to temperature and velocity (i.e., Cali-Both) give the best solution in terms of temperature error compared with calibrating to temperature or velocity only (i.e., Cali-Tem or Cali-Vel). Calibrating to velocity only (Cali-Vel) gives the worst result in terms of

temperature fit. Simulation of vertical temperature, vertical velocity, vertical eddy diffusivity, and vertical eddy viscosity (Fig. S2-S5) also shows that the solution in the Cali-Both scenario is much better than solutions in the Cali-Tem and Cali-Vel scenario. For example, the solution in the Cali-Both scenario can almost capture the vertical time-varying temperature profiles of the true solution. In contrast, calibrating to one variable did not fully capture the vertical time-varying temperature profiles (e.g., April-May for the Cali-Tem scenario; Mar-May and Aug-Sep for the Cali-Both scenario in Fig. S2.) The solution in the Cali-Both scenario also gives much smaller vertical velocity, eddy diffusivity, and eddy viscosity error than solutions in the other two scenarios (in Fig. S3-S5). The result indicates that using both temperature and velocity data in model calibration also helps to improve the complex time-varying vertical mixing behavior.

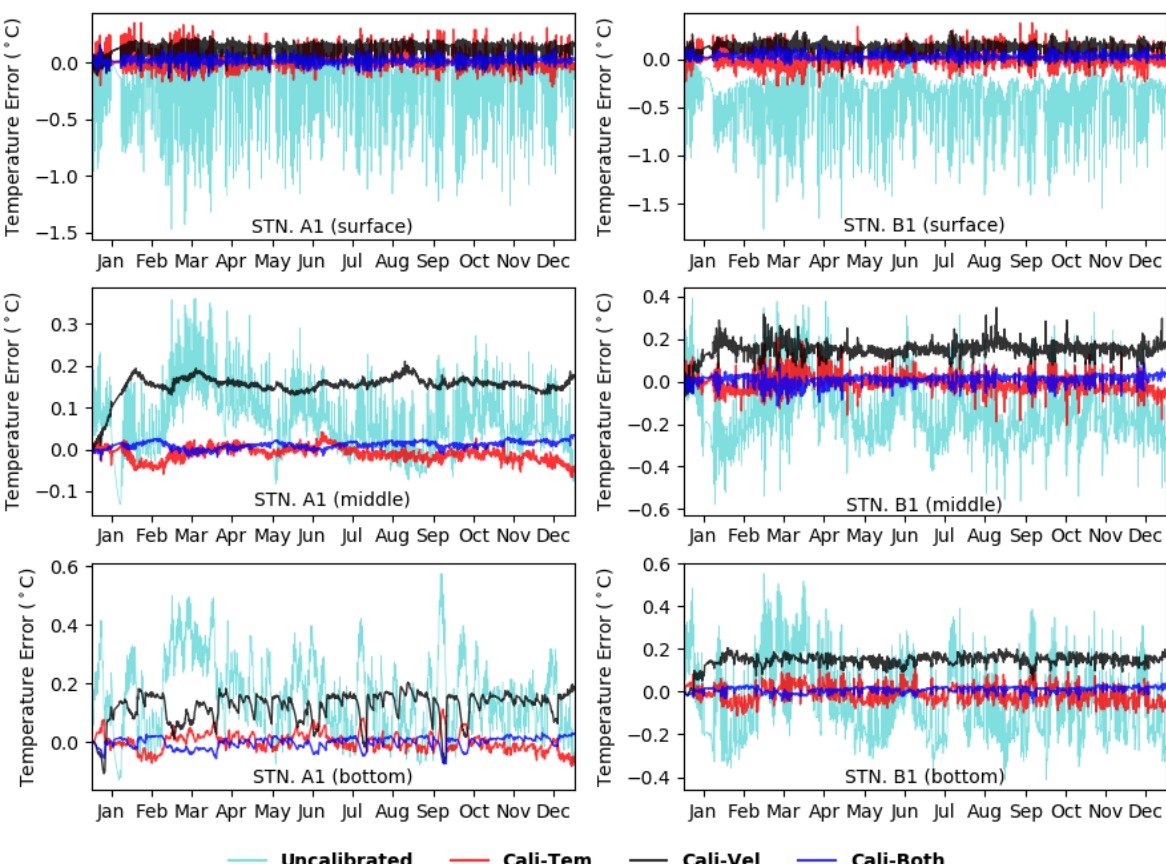

**Figure 4.** Time-series plots of temperature error $\Delta T$ between simulated water temperature and $Sim_{t,j}^{Tem}(X^*)$ and observed water temperature ($Obs_{t,j}^{Tem}$) at location $j$ where $j$ = surface layer of STN4 for left panel and $j$ = STN1 for the right panel. $X^*$ is the optimal solution found by P-DYCORS in each scenario: Cali-Tem (Red lines); Cali-Vel (Black lines) and Cali-Both (blue lines) as listed in Table S1. An initial uncalibrated solution (cyan lines) is plotted for reference.

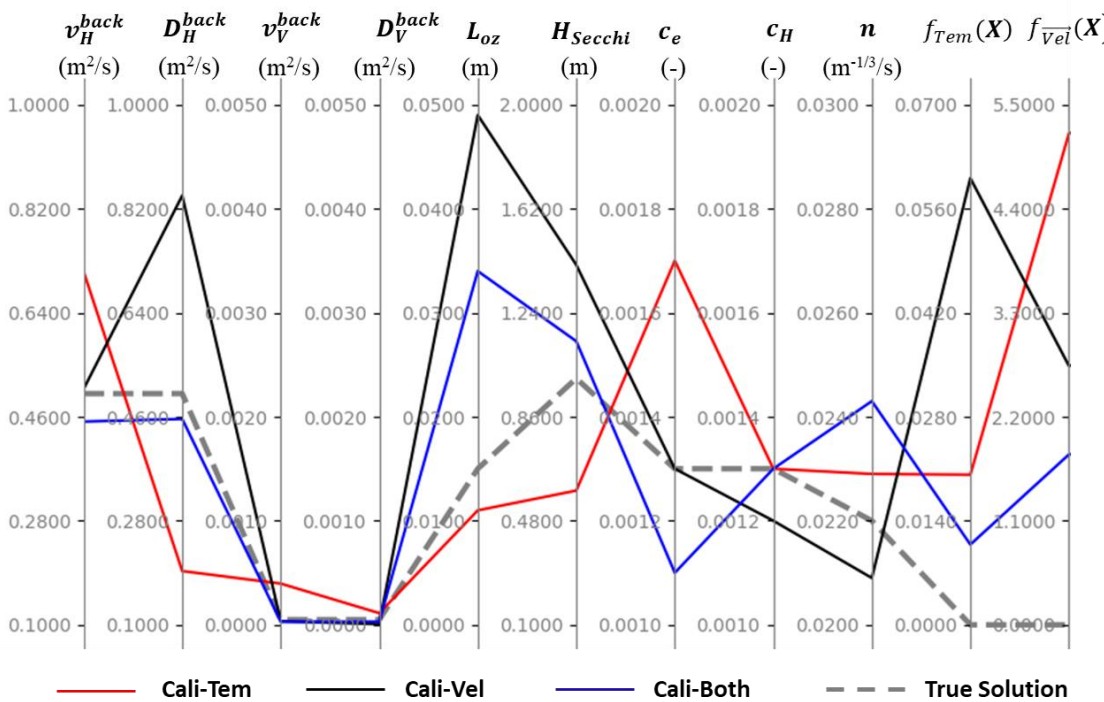

**Figure 5.** The parallel axis plot for the parameter value and the composite error of temperature and velocity of calibration solutions under different scenarios (Cali-Tem, Cali-Vel, and Cali-Both). True solution defined in Table 2 is given for reference. Smaller variable errors ($f_{Tem}(\boldsymbol{X})$ (see Eq. (7)) and $f_{\overline{Vel}}(\boldsymbol{X})$ (see Eq. (10))) are better, and the variable errors of the true solution ($\boldsymbol{X}^R$) are zero (for both $f_{Tem}(\boldsymbol{X})$ and $f_{\overline{Vel}}(\boldsymbol{X})$). The parameter symbols are defined in Table 2.

### 3.2 Optimization Search Dynamics under Different Calibration Scenarios

We further analyze the calibration progress of PODS for Cali-Tem, Cali-Vel and Cali-Both, to understand the calibration convergence speeds of the three formulations. The purpose of the calibration progress analysis is to visualize the improvement in calibration quality of both temperature and velocity variables from the LHD, for all three formulations. As discussed in the experiment setup section, we conducted 8 iterations of the optimization search. This is a reasonable number of iterations for our case, given that 1) the problem is computationally expensive (one experiment takes about 64 hours to run and there are 9 experiments) and 2) the calibration progress plot in iterations (Figure S1) indicates that the optimization search almost converged in 8 iterations.

Figure 6 plots the calibration progress of the three formulations (i.e., Cali-Tem, Cali-Vel and Cali-Both) using PODS. Each subplot within Fig. 6, corresponds to the different concurrent optimization trials (i.e., trials of the stochastic optimization method using the same initial points from LHD) for each formulation. The best solutions are near the origin of each graph. Moreover, Fig. 6 plots the progress (quantified by visualizing both temperature and velocity errors) of the best solution found (measured in terms of the objective function value in each calibration scenario) during the search. Figure 6 indicates that when calibrating to temperature or velocity only, the optimization search cannot guarantee the improvement of the fit of another variable. For example, in Fig. 6 (a), when calibrating to velocity only, the temperature error of the best solution found at the end of the optimization search stage is worse than the temperature error of the best solution found after the initial LHD, even though there is improvement in terms of velocity fit. Similarly, when calibrating to temperature only, the improvement in velocity fit is also not significant (for instance, in Fig. 6 (a)). When calibrating to the fit of both

temperature and velocity using the DYNO formulation, the fit of both temperature and velocity improves in all
trials, and the improvement remains balanced during the optimization search. Figure 6 also indicates that the final
solution found in Cali-Both scenarios dominates the best solution found by PODS in Cali-Tem and Cali-Vel in
terms of both temperature and velocity fit.

Figure 6 also shows that when calibrating to one variable, the optimization search is easily convergent

(i.e., the best solution does not continue improving after a few iterations even in terms of the fit of the variable
considered in calibration). For example, in Fig. 6(a), when calibrating to temperature only, the best solution in
terms of temperature error does not improve much (in the last few iterations). The reason might be that when the
velocity error is large, it is less likely that the temperature fit would be improved further. In contrast, when
calibrating to both temperature and velocity, the optimization search continues improving in both the temperature
and velocity fit. Hence only considering one variable in the calibration, it is difficult to get a solution that has
small (or close to zero) errors of the variable considered. We should also highlight that even though the
optimization gets a solution that has zero error in one variable, it does not mean that the error of another variable
would be zero. The reason is that only the observation in part of the simulation space is used for calibration (not
the observation data at each grid and each time step of the simulation space are used to calculate the temperature
error). So the temperature error may be 0 at these observation locations, while the temperature error is not 0 at
other locations where observation is not used in calibration. In this case, getting a temperature error 0 at
observation locations cannot guarantee the velocity error is 0.

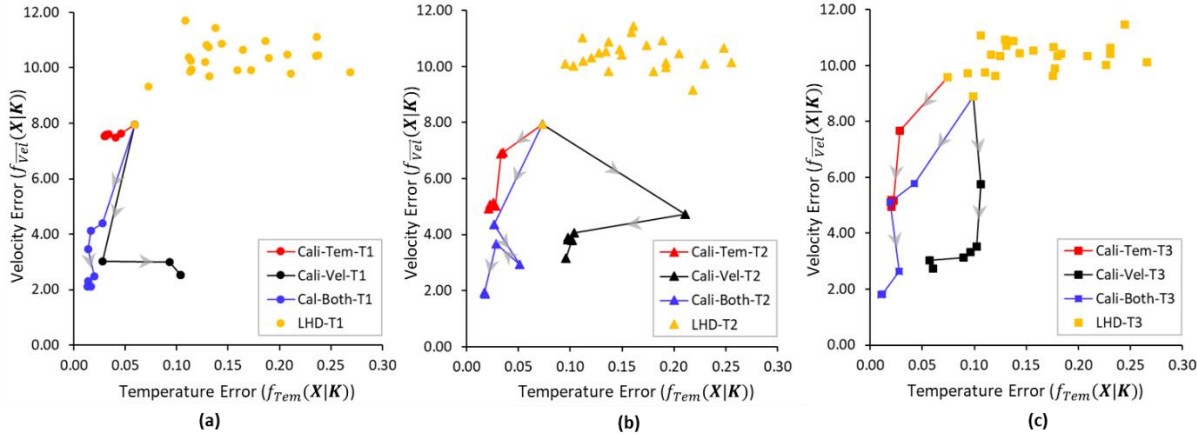


**Figure 6.** Calibration progress plot of the best solution found (in terms of objective function value) during
optimization search by PODS when calibrating to temperature only (Cali-Tem), calibrating to velocity only (Cali-
Vel), and calibrating to both temperature and velocity (Cali-Both). Three random trials (i.e., T1, T2, and T3) are
plotted in (a), (b), and (c). Lower velocity and temperature error are better. The yellow makers are evaluation
point in initial experiment design using LHD. Besides solutions in LHD, only the best solution in each of the
optimization iterations is plotted (i.e., makers lined with lines). The line links the best previous solution in one
iteration to the best solution in next iteration. The arrow indicates the direction from the previous solution to the
next solution.

It is also important to understand the 'frequency' or likelihood with which PODS can find good

temperature and velocity calibrations via the three different formulations proposed in this study. Hence, we also
do a comparative frequency analysis of the errors (for velocity or temperature) of all evaluated points ($X_i, i =$
$1, ..., 3 \times N_{max}$) from all trials (3 trials) of PODS when using different calibration formulations (see Table 3). The
purpose of this frequency analysis is to understand the likelihood with which the three different formulations can
obtain good velocity and temperature calibrations. The frequency analysis results are presented in Fig. 7 via
visualizations of empirical histograms of both velocity error and temperature error (from all solutions of 3 trials
of PODS) for each calibration scenario.

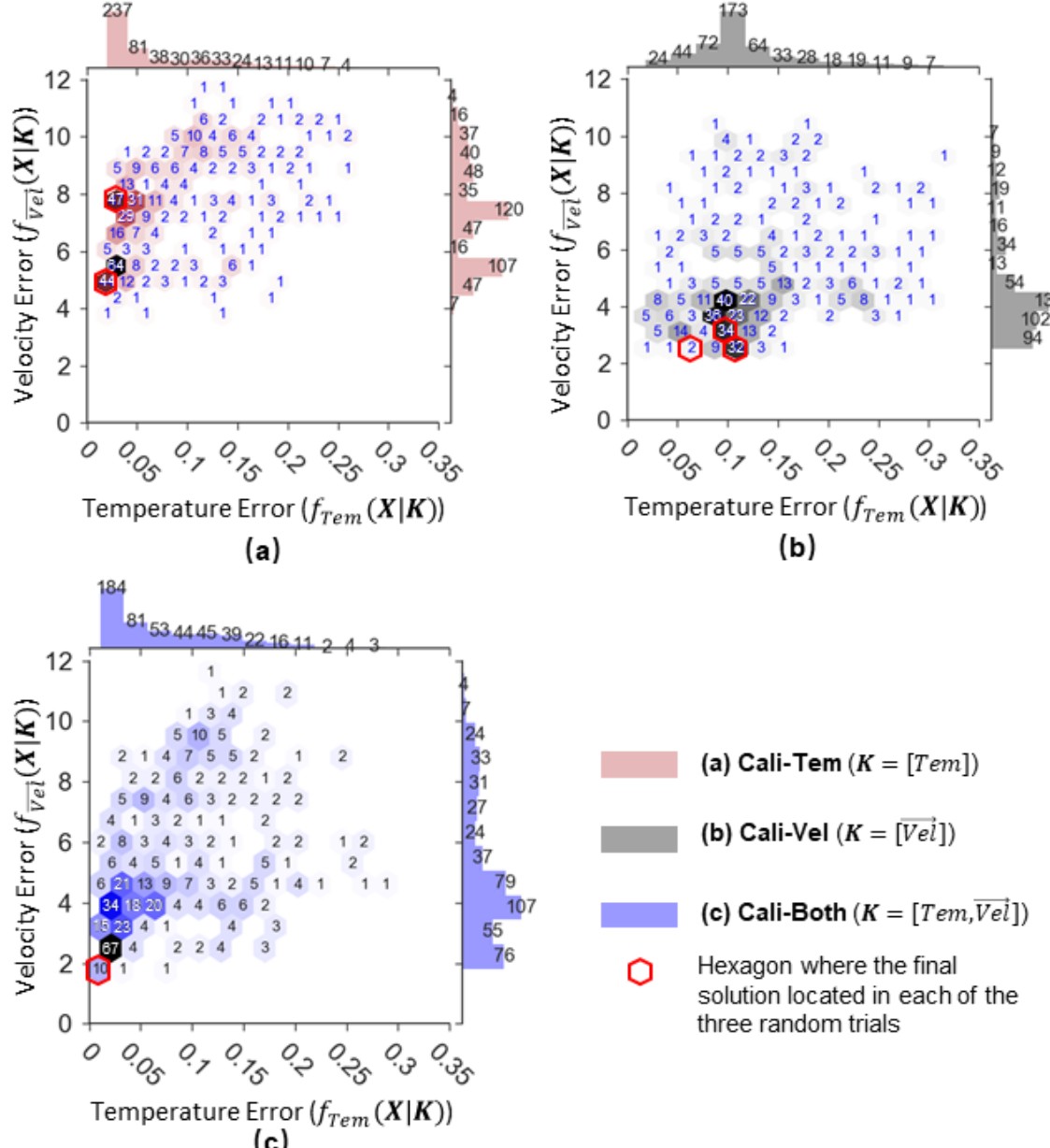

**Figure 7.** Distribution plot of all the evaluated points found by PODS (over 3 trials) in terms of temperature
composite error $f_{Tem}(X|K)$ and velocity composite error $f_{\overline{Vel}}(X|K)$ in each scenario: Cali-Tem ($K = [Tem]$),
Cali-Vel ($K = [\overrightarrow{Vel}]$), and Cali-Both ($K = [Tem, \overrightarrow{Vel}]$). The number inside each hexagon represent the number
of evaluated points located in that hexagon (e.g. with the combination temperature and velocity error associated
with the corresponding values on the axes.) Darker color in hexagon means larger number of evaluated points is
located in that hexagon. The bar plot along the upper x axis ($f_{Tem}(X|K)$) are the distribution of the evaluation
points in terms of temperature error only. The bar plot along y axis ($f_{\overline{Vel}}(X|K)$) are the distribution of the
evaluation points in terms of velocity error only. The number above the bar shows how many evaluated points
located in that bin. Smaller error ($f_{Tem}(X|K)$ or $f_{\overline{Vel}}(X|K)$) is better. The true solution ($f_{Tem}(X^R|K), f_{\overline{Vel}}(X^R|K)$)
is the origin of each subplot.
Figure 7 plots the error distribution of all the evaluated points over three trials (576 evaluations) for each
scenario: Cali-Tem ($K = [Tem]$), Cali-Vel ($K = [\overrightarrow{Vel}]$), and Cali-Both ($K = [Tem, \overrightarrow{Vel}]$). The different subplots
in Fig. 7 provide a visualization of the velocity (vertical axis) and temperature (horizontal axis) error distribution
via hexagonal bin (hexbin) plots (inside the square) and error histograms (outside the square) for each of the
calibration scenarios. The number inside each hexbin denotes the number of evaluated points (for that combination
of temperature error and velocity error) located in that hexbin. Furthermore, the hexbin with a larger number of
evaluated points is highlighted with a darker color shade. The temperature histogram columns (above the square)
represent the sum of all the hexbins inside the square directly beneath the number in the column. For the velocity
histogram (on the right side of square), the column height depends on the sum of all the hexbins in the row to the
left of the number.

The temperature and error velocity distribution visualizations of Fig. 7 clearly show that calibrating to

both temperature and velocity data (see Fig. 7 (c), i.e., error distribution for the Cali-Both scenario), provides good
temperature and velocity calibrations with a higher frequency. Figure 7 (c) shows that it is highly likely that both
temperature and velocity errors are lower (indicated by darker hexbins with temperature error $f_{Tem}(X|K)$ less
than 0.05 and velocity error $f_{\overline{Vel}}(X|K)$ less than 4). Consequently, Fig. 7(c) also illustrates that the newly
proposed DYNO (see Eq. (3)) works effectively, in this case, to calibrate multiple variables simultaneously.

Figure 7 also illustrates that it is better to calibrate the hypothetical hydrodynamic model to velocity data

rather than temperature data (see Fig. 7(a) and Fig. 7(b)) (if data for both variables is not available). Figure 7(a)
indicates that calibrating to temperature only (i.e., the Cali-Tem scenario) results in a high chance that velocity
error would be high (see the velocity error histogram in Fig. 7(a)). However, Fig. 7(b) illustrates that the errors in
temperature when calibrating to velocity only (Cali-Vel) are likely to be relatively small in magnitude (see the
temperature error histogram of Fig. 7(b)).

From the above discussion, we can conclude that calibrating to both temperature and velocity data with

the newly proposed DYNO (implemented within the efficient surrogate algorithm PODS) is effective in obtaining
a balanced calibration of both temperature and velocity variables. In real-world lake hydrodynamic applications,
if available, both temperature and velocity data should be used for lake hydrodynamic model calibration.
However, the very common practice of calibrating only to temperature data is shown to be unable to reproduce
the flow dynamics well. This supports the extra effort and expense to collect velocity data is expected to give a
beneficial effect.
**3.3 Impact of Different Forms of Normalization on the Performance of DYNO**
This section investigates the impact of using different forms of normalization in the new objective function DYNO
on optimization search performance. In Eq. (3), the error of each variable is normalized by the maximum and
minimum values $f_k^{max}(X)$ and $f_k^{min}(X)$ of $f_k(X)$ among all the evaluations evaluated so far. One concern of
using the maximum value $f_k^{max}(X)$ is that the objective function can be affected by extremely bad evaluation
points. Another approach is to use the median value $f_k^{median}(X)$ of $f_k(X)$ among all the evaluations evaluated so
far as a replacement of $f_k^{max}(X)$ to normalize the error of each variable. We refer to DYNO using the median
value $f_k^{median}(X)$ as DYNO-N2 (as shown in Eq. (13)) to differentiate it from DYNO using the maximum value
$f_k^{max}(X)$ (as shown in Eq. (3)), which we refer to as DYNO-N1 in the following text.
$$F(X|K) = \sum_{k \in K} \frac{f_k(X) - f_k^{min}(X)}{f_k^{median}(X) - f_k^{min}(X)} \qquad (13)$$
$$f_k^{median}(X) = \text{med}\{f_k(X) \ for \ all \ X \in \psi\} \qquad (14)$$

where $f_k^{median}(\boldsymbol{X})$ and $f_k^{min}(\boldsymbol{X})$ are the median and minimum values of $f_k(\boldsymbol{X})$ among all the evaluations evaluated so far, and hence they are updated dynamically in each iteration during optimization.

The implementation of DYNO-N2 is similar to the implementation of DYNO-N1 (Eq. (3)). The only change is replacing the calculation related to $f_k^{max}(\boldsymbol{X})$ with $f_k^{median}(\boldsymbol{X})$. We tested the relative efficacies of DYNO-N1 and DYNO-N2, by comparing three calibration trials, of each DYNO variant (using PODS), where each concurrent calibration trial was initialized using the same LHD. Figure 8 shows the progress of PODS with the two forms of DYNO as the objective functions. Figure 8 is similar in design to Fig. 6, and indicates that both forms of DYNO are able to balance the calibration on temperature and velocity. There are two trials where PODS with DYNO-N1 (using $f_k^{max}(\boldsymbol{X})$ for normalization) found a better solution than PODS with DYNO-N2 (using $f_k^{median}(\boldsymbol{X})$ for normalization).

The results here indicate that DYNO-N1 seems not adversely affected by the bad solution. A reason for this may be that PODS typically do not generate extremely bad solutions (i.e., outlier solutions with extremely large errors), since algorithm search is concentrated around the best solution found so far. However, if other optimization algorithms are used for calibration, especially algorithms that explore the search space more, there might be a higher likelihood of encountering outlier /extremely bad solutions during optimization search. Consequently, the performance of such an algorithm with DYNO-N2 might be better than with DYNO-N1, which might need further investigation. The outlier solutions here mean solutions (obtained during the optimization search phrase) that have much larger errors than other solutions found so far. Outlier or extremely bad solutions are also likely to happen for calibration problems where the model output is very sensitive to the calibration parameters (i.e., a small change in model parameters can cause huge changes in the model output that leads to much worse solutions).

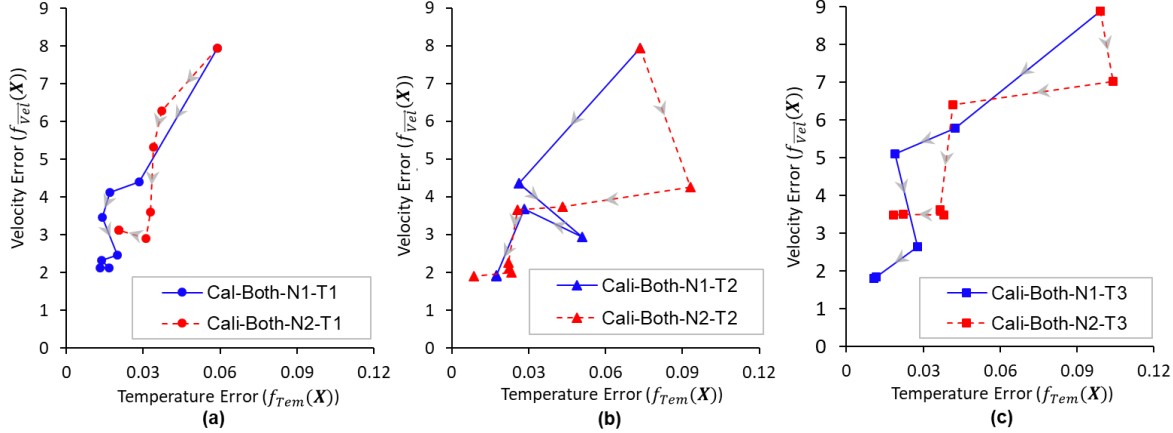

**Figure 8.** Calibration progress plot in terms of the best solution found during optimization search when using DYNO-N1 and DYNO-N2 as the objective function. Three random trials (T1, T2, and T3) are plotted in (a), (b), and (c). Lower velocity and temperature error are better. Figure 8 uses the same format as Figure 6.

**3.4 Value of Velocity Measures in 3D Lake Model Calibration**

High quality hydrodynamic simulations (e.g., thermal structure, current velocities, flow advection and vertical mixing) are vital for accurate spatial modelling of water quality in lakes. The hydrodynamic process influences the transport & production or transformation of biological and chemical components. Hence, if the simulation of flow dynamics is not adequately accurate, there is no way to achieve accuracy in the simulation of water quality.

Previous studies use mostly temperature observations for the 3D lake hydrodynamic model calibration. Whereas,
velocity data is less commonly used compared with temperature data for model calibration.
Our results in section 3.1 indicate that calibrating to temperature data only cannot guarantee accuracy in
velocity simulation in our case. Not using velocity data in model calibration (i.e., using temperature data in model
calibration only) thus, may lead to large velocity errors (as indicated in the Fig. 3). The inclusion of velocity
measurements in calibration not only reduces velocity error but also helps improve the temperature fit. For
example, in Fig. 4, when calibrating to both temperature and velocity data, the temperature error is smaller than
the temperature error when calibrating to temperature data only. This is most obvious in the surface layers of both
STN. A1 and STN. B1, where the temperature error when calibrating to both temperature and velocity (i.e., Cali-
Both) is much smaller compared to calibrating to temperature only (i.e., Cali-Vel). The better result (better fit of
temperature as well as velocity) in Cali-Both demonstrates the effectiveness of using velocity measures in 3D
hydrodynamic lake model calibration. The comparison of calibrated parameter values in Cali-Both and Cali-Tem
scenarios (in Fig. 5) also demonstrates the value of using velocity data besides temperature data in model
calibration. In Fig. 5, we can see that the calibrated value of viscosity and diffusivity parameters in Cali-Both is
much closer to the true value than that in Cali-Tem. This shows that the use of velocity measures helps to improve
the calibration of these viscosity and diffusivity parameters. Our analysis is based on synthetic observation data
from the physical model since we do not have real velocity measurements. These physical models are based on
physics laws. The analysis from modelling can provide some implications for the real-world situation. Hence, it
is worthwhile to repeat the analysis based on real data if there are real velocity measurements available in the
future.
The risk of using only temperature data without velocity data, even for accurately simulating water
temperature, is that temperature simulation is affected by both the flow dynamics and the heat transfer process.
The fit of temperature data is a result of the combination of these two processes. However, the fit of the
temperature data cannot guarantee accurate simulation of each of the processes, though accurate simulation of
each process does guarantee the fit of temperature data. The velocity observation hence is valuable to help improve
the flow dynamics simulation of the model, which is not only important for temperature simulation but also for
other water quality substances simulation (e.g., biological and chemical components). Our research implication
of the use of velocity observations is also in line with the study of Baracchini et al. (2020), where they also suggest
having both temperature and current velocity for complete system calibration.
**3.5 Possibilities for Other Applications**
In this study, we only demonstrate how DYNO can be incorporated into PODS parallel surrogate global
optimization algorithm. (see section 2.6). However, the new objective function DYNO could also be easily
utilized with other heuristic optimization methods (e.g., serial or parallel versions of Genetic Algorithm (Davis,
1991) and Differential Evolution (Tasoulis et al., 2004)) for effectively calibrating other multi-variable calibration
problems. We have not provided a precise methodology for incorporating DYNO into other optimization methods
though, since the incorporation of DYNO depends on the structure of an optimization method, and structures of
optimization methods vary a lot. We did illustrate in section 2.6 and Figure 3 on how components of parallel
PODS are modified in order to use DYNO. Other optimization methods could be modified in a similar way to
incorporate DYNO for use in multi-variable calibration.

Also, there are numerous other model calibration paradigms in general hydrology and water resources (besides the hydrodynamic model calibration) where simultaneous multi-variable and multi-site calibrations are required. Some examples of such multi-variable & multi-site calibration problems include watershed model calibration (Franco et al., 2020; Odusanya et al., 2019), and seawater intrusion model calibration (Coulon et al., 2021), and water quality model calibration (Xia and Shoemaker, 2021; Xia and Shoemaker, 2022b) etc. In these problems, there are usually multiple constituents (e.g., substances) to be calibrated and the observations are usually available at multiple locations. Our new DYNO can potentially be used to calibrate them simultaneously. A popular calibration strategy for such problems in general hydrology is to use multi-objective calibration where it is assumed that a trade-off exists between multiple hydrologic responses (e.g., high flow, low flow, water balance, water quality etc.)

Using multi-objective algorithms, however, for calibrating hydrologic and watershed quality models may not be the most suited strategy for some case studies because i) multi-objective calibration can be computationally intensive if underlying simulations are computationally expensive and ii) meaningful trade-offs between different objectives may not exist. Kollat et al. (2012) demonstrate that prior multi-objective calibration exercises may have over-reported the number of meaningful trade-offs in hydrologic model calibration. DYNO is a reasonable alternative to classical multi-objective calibration in calibration problems where the trade-off between multiple component calibration objectives is not significant, because i) a balance between multiple constituent objectives is maintained with DYNO and ii) a single objective algorithm can be used with DYNO, which is computationally more efficient than a multi-objective algorithm. This is especially true for multi-constituent watershed model calibration problems where the achievable objective functions ranges for different constituents (e.g., flow, sediment, phosphorus etc.) are quite different. Multiple prior studies (Moriasi et al., 2012; Moriasi et al., 2015) highlight that achievable ranges of statistical calibration measures (e.g., Nash Sutcliffe Efficiency (NSE), bias etc.) are significantly different for different constituents (e.g., streamflow, sediment, total phosphorus etc.). Moriasi et al. (2015) note that in most watershed model case studies, the achievable range of NSE for streamflow is higher than the achievable range for total phosphorus. Hence, DYNO may be extremely effective in balancing simultaneous calibration of streamflow and phosphorus for such case studies. We believe that there is immense potential in the application of DYNO for multi-constituent watershed model calibration.

DYNO is also applicable to multi-constituent calibration problems where sampling locations and temporal frequencies for the different constituents are different. For instance, in real world hydrodynamic settings, it is very likely that sampling locations and frequencies of temperature and velocity observations are different. This is also true for watershed sampling settings, where sampling locations and frequencies for water quality (e.g., phosphorus) constituents are, typically, less than sampling distributions of streamflow. While the synthetic experiments of this study assume identical sampling locations & frequencies for temperature and velocity, DYNO requires the observations of multiple constituents (e.g., temperature and velocity in our case). It is worth mentioning that DYNO does not require the same number of locations or same time frequency for different observation constituents. This is because DYNO first calculates the composite error of each constituent separately and then normalizes the composite error of each constituent dynamically, to balance the calibration of each constituent. This feature of DYNO allows it to be used in cases where different constituents are measured in different locations or time frequencies.

## 3.6 Future Work

Our analysis of the role of temperature and velocity measurements in 3D hydrodynamic model calibration is based on synthetic observation data generated from models. We do think it is worthwhile to further investigate the role of temperature and velocity measurements in hydrodynamic model calibration if there are velocity measurements available in the future. Moreover, the synthetic observation data used in our analysis did not account for the measurement uncertainty of observation data. Further investigations related to the impact of measurement errors on the calibration setup proposed here will also be beneficial. It is important to note that the measurement uncertainty and distribution of different variables could be different (and thus, our new objective function formulation DYNO could be very useful in balancing the calibration process in such a scenario). For example, Baracchini et al. (2020) reported that the measured and computed velocity value (in the magnitude of 1 cm s$^{-1}$ for velocity in hypolimnion layer) is close to velocity measurement uncertainty 0.8cm s$^{-1}$ (the velocity measurement instrument precision) while the computed and measured temperature value is an order of magnitudes larger relative to temperature measurement uncertainty in their study. The difference in terms of measurement accuracy and measurement value could lead to a different magnitude of error function value for each variable (temperature or velocity). (In their study, the error function is the square of temperature (or velocity) difference between computed and measured value divided by the observational uncertainty). Baracchini et al. (2020) pointed out that such discrepancy hinders the use of different kinds of data (e.g., temperature and velocity) simultaneously because the impact of velocity on the cost function is almost negligible compared with temperature observations. Hence, they carried out a separate discussion for both types of observation data. Their argument is true if the calibration objective function is a sum of temperature or velocity's error function with a fixed weight. In this case, the difference of the error function value's magnitude might lead to a biased calibration to the variable that has a larger impact on the error function.

However, our proposed new objective function DYNO dynamically normalizes the error function value of each variable using the maximum and minimum value of each variable's error function value obtained during the calibration and hence balances the impact of each variable on the objection function. Hence DYNO is designed to work well in scenarios where the error function values of each variable are significantly different due to differences in measurement uncertainty and the distribution of each variable's observations.

Another possible future work is the consideration of the spatial-temporal variability of calibration parameters (such as Secchi depth, Ozmidov length scale, Dalton number, and Stanton number). We considered them as constant parameters in our study to simplify the problem. This is reasonable since our study area is relatively small and there is not much seasonal variation. In cases where the study areas are large and there is significant seasonal variation, there might be a need to consider these parameters as space and time-varying calibration parameters. The consideration of space and time variability will, of course, increase the number of decision variables for the optimization problems, which will bring more challenges. In that case, new methods on how to reduce the parameter dimensions might be needed (e.g., designing some low dimensional controlling parameters, like curve number in hydrology (Bartlett et al., 2016), to represent the high dimensional space-time variability of these parameters).

**4 Conclusions**

We conclude that the DYNO objective function that we propose is a new effective way to balance the calibration to different variables (i.e., temperature and velocity) in optimization-based -calibration. It is possible that the magnitudes of goodness-of-fit measures for different variables are very different (which may fluctuate during the optimization search), and thus the optimization search cannot maintain the balance between different variables. Hence DYNO dynamically modifies the objective function, for multi-variable calibration, so that the error for each variable is dynamically normalized in each iteration. This is to ensure that the search is giving approximately equal weight to each variable (e.g., velocity and temperature).

The proposed DYNO is tested in this study for simultaneous temperature and velocity calibration of a lake model. Moreover, DYNO is integrated with the PODS algorithm for testing on expensive lake hydrodynamic model calibration in parallel. Results indicate that using DYNO ensures a balanced calibration between temperature and velocity. We provide a detailed analysis to illustrate that DYNO balances the weight between different objectives dynamically, and thus allows for a balanced parameter search during optimization.

We conclude that calibrating to the error of one variable (either temperature or velocity) cannot guarantee the goodness-of-fit of another variable in our case. Of course, the most accurate predictions can be obtained by having both temperature and velocity data. These comparisons are possible because we have, via synthetic simulation, the true solution for the lake model. Our analysis suggests that for practical applications, both temperature and velocity data might need to be considered for model calibration. The common practice of calibrating only to temperature data might not be sufficient to reproduce the flow dynamics accurately and extra effort and expense to collect velocity data is expected to give a beneficial effect. However, our analysis is based on synthetical data from models, hence it is worthwhile to further investigate the role of temperature and velocity in model calibration with real temperature and velocity measurements.

There are many possible future areas for the application of this method. DYNO would be effective for other multi-variable and multi-site calibration problems (especially for problems with many variables). Future research could apply the DYNO methods to other problems and use other optimization algorithms.

**Code and Data availability**

The tropical reservoir hydrodynamic numerical model and data were provided by PUB, Singapore's National Water Agency (https://www.pub.gov.sg/). The Delft3D open source code could be downloaded from https://oss.deltares.nl/web/delft3d/source-code. The PODS open source code could be download from https://github.com/louisXW/PODS. The code for objective function can be download from https://github.com/louisXW/DYNO-pods.

**Author contributions**

WX took responsibility for the methodology, software, formal analysis, investigation, original draft preparation, and visualization. WX, TA, CAS discussed the design and results and edited the manuscript.

**Competing interests**

The authors declare that they have no conflict of interest.

**Acknowledgments**

This research was supported by the National Research Foundation, Prime Minister's Office, Singapore under its Campus for Research Excellence and Technological Enterprise (CREATE) programme and from Professor Shoemaker's start-up grant from the National University of Singapore (NUS). The authors acknowledge PUB, Singapore's National Water Agency for providing the tropical reservoir hydrodynamic numerical model and data. The computational work for this article was entirely performed on resources of the National Supercomputing Centre, Singapore (https://www.nscc.sg).

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
