# Peer review of "A Novel Objective Function DYNO for Automatic Multi-variable Calibration of 3D Lake Models"

_Hydrology and Earth System Sciences, 2021_

## Referee Comment (RC1)

Comments to the manuscript submitted to HESS
A Novel Objective Function DYNO for Automatic Multivariable Calibration and Application to Assess Effects of Velocity versus Temperature Data for 3D Lake Models Calibration

Below few minor comments/suggestions to improve the paper. I strongly recommend the authors to revise the English language as I found some typos (missing spaces, wrong singular/plural verbs) and some passages to be improved, especially in the Abstract, Introduction and Methods. Figures are ok but some "transparency" boxes from the png (my guess) are visible so please consider improving the quality or changing the figure format.

Title: Please synthetize the title: my suggestion:
A Novel Objective Function DYNO for Automatic Multivariable Calibration of 3D Lake Models

L19-20: "by comparing the result of using DYNO to results of calibrating to either temperature or velocity observation only" please rephrase

L27-33: please revise the English form and make it less general. An example: "*Hydrodynamic models simulate the hydrodynamic and thermodynamic processes in lakes and reservoirs*": not really, "hydrodynamic models" is a very wide definition for models that can be used to simulate either hydrodynamics only or hydro-thermodynamics (as for Delft3D), and to different water environments, not only lakes. This is just a formal comment and applies to the entire Introduction, please avoid generalized and rough statements as well as repetitions.

L30: The authors say that hydrodynamic models simulate specific water quality variables. What do they mean with "specific water quality variables"?

L46: "some" water variables. I'd say all of them! If the model is 1D, all variables will be 1D. Also in this case, please be more precise: "spatial" is very general, 1D models typically consider the vertical dimension, so what they don't provide is the horizontal spatial distribution and in general they can't capture the 3D processes (e.g. circulations, 2D waves…).

L66-75: The authors could be interested in reading this work https://doi.org/10.1016/j.envsoft.2021.105017 and references therein where some of the issues mentioned in this paragraph are addressed in a manual calibration of Delft3D in a lake.

L85-90: a little confused, please revise English form.

L100: etc → among others; desire → require

L108: "*A key challenge for automatic calibration of multi-variable calibration problems is in defining a suitable objective function to calibrate multiple variables simultaneously*": please remove some "calibrations", e.g.: A key challenge for automatic calibration of multi-variable problems is in defining a suitable objective function.

L110: varying → vary

L118: Anticipate MOO to the first time it is mentioned (L113). Does SOO refer to the optimization methods mentioned in lines (105-112)? If yes, please anticipate SOO as well.

Tab.1 check spaces

L164: (e.g. calibrating temperature and…) we got that the authors are dealing with multi-variables problems and in particular with temp and vel. Please revise the paper critically and remove repetitions.

Sect.2.3: I see that the point of this work is not the simulation of one specific case study, but since the name of the section is "Study site and data" the authors could at least include the name of the lake and a few morphological characteristics (e.g. where it is located, how deep and large it is) and then refer to Xia et al. 2021 for all other details. Also what year was simulated should be reported for completeness (in the text and in fig. 1).

L189: just a curiosity, why are the names of the station A1, B1-4? Does this A1 station mean something different than the others?

L211-212: "the water utilities′ employees and consultants": some specification is missing here... maybe Singapore?

L234: Secchi depth: what about the space-time variability of this parameter? Delft3D allows to consider both variations, as transparency is not a constant and uniform feature of water. Did the authors consider this possibility?

L237: please consider moving here the sentence in lines 228-230.

L241: Model parameter(s) in table caption

Tab.2: Why didn't the authors include the coefficient of free convection, whose value greatly affects the modeled temperature? The default value in Delft3D is 0.14 but it strongly modifies the thermal profile when tuned.

L261: Have the authors considered normalization of RMSE by the standard deviation instead of the mean and why did they eventually chose the mean as normalization factor?

L272-273 and Eq.11: It looks to me that there is a power of 2 missing in eq.11. The referred papers (starting from Beletsky et al. 2006) present this formula with the module of the difference between observations and models (at the numerator) and the module of the observations vector (at the denominator) both to the power of 2, and then everything under the square root. If the authors prefer to use the Euclidean norm instead of the module it is fine with me, as $||x||_2 = |x|$ in $R^2$,

but the power of 2 should be maintained anyway, right? I checked the codes uploaded on github and I see a **2 in the computation of the Fourier norm, but please double check the equation, the code and the references.

L280: "being summed them" remove them

Sect.2.6.1. This paragraph seems like an advertisement of PODS and is highly technical. I'm not sure it is adequate to the wide audience of HESS. The authors could consider limiting the acronyms to those that are really needed (e.g. do we really need to know that "*DYCORS inherits the dynamic coordinate search idea from DDS (Tolson and Shoemaker, 2007) to improve its effectiveness and efficiency for high dimensional problems*"?) and try to clarify a bit. What does RBF surrogate mean? Maybe the authors could consider merging sect 2.6.1. and 2.6.2. and try to explain things in easier terms (and referring to publications for high-technical details), eventually splitting some very long sentences.

Figure2 and commenting text: two aspects are not completely clear to me:

1) how does the RBF surrogate model interact with the new fitness F and how does it then communicate to worker-i such that the latter can run a new hydrodynamic simulation? I believe that lines 318-320 are crucial here. The authors could consider expanding these two lines as it seems to me they are taking for granted too many things.

2) how are workers-p related one another? Are all steps M1-M6 performed independently on each processor?
If yes, how do they communicate at the end to define the best solution found?
If not, do M1-M6 steps consider all trials from workers-p within ψ? And how does M6 discriminate to which worker-i it should communicate the new X?

Table 4 and commenting text: a "good model" range for Fn should be between 0 (perfection) and 1. How do the authors comment such large values of Fn? Their best solution Cali-borh gives almost 2, while the Cali-Vel, which should minimize only Fn, gives almost 3!

L395: missing dot before Figure3?

L421: overestimated → overestimation

L433-434: Latin Hypercube Designs (LHD) is mentioned here for the first time. Please specify what it is used for in the methods section.

L456: "only the best solution in each of the optimization iterations are plotted" → is plotted

L462 please change the asterisk with standard math notation

Figure5 and 6 Please make the axis labels consistent between the two figs. fvel(X) and ftem(X) should be fine for fig. 5 (if I got correctly that only the best K is plotted), while ftem(X|K) and fvel(X|K) should be fine in fig. 6

Figure 6: please improve the readability of the number inside the darkest hexagon by either changing the darkest color or modifying the color of the number (e.g.) yellow. Please note that there is a missing C in (c) in the figure legend.

I would have expected the darkest hexagons to be the closest to the origin. Why isn't it like that? What is the hexagon containing the final solution? The authors could consider highlighting it e.g. with a bold colored contour line.

L485: represent (remove s)

L534: please correct, which one is better than the other? N2 better than N1?

L537: like→ likely?

L565: helps to improve the calibrate of → calibration

---

## Author Comment (AC1)

**Response to referee comment Referee #1**

We appreciate and would like to thank Referee #1 for taking the time and effort to read our manuscript and expressing the generally positive impression of our work. We will improve our manuscript based on reviewer's helpful comments. Our point-to-point response are below (comment of the referee in black, our response in purple).

**Point-to-point response**
* * *
**R1-1:** This paper presents a new objective function for automatic calibration of 3D hydrodynamic models based on water temperature and velocity data. The case study is a tropical lake, where the authors tested their DYNO+PODS for calibrating a Delft3D model against a subset of previously simulated 3D flow and temperature fields from a run assumed to be the "truth". I strongly appreciate this approach as it provides the full control of the optimization. Moreover, as the authors stressed many times (maybe too many, a lot of repetitions could be avoided), the use of velocity data together with water temperature is not so diffused yet in the field of lakes hydrodynamic modeling but it is crucial to have consistent results and realistic flow fields. Hence I welcomed the author's effort in quantifying how important it is.

Response: Thanks for reviewer's comments.

**R1-2:** Optimization algorithms as well as suitable objective functions for calibrating complex models are needed in the wide environment of hydrological numerical applications. I enjoyed the reading of the manuscript, which is well written and clearly structured. I appreciated how the authors describe their DYNO and tested its performances. Some clarifications are required, in my opinion, to make the optimization part (which is not the focus of the manuscript but still a key part of it) more accessible to the wide public of HESS, but in general I believe this work is worthy for publication on HESS after some minor revisions.

Response: Thanks for the reviewer's comments. We will revise the optimization part in the revised manuscript (based on this comment and comment at R1-29 and R1-30). Detailed response refers to the response to the comment R1-29 and R1-30.

**R1-3:** I'd like the authors to consider deepening the analysis on two more aspects which I believe worth a little discussion:

Computational costs: Addressing this aspect is mandatory in a paper on optimization algorithms. The authors make some general considerations here and there, but maybe a dedicated paragraph would be more appropriate. My questions: How many (real) runs of the hydrodynamic model were necessary to get the final solution for e.g. each trial/each configuration of Dyno (temp, vel, both)? What is the computational cost (wall clock time) of these tests? e.g., how much time compared to the error?

Response: Our study set the same evaluation budget (i.e., the maximum number of hydrodynamic model runs) for each trial and calibration scenario (i.e., Cali-Tem, Cali-Vel, and Cali-Both). The maximum number of hydrodynamic model runs in each trial is 192, which is about 8 iterations

with 24 evaluations in each iteration. The result indicates that 8-iterations is a sufficient budget as the calibration progress plot in Figure R1 shows the optimization experiments almost converged in the last few iterations.

The computational time of one simulation takes about 5 hours on a windows desktop with CPU Intel Core i7-4790. However, when running 24 simulations simultaneously on a multi-core platform, the computational time gets longer because of the limited cache memory resources (as discussed in Xia and Shoemaker (2022)). Cache memory is a small amount of much faster memory than main memory. The wall-clock time for one iteration with 24 cores simultaneously running is about 12 hours if using the default process scheduling of the nonuniform memory access (NUMA) multi-core system. We used the mixed affinity scheduling proposed by Xia and Shoemaker (2022), and the wall-clock time is reduced to about 8 hours per iteration. The mixed affinity scheduling changed the default affinity setting by setting a hard affinity on the simulation of each PDE model (i.e., fixing the process of each PDE simulation to one core). This approach proved to be efficient for memory usage and reduced the simulation time. More details about the mixed affinity scheduling and the NUMA system can be found in the study of Xia and Shoemaker (2022). Hence, the wall-clock time of each trial takes about 64 hours (8 iteration×8 hours/iteration).

*Xia, W., & Shoemaker, C. A. (2022). Improving the speed of global parallel optimization on PDE models with processor affinity scheduling. Computer-Aided Civil and Infrastructure Engineering, 37(3), 279-299.*

We will add a separate paragraph in the revised manuscript as suggested by reviewers to discuss the computational cost of the experiments and put the calibration progress plot (Figure R1) in the supplementary materials.

[Figure]

Figure R1. Calibration progress plot of PODS in Cali-Tem, Cali-Vel, and Cali-Both. The best solution found so far in average of three trials is plotted with the number of evaluations.

**R1-4:** Application to real data from observations: The authors auspicate that future users will test the DYNO against observations and so do I. So my questions are: Are there any constraints in the time/space frequency of the observations? In order to calibrate e.g. their Delft3D lake model to some temperature profiles and some current measurements, how should this data be? E.g. should temp and vel be simultaneous/in the same locations/same depths? As far as I understood, this is indeed the case of the data used in the authors' application, but this is not that common in standard monitoring schemes, where data are sparse and often not simultaneous. So basically, does this optimization (I guess this applies to PODS rather than DYNO) handle sparse observation? Did the authors test their DYNO+PODS by changing e.g. the sampling time or the number of locations of their "truth"? I guess the more data the better, but I'd greatly appreciate some discussion on this. Is there an optimal number of locations/time frequency which gives a satisfactory calibration?

Response: The reviewer is right that we used the temperature and velocity data at the same locations and depths. We are not sure what the result would be if the temperature and velocity data were collected at different places. In reality, we think it is possible that people do measurements of temperature and velocity at the same locations (or locations sufficiently close to each other). We agree that people might measure temperature and velocity at different temporal frequencies (for example, the observation data of one variable might be sparser than that of another). In general, there is no problem in using PODS on problems with sparse observation in the technical aspect. PODS can handle sparse observation. And we don't require the observation of temperature or velocity to be at the same location or with the same frequency since the error of temperature and velocity are calculated separately.

DYNO also does not require the same number of locations / same time frequency for different calibrations constituents. Regardless of number of locations / sampling frequencies, DYNO normalizes the objective function of each constituent dynamically, to allow equal weight to each constituent in the calibration process. In the calibration setup of this study, we believe DYNO has an advantage since it can dynamically adjust the weights between the error of temperature and velocity. In cases when the locations and time-frequencies are vastly different for the two variables, it may be reasonable to introduce custom weights to give more weightage to one constituent. For instance, if velocity observations are limited in both space and time, a custom weight could be introduced in DYNO to reduce overall weightage of velocity in calibration.

We don't think there is an answer to the optimal number of locations and time-frequency. The frequency needs to be small enough to capture the time variation (e.g., diurnal variation or seasonal variation). The number of locations seems dependent on the geography or the reservoir's shape. The location should be enough to capture the spatial variations. The number of locations might also depend on the budget available to do these measures in real practice.

We will add the above discussion into the revised manuscript.

**Attached are few minor comments/suggestions to improve the paper.**

**R1-5:** Below few minor comments/suggestions to improve the paper. I strongly recommend the authors to revise the English language as I found some typos (missing spaces, wrong singular/plural verbs) and some passages to be improved, especially in the Abstract, Introduction and Methods. Figures are ok but some "transparency" boxes from the png (my guess) are visible so please consider improving the quality or changing the figure format.

Response: Thanks for the reviewer's comments. We will revise the English language based on reviewer's detailed comments below. We will do a careful proofreading for the revised manuscript. We will also improve the quality of the figures by removing these "transparency" boxes.

**R1-6:** Title: Please synthetize the title: my suggestion: A Novel Objective Function DYNO for Automatic Multivariable Calibration of 3D Lake Models

Response: Thanks for the reviewer's suggestion. We will change the title to be: "A Novel Objective Function DYNO for Automatic Multivariable Calibration of 3D Lake Models."

**R1-7:** L19-20: "by comparing the result of using DYNO to results of calibrating to either temperature or velocity observation only" please rephrase

Response: We will revise the sentence to be: "by comparing the calibration result obtained with DYNO to the result obtained through calibrating to only one variable (i.e., temperature or velocity)"

**R1-8:** L27-33: please revise the English form and make it less general. An example: "Hydrodynamic models simulate the hydrodynamic and thermodynamic processes in lakes and reservoirs": not really, "hydrodynamic models" is a very wide definition for models that can be used to simulate either hydrodynamics only or hydro-thermodynamics (as for Delft3D), and to different water environments, not only lakes. This is just a formal comment and applies to the entire Introduction, please avoid generalized and rough statements as well as repetitions.

Response: We will revise the sentence to be "Lake hydrodynamic models simulate the hydrodynamic or thermodynamic processes in lakes and reservoirs"

**R1-9:** L30: The authors say that hydrodynamic models simulate specific water quality variables. What do they mean with "specific water quality variables"?

Response: We want to say hydrodynamic models are often built to support water quality modelling of variables such as nutrients, toxins. We will rephrase the sentence to be: "These simulation models (e.g., hydrodynamic modelling) play a critical role in managing water bodies (e.g., rivers, lakes, and coastal areas), as they are built to support the simulation of the spatial and temporal distributions of specific water quality variables (e.g., nutrients, toxins), and to study the response of a water body to different future management scenarios."

**R1-10:** L46: "some" water variables. I'd say all of them! If the model is 1D, all variables will be 1D. Also in this case, please be more precise: "spatial" is very general, 1D models typically consider the vertical dimension, so what they don't provide is the horizontal spatial distribution and in general they can't capture the 3D processes (e.g. circulations, 2D waves…).

Response: We will revise the sentence to be: "However, one-dimensional models are unable to simulate the horizontal spatial distribution and cannot capture the 3D processes, and thus may not be suitable for certain studies."

**R1-11:** L66-75: The authors could be interested in reading this work https://doi.org/10.1016/j.envsoft.2021.105017 and references therein where some of the issues mentioned in this paragraph are addressed in a manual calibration of Delft3D in a lake.

Response: Thanks for reviewer's recommendation. The study mentioned by the reviewer is interesting and relevant. The study used different sources of temperature data (from in situ observations, multi-site high-resolution profiles and remote sensing data) to compensate for unavailability/scarcity of velocity measurements. This is a practicable approach when there is no velocity data available and there are such different sources of temperature data available. In cases, there is no high-quality remote sensing data (for example because of cloud) or a large amount of high-resolution profiles of temperature measurement it is still challenging to verify the spatial simulation of hydrodynamic quantities. We will discuss this study in our revised manuscript.

**R1-12:** L85-90: a little confused, please revise English form.

Response: We will revise the sentence to be: "Hydrodynamics models predict the velocities throughout the water body. These results are important to understand the spatial distribution of water quality problems in sizeable lakes. For the purposes of model calibration it is useful to know whether efforts to measure velocity directly are justifiable if temperature data is already available. We will examine the extent to which direct measurement of velocities justify the extra effort by giving more accurate results for hydrodynamics models. We will also look at the error of the spatial distribution of hydrodynamics associated with calibrating to temperature only, which is rarely studied in literature."

**R1-13:** L100: etc -> among others; desire -> require

Response: We will replace "etc" with "among others" and replace "desire" with "require".

**R1-14:** L108: "A key challenge for automatic calibration of multi-variable calibration problems is in defining a suitable objective function to calibrate multiple variables simultaneously": please remove some "calibrations", e.g.: A key challenge for automatic calibration of multi-variable problems is in defining a suitable objective function.

Response: We will revise the sentence to be: "A key challenge for automatic calibration of multi-variable calibration problems is in defining a suitable objective function"

**R1-15:** L110: varying ◊ vary

Response: We will replace "varying" with "vary".

**R1-16:** L118: Anticipate MOO to the first time it is mentioned (L113). Does SOO refer to the optimization methods mentioned in lines (105-112)? If yes, please anticipate SOO as well.

Response: We will anticipate MOO to the first time it is mentioned. We will also anticipate SOO at line 109-112: "Traditional approaches typically formulate the goodness-of-fit of multiple variables into a single objective function by adding weights between the goodness-of-fit of multiple variables and solve the problem with single objective optimization (SOO) techniques"

**R1-17:** Tab.1 check spaces

Response: We will correct the spaces in Table 1.

**R1-18:** L164: (e.g. calibrating temperature and…) we got that the authors are dealing with multi-variables problems and in particular with temp and vel. Please revise the paper critically and remove repetitions.

Response: Thanks for reviewer's comments. We will revise the paper critically and remove repetitions.

**R1-19:** Sect.2.3: I see that the point of this work is not the simulation of one specific case study, but since the name of the section is "Study site and data" the authors could at least include the name of the lake and a few morphological characteristics (e.g. where it is located, how deep and large it is) and then refer to Xia et al. 2021 for all other details. Also what year was simulated should be reported for completeness (in the text and in fig. 1).

Response: We agree with the reviewer that it would be better to provide the name of the lake. However, we are constrained by the local agency, which provided us with data, from releasing the lake name, including other confidential information such as reservoir locations and inflow and outflows. We appreciate if the reviewer understands our situation. But we are permitted to mention the depth and the surface of the reservoir. Hence we will add these information in the revised manuscript.

**R1-20:** L189: just a curiosity, why are the names of the station A1, B1-4? Does this A1 station mean something different than the others?

Response: Station A1 is the station where the real measured temperature data was used for the calibration of the hydrodynamic model. While there is no real observation data at other stations.

**R1-21:** L211-212: "the water utilities' employees and consultants": some specification is missing here... maybe Singapore?

Response: As responded to the comment above, we are not allowed to disclose the confidential information such as reservoir locations. So we did not mention the name of the local agency.

**R1-22:** L234: Secchi depth: what about the space-time variability of this parameter? Delft3D allows to consider both variations, as transparency is not a constant and uniform feature of water. Did the authors consider this possibility?

Response: The reviewer is right that Delft3D allows considering space-time variability of these parameters. We consider time-varying parameters such as Secchi depth, Ozmidov length scale, Dalton number, and Stanton number to be constant mainly because it would be very challenging to calibrate them as time-varying variables. Considering these parameters as time or space-varying parameters will substantially increase the number of decision variables in the optimization. In addition, the reservoir we study is relatively small (with maximum depths of 22 m and a surface of 250 hectares), and it is located in a tropical region where there is no significant seasonal variation. Hence we think it is acceptable to consider them as constant. But we think that considering these parameters' space-time variability into optimization calibration would be an interesting future topic. In that case, new methods for reducing the parameter dimensions are needed (e.g., design some low dimensional controlling parameters to represent the high dimensional space-time variability of these parameters). We will add these discussions at the end of the manuscript.

**R1-23:** L237: please consider moving here the sentence in lines 228-230.

Response: We will move the sentence in lines 228-230 to Line 237 and revise it to be: "The last parameter is the manning coefficient, which affects the roughness of the bottom of the lake and a direct impact on velocity."

**R1-24:** L241: Model parameter(s) in table caption

Response: We will replace "Model parameter" with "Model parameters".

**R1-25:** Tab.2: Why didn't the authors include the coefficient of free convection, whose value greatly affects the modeled temperature? The default value in Delft3D is 0.14 but it strongly modifies the thermal profile when tuned.

Response: We chose the calibration parameter set as suggested by the local experts on lake modeling (for the region of the study site) and they did not suggest the coefficient of free convection as an optimization parameter. As the reviewer pointed out, this parameter affects the thermal profile, and could be considered as a parameter to be optimized in future.

**R1-26:** L261: Have the authors considered normalization of RMSE by the standard deviation instead of the mean and why did they eventually chose the mean as normalization factor?

Response: We did not considered normalization of RMSE by standard deviation. RMSE could be normalized with standard deviation or mean. We just chose one of them (in our case we use the mean).

**R1-27:** L272-273 and Eq.11: It looks to me that there is a power of 2 missing in eq.11. The referred papers (starting from Beletsky et al. 2006) present this formula with the module of the difference between observations and models (at the numerator) and the module of the observations vector (at the denominator) both to the power of 2, and then everything under the square root. If the authors prefer to use the Euclidean norm instead of the module it is fine with me, as $\|x\|\_2 = |x|$ in R^2, but the power of 2 should be maintained anyway, right? I checked the codes uploaded on github and I see a **2 in the computation of the Fourier norm, but please double check the equation, the code and the references.

Response: The reviewer is right that there is a power of 2 missing in eq.11. We will revise eq.11 in the revised manuscript.

**R1-28:** L280: "being summed them" remove them

Response: We will remove them.

**R1-29:** Sect.2.6.1. This paragraph seems like an advertisement of PODS and is highly technical. I'm not sure it is adequate to the wide audience of HESS. The authors could consider limiting the acronyms to those that are really needed (e.g. do we really need to know that "DYCORS inherits the dynamic coordinate search idea from DDS (Tolson and Shoemaker, 2007) to improve its effectiveness and efficiency for high dimensional problems"?) and try to clarify a bit. What does RBF surrogate mean? Maybe the authors could consider merging sect 2.6.1. and 2.6.2. and try to explain things in easier terms (and referring to publications for high-technical details), eventually splitting some very long sentences.

Response: Thanks for reviewer's comments. We will revise the section 2.6 by merging section 2.6.1 and 2.6.2 as suggested by the reviewer. We will rewrite the text by explaining things in easier

terms and add more details than we previous did. The changes including the changes including more details about RBF surrogates and how it is used in the optimization.

**R1-30:** Figure2 and commenting text: two aspects are not completely clear to me: 1) how does the RBF surrogate model interact with the new fitness F and how does it then communicate to workeri such that the latter can run a new hydrodynamic simulation? I believe that lines 318-320 are crucial here. The authors could consider expanding these two lines as it seems to me they are taking for granted too many things. 2) how are workers-p related one another? Are all steps M1-M6 performed independently on each processor? If yes, how do they communicate at the end to define the best solution found? If not, do M1-M6 steps consider all trials from workers-p within ψ? And how does M6 discriminate to which worker-i it should communicate the new X?

Response: After one iteration is finished (e.g., the simulation of the $P$ hydrodynamic simulations). The objective function value $F(X|K)$ of all evaluations in ψ (including the newly finished evaluations in this iteration and previous iterations) are recalculated. The RBF surrogate model is rebuilt with the new objective function value $F(X|K)$ that are calculated based on Eq. (3). The newly built RBF surrogate model is then used for selecting the evaluation points for the next iteration. In PODS, the algorithm first generates a large number of candidate points around the best evaluation found so far. Then $P$ evaluation points are selected from the candidate points using a Surrogate-Distance Metrics discussed in the PODS paper. The metrics consider the approximated objective function value based on the surrogate model and the distance of the evaluation points from the evaluated points. We will revise lines 318-320 by adding more explanations.

To reviewer's question 2), the tasks of $P$ workers are independent of each other (i.e., each worker evaluates one hydrodynamic model). The steps M1-M6 are master's tasks, performed after the tasks of $P$ workers are finished in each iteration. So each worker sends back to the master the results of one simulation (i.e., temperature and velocity error of one evaluation). Then the master adds all these newly obtained results to the history list ψ, which contains the results of all evaluated points in previous iterations. M1-M7 only need to be performed on one processor. The jobs of M1-M7 is to generate P evaluation points and then distribute the P evaluation points to P processors for the workers' tasks. The master M7 distributes the P evaluation points randomly to the P processors so each processor will get an evaluation point and send back the evaluated results to master. We will modify the Figure 2 as below to make it clear.

[Figure]

**Figure R2 (Revised from Figure 2 in original manuscript).** Diagram of the implementation of DYNO with the parallel algorithm PODS. *P* is the number of processors available. The green texts (i.e., steps W3, M1-5) are changes made on PODS to incorporate DYNO. The rest part follows the original PODS method.

**R1-31:** Table 4 and commenting text: a "good model" range for Fn should be between 0 (perfection) and 1. How do the authors comment such large values of Fn? Their best solution Cali-borh gives almost 2, while the Cali-Vel, which should minimize only Fn, gives almost 3!

Response: The value in Table 4 is the sum of Fn value at multiple stations (in total 12 locations). So the Fn value at one location in average is about 0.167 for the best solution in Cali-Both scenario and 0.25 for best solution in Cali-Vel scenario. Hence it is not a large value of Fn. We will add these explanations in the manuscript.

**R1-32:** L395: missing dot before Figure3?

Response: We will add the missing dot.

**R1-33:** L421: overestimated ◊ overestimation

Response: We will replace "overestimated" with "overestimation".

**R1-34:** L433-434: Latin Hypercube Designs (LHD) is mentioned here for the first time. Please specify what it is used for in the methods section.

Response: We will introduce the Latin Hypercube Designs (LHD) in the method section (i.e., section 2.6.2).

**R1-35:** L456: "only the best solution in each of the optimization iterations are plotted" ◊ is plotted

Response: We will replace "are" with "is".

**R1-36:** L462 please change the asterisk with standard math notation

Response: We will change "$X_i, i = 1, ..., 3 * N_{max}$" to "$X_i, i = 1, ..., 3 \times N_{max}$"

**R1-37:** Figure5 and 6 Please make the axis labels consistent between the two figs. fvel(X) and ftem(X) should be fine for fig. 5 (if I got correctly that only the best K is plotted), while ftem(X|K) and fvel(X|K) should be fine in fig. 6

Response: Thanks for reviewer's suggestion. We will revise the Figure 5 as the reviewer suggested.

**R1-38:** Figure 6: please improve the readability of the number inside the darkest hexagon by either changing the darkest color or modifying the color of the number (e.g.) yellow. Please note that there is a missing C in (c) in the figure legend.

Response: Thanks for reviewer's comment. We will modify the color of the number inside the darkest hexagon and also add the missing "C" in the figure legend.

**R1-39:** I would have expected the darkest hexagons to be the closest to the origin. Why isn't it like that? What is the hexagon containing the final solution? The authors could consider highlighting it e.g. with a bold colored contour line.

Response: We would like clarify here that Figure 6 represents the joint distribution of the i) Velocity Error and the ii) Temperature error components of DYNO for "all simulations evaluated" during the different calibration scenarios. Hence, Figure 6 is a representation of the optimization search dynamics for the three different objectives analyzed (i.e., Cali-Tem, Cali-Vel and Cali-Both). Figure 6 shows that when calibrating to only temperature or velocity, the search of the optimization only considered the error of one variable and ignored the error of another variable. Hence the darkest hexagon (i.e., the concentration of error distribution) is expected to be close to one of the coordinate axes instead of the origin (e.g., in Figure 6(a) more solutions are found with only better Temperature error, hence darker hexagons are close to vertical axis). When calibrating to both temperature and velocity, the darkest hexagons are close to the origin but not necessarily the closest. This could happen for many reasons. For example, the solution space is multi-modal, and many solutions have the same error around the value of the darkest hexagons. It could also happen that the algorithm searched more around the region that is not close to the best solution. In general, we think it is true that the best solution is the closest to the origin when calibrating to both temperature and velocity, but we don't think it is true that the darkest hexagons must be the closest to the origin.

**R1-40:** L485: represent (remove s)

Response: We will remove "s".

**R1-41:** L534: please correct, which one is better than the other? N2 better than N1?

Response: We will correct the sentence to be "…with DYNO-N2 might be better than with DYNO-N1"

**R1-42:** L537: like◊ likely?

Response: We will revise the sentence to be: "… also likely happen for…"

**R1-43:** L565: helps to improve the calibrate of ◊ calibration

Response: We will replace "calibrate" with "calibration".

---

## Author Comment (AC3)

**Response to referee comment Referee #3**

We appreciate and would like to thank Referee #3 for taking the time and effort to read our manuscript and expressing the generally positive impression of our work. We will improve our manuscript based on reviewer's helpful comments. Our point-to-point response are below (comment of the referee in black, our response in purple).

**Point-to-point response**

**General evaluation and major comments**

**R3-1:** Overall, I think the manuscript is well structured and the methodology is well prepared. The presented tables and figures support the findings that were presented in the text and provide good insights in the functionality of the presented algorithm. While I think the overall quality of the manuscript in its present form is already good, I see some crucial points that require clarification. Other less significant points should also be improved to improve the quality and readability of the manuscript. I will outline my major concerns in the following and will address smaller issues in a line-by-line notation in the following section.

Response: Thanks for reviewer's comment. We will revise the manuscript based on the below comments raised by reviewer.

1. *Synthetic study design:*

**R3-2:** The observation data were generated with the same model structure that was later on investigated in the case study. Which means that "observed" variables (in this case velocity and temperature) were calculated with the exact same set of equations that in the following calculated the simulated variables. I think this property of the observation data could potentially impair the entire analysis and favour the simultaneous calibration of velocity and temperature.

As the authors outline in section 3.5 of the manuscript, multi-variable calibration problems are often affected by trade-offs between the variables for which performance metrics should be optimized (minimized). One reason for that is that models are simplification of the represented reality and parameters can affect multiple processes, sometimes in the opposite directions. Thus a change of a parameter value into one direction could improve the performance of one metric, while deteriorating the performance of another metric at the same time. This is different in the synthetic example where in fact the reality and the model are the same thing. Thus the "observed" time series of velocity and temperature perfectly agree with the model simplifications and assumptions.

Thus, in such a case providing both, temperature and velocity, to the search algorithm should better constrain the parameter response surface than only providing one of the two variables. If this hypothesis is sound, then the presented results would by affected by this effect. In this case only a real case scenario could provide an honest comparison of the three cases.

Response: The reviewer questioned the rationality of using data generated from the model as observations. We fully agree that using real observation data for such investigation is better. However, real observations for velocity are not available for this study, and are usually difficult to

find. Due to scarce availability of observed velocity data, the use of both temperature and velocity for calibration of 3D lake hydrodynamic models is also rare both in practice and research. However, understanding of the importance of velocity data in calibrating such models is important and could be possible by generating synthetic observed data from lake hydrodynamic model simulations. These underlying hydrodynamic models are physically-based and precise. Hence, we believe that generating synthetic observation data from such models is reasonable. We agree with the reviewer that generating synthetic data from conceptual environmental simulation models may not be reasonable though (e.g., conceptual hydrologic models that have intrinsic structural errors and uncertain conceptual parameters). Of course, even in the case of 3D lake hydrodynamic models, synthetic model scenarios cannot be considered the same as real-world situations, but they are close representations of real-world situations. We think that our analysis on model-based synthetically generated observed data can provide at least some implications for the investigation people could do with real-world data. Our research implication of the use of velocity observation is also in line with the study of Baracchini et al. (2020), where they also suggest having both temperature and current velocity for complete system calibration.

*Baracchini, T., Hummel, S., Verlaan, M., Cimatoribus, A., Wüest, A., & Bouffard, D. (2020). An automated calibration framework and open source tools for 3D lake hydrodynamic models. Environmental Modelling & Software, 134, 104787.*

The reviewer also highlighted the discussion of trade-offs in section 3.5. We need to clarify that section 3.5 discusses other possible applications for which our new method, DYNO, could be used. But we don't think the multi-variable hydrodynamic calibration problem we are dealing with is a trade-off problem. As we have discussed in the introduction (lines 113-122 of the original manuscript), it is not apparent that there is usually a trade-off between the fit of multiple variables / constituents. This might be true for conceptual models, e.g., hydrologic models, where equations are experimental, and there is lots of uncertainty embedded in parameterization. In such models, trade-offs typically exist between different sub-objectives (e.g., using different performance metrics). But hydrodynamic models are physically based. Moreover, lake water velocity is important for determining distribution of water temperature. Hence, we believe that simulating velocity more accurately in hydrodynamic models, is expected to improve accuracy of water temperature simulation and distribution. Consequently, a trade-off may not exist between the fits of temperature and velocity. We agree that investigation on real data might be more convincing in ensuring the above hypothesis as well, and this could be part a future study where observed data for both temperature and velocity is available. We will revise our statements in the manuscript to be more cautious about implications for real-world practice based on modeling results.

*2. Number of iterations of the search algorithm*

**R3-3:** This comment is somehow related to the previous one. In theory it should be possible for the search algorithm to identify the "true" parameter set that was used to generate the synthetic observation data. This global minimum is present on the parameter response surfaces of all three calibration scenarios and has a value of 0 independent of the used metric (DYNO or the single performance metrics). Thus, when not ending up in a local minimum all three cases should converge towards this global minimum. As outlined above, I simply think that the Cal-Both scenario does this quicker due to the given reasons.

In the presented results all "best" solutions did not find the global minimum. As far as I got it right from the text, each experiment involved 8 iterations with 24 parallel evaluations in each iteration step. Given a computation time of 5 hours per simulation run (according to the text and thus 5 hours per parallel iteration) one experiment takes 1.6 days. I am wondering if experiments with larger numbers of iterations were performed (that are maybe just not shown). I would be interested how the convergence of the three calibration experiments develops with larger number of iterations.

Response: The reviewer brings up an interesting aspect to discuss. In theory, the global minimum has a value of 0 for all three calibration scenarios. However, it might not be correct that having zero temperature error at these observation locations would mean zero velocity error at these observation locations. This is because only the observation at part of the simulation space is used for calibration (not the observation data at each grid and each time step of the simulation space are used to calculate the temperature error). So the temperature error may be 0 at these observation locations, while the temperature error is not 0 at other locations where observation is not used in calibration. In this case, getting a temperature error 0 at observation locations cannot guarantee the velocity error is 0. Hence it is possible that the optimization ends up in a local minimum and does not converge towards the global minimum (both temperature and velocity error are zero) in cases calibrating to only temperature or velocity. In optimization search, the algorithm might not even reach the solution with zero temperature or velocity error when calibrating to only one variable. This is because the optimization only focused on one variable (temperature or velocity) and ignored the calibration on another. For example, if only calibrated to temperature, the velocity error might still be large. In this case, the algorithm is less likely to find the solution with temperature error 0 while the velocity error is large. This is exactly what Figure 5 (a) in the original manuscript showed (We copied Figure 5 (a) below). In Figure 5 (a), when calibrating to temperature only, the best solution found after each iteration is converged at places where the velocity error is large, and temperature error is not reduced to 0. In contrast, the optimization search considered both temperature and velocity error when calibrating to temperature and velocity. Hence the best solution found after each iteration is improved in both temperature error and velocity error.

The reviewer asks for experiments with a large number of iterations. We did not increase the number of iterations because, 1) we found the optimization search almost converged at the last few iterations (As shown in the calibration progress plot Figure R1 below) and 2) our problem is computationally very expensive; one single simulation takes about 5 hours; 24 simultaneous simulations take about 8-12 hours because of limited memory resources. It takes more than 64 hours to conduct one calibration experiment, and we have three scenarios, and each scenario is repeated in three trials. This is a huge amount of computing. Meanwhile, in real practice, users are also unwilling to wait too long for the calibration process. That is why we compared the accuracy of the solutions within a limited computing budget.

In the revised manuscript, we will add the above discussion into the manuscript, which we think is interesting.

[Figure]

Figure 5 (Copied from the original manuscript). Calibration progress plot of the best solution found (in term of objective function value) during optimization search by PODS when calibrating to temperature only (Cali-Tem), calibrating to velocity only (Cali-Vel), and calibrating to both temperature and velocity (Cali-Both). Three random trials (i.e., T1, T2, and T3) are plotted in (a), (b), and (c). Lower velocity and temperature error are better. The yellow makers are evaluation point in initial experiment design using Latin Hypercube Design (LHD). Besides solutions in LHD, only the best solution in each of the optimization iterations are plotted (i.e., makers lined with lines). The line links the best previous solution in one iteration to the best solution in next iteration. The arrow indicates the direction from the previous solution to the next solution.

[Figure]

Figure R1. Calibration progress plot of PODS in Cali-Tem, Cali-Vel, and Cali-Both. The best solution found so far in average of three trials is plotted with the number of evaluations.

*3. Consistency in the nomenclature*

**R3-4:** Overall I think the outline of DYNO and the explanation of variables is well done. Yet, I found some inconsistencies in the nomenclature and some mathematical definitions that must be improved. I will address these (at least the ones that I found) in the line-by-line comments.

Response: Thanks for the reviewer's comments. We will carefully revise the nomenclature and mathematical definitions. We also have a detailed response to the line-by-line comment below.

*4. English language*

**R3-5:** Overall I think the manuscript is well written. In some sections of the manuscript I found that the same errors repeat in every second line (e.g. singular/plural, missing articles). Although I am not a native speaker myself I had the impression that the manuscript could require some proof reading. Some sentences I had to read over and over again, but they still do not make sense to me. I addressed them as well in the line-by-line comments.

Response: Thanks for the reviewer's comments. We have carefully looked at the reviewer's comments about these language issues. We will revise them as the reviewer suggested in the revised manuscript.

*5. Non-color blind color theme in figures*

**R3-6:** Although this does not affect me, I would advise to change the color theme in the figures and refrain to use red and green at the same time (8% of males are affected by red-green color blindness).

Response: Thanks for the reviewer's comments. We agree with the reviewer's suggestion, which is very important. We will change the red and green to red and blue colors for all related figures.

**Line-by-line comments**

**p.1 L13, p.4 L123 and further**: Please use a consistent naming for Dynamically Normalized Objective function. Either all first letters caps or none. It differs throughout the text. Also only use the acronym after it was first introduced in the text.

Response: We will revise the manuscript with a consistent naming for Dynamically Normalized Objective Function and used acronym after the first introduction in the text.

**p.1 L21 - 22:** Please rephrase this sentence. Further, I think the statement is not universally true how it is formulated, that calibrating only one variable does not improve the calibration of another variable.

Response: Our statement is not to say that calibrating one variable does not "improve" the calibration of another. We say calibrating one variable cannot "guarantee" the fit of another. Our results support our statement. When calibrating to temperature only, the velocity error is large, and vice versa. It is true in our case. Of course, it might not be universally true for other cases. We will rephrase the sentence to be: "The result indicates that DYNO can balance the calibration in terms of water temperature and velocity and that calibrating to only one variable (e.g., temperature or velocity) cannot guarantee the goodness-of-fit of another variable (e.g., velocity or temperature) *in our case*."

**Eq. 2, Table 1:** Consistent notation of *Sim* and *Obs* necessary. Either capital first letter or lower case.

Response: We will capitalize the notation of Sim and Obs in Table 1 and throughout the manuscript.

**Table 1**: Please rephrase the sentence: "The set of variables the observation data of which is used in calibration"

Response: We will rephrase the sentence to be: "The set of variables whose observation data is used in calibration."

**Table 1:** I would suggest to add the vector $\{x_1, ..., x_d\}$ to the definition of $X$ as e.g. Table 2 refers to them.

Response: We will add following text into the definition of X in Table 1: "$X = (x_1, x_2, \ldots, x_d)$."

**Table1 and throughout the text:** I would suggest to use the nomenclature $Sim^k_{j,t}$ (same for *Obs*) in the definition and throughout the text to indicate the time dependency. This nomenclature is in fact introduced later in the manuscript. Also the definition refers to the time *t* which is not indicated in the variable.

Response: We will add the definition of $Sim^k_{t,j}(X)$ and $Obs^k_{t,j}$ in Table 1. We will bold the definition of $\boldsymbol{Sim}^k_j(X)$ and $\boldsymbol{Obs}^k_j$ to denote the vector of simulation output and observations data at all time steps ($\boldsymbol{Sim}^k_j(X) = \left(Sim^k_{1,j}(X), \ldots, Sim^k_{N,j}(X)\right)$ and $\boldsymbol{Obs}^k_j = (Obs^k_{1,j}, \ldots, Obs^k_{N,j})$). We keep the definition of $\boldsymbol{Sim}^k_j(X)$ and $\boldsymbol{Obs}^k_j$ for the simplicity of Eq. (2) (since the goodness of fit function takes the simulation output and observation vector as input).

**p.5 L153**: ...multiple variables *with* a single objective function.

Response: We will revise the sentence to be: "… formulate the error of multiple variables with a single objective function".

**p.5 L159:** Please remove \textit{value} as the distribution of NRMSE is not a single value.

Response: We are trying to make two comparisons: one is the highest attainable NRMSE value another is the distribution of the NRMSE value. We will revise the sentence to be: "However, it is still possible that the highest attainable value (or distribution) of NRMSE (or KGE) across the parameter space for one variable maybe be much higher than the highest attainable value (or distribution) of NRMSE (or KGE) of another variable.

**p.5 L160:** Same as above.

Response: We have resolved this comment in response to the comment above.

**p.5 L160:** Hence **,** ...

Response: We will revise the sentence to be: "Hence, … ."

**p.5 L163:** Again consistent naming of DYNO or removing the full name as the acronym was already introduced.

Response: We will remove the full name of DYNO.

**p.5 L166:** ...all evaluations in $\psi$ found so far?

Response: $\psi$ is the set of evaluations found so far by the optimization. We will revise the sentence to be: "…. Let $\psi$ be the set of evaluations found so far by the optimization, DYNO (as shown in Eq. (3)) normalizes the error of each variable $f_k(X)$ with its upper and lower bound, $f_k^{max}$ and $f_k^{min}$ of all evaluations in $\psi$"

**p.5 L166:** The variable $\psi$ was not introduced at this point and is I think not defined in the manuscript.

Response: We will add the definition of $\psi$ in response to the comment above.

**p.5 L167:** *The* mathematical formulation...

Response: We will revise the sentence to be: "The mathematical formulation ..."

**p.6 L172**: ...all evaluation*s* ...

Response: We will revise the sentence to be: "… all evaluations …"

**p.6 L172 - 174:** Please rephrase this sentence. It is in my opinion not clear what is meant.

Response: We will break the sentence to two short sentence: "where $f_k^{max}(X)$ and $f_k^{min}(X)$ are the maximum and minimum values of $f_k(X)$ for all evaluations in  .  $f_k^{max}(X)$ and $f_k^{min}(X)$ have to be updated dynamically in each iteration during optimization."

**p.6 L180 - 185:** The procedure as described is confusing and does not really make sense to me. A calibrated model exists from a previous study. This calibrated model was taken, but different parameters based on "expert guessing" were then used to run the model. So it is not the calibrated model anymore?

Response: We will revise the words "expert guessing". The set of "true" value we use is the manual calibration results from expert. The words "expert guessing" might cause confusion. We will replace it with "manual calibration by experts".

**p.6 L186:** Replace *a* by *the* as you did not use a parameter set but the parameter set $X^R$.

Response: We will revise the sentence to be: "the vector of model parameters $X^R$"

**p.6 L193:** ...the true value*s* of the parameter vector $X^R$ *are*...

Response: We will revise the sentence to be: "...the true value*s* of the parameter vector $X^R$ *are*...".

**p.6 L194**: I would replace *value* with *set*.

Response: We will replace value with set.

**p.6 L202 - 203**: This sentence is not clear to me.

Response: We will revise the sentence to be: "We have the five sampling locations across the reservoir. The observations data at these five locations are used to calibrate the model parameters."

**p.7 L226**: coefficient*s*

Response: We will change coefficients to coefficient: "The vertical exchange of horizontal momentum and mass is affected by vertical eddy viscosity and eddy diffusivity coefficient."

**p.8 L229**: *the* Manning formulation

Response: We will add "the".

**p.8 L229:** ...which is a parameter tha should also be calibrated.

Response: We will revise the sentence to be: ", which is a parameter that should also be calibrated."

**p.8 L232:** ...*are* parameterized...

Response: We will revise the sentence to be: "...*are* parameterized..."

**p.8 L237:** *M*anning coefficient

Response: We will capitalize the word "Manning".

**Section 2.5:** This section is actually a repetition of L146 – 150

Response: We will remove the repeated information. Section 2.5 gives the detailed calibration formulations. We will revise the first paragraph of section 2.5 to be: "Three scenarios are considered to investigate the impact of model calibration against temperature and/or velocity observations (as discussed in section 2.1). The first two scenarios calibrate to only one variable, and the last scenario calibrates both variables simultaneously. This section give the detailed calibration formulations of these three scenarios."

**p.9 L280:** Remove *them*

Response: We will remove "them".

**p.9 L287:** I would suggest to replace *a roughly* by *an approximately*

Response: We will replace "a roughly" by "an approximately".

**Caption Table 3:** formulation*s*

Response: We will change "formulation" to "formulations".

**p.10 L309:** Use the acronym DYNO instead

Response: We will replace "the Dynamically Normalized Objective Function" with "DYNO".

**p.10 L317:** Replace *is* with *are*

Response: We will replace "is" with "are".

**Figure 2**: The positions of the labels *Yes* and *No* in the stopping criteria are not clear to me.

Response: We will correct the position of "Yes" and "No" in Figure 2.

**p.12 L359 - 366:** The argumentation in this paragraph could be affected by the affect of synthetic observation data as outlined in the first major remark.

Response: We have responded the first major remark in response to R3-1. The discussion in line 359-366 is stating the results based on physical models. These hydrodynamic models are built based on physics and knowledge human learned in the past hundreds of years. Hydrodynamic models are not like most of hydrology models that with lots of uncertainty. They of course cannot be try the same as real world situation but they are close representations of the real world situation. We think the investigation on models can provide at least some implications for the investigation people could do with real world data. Our research implication of the use of velocity observation is also in line with the study of Baracchini et al (2020), where they also suggest have both temperature and current velocity for a complete system calibration. We will add these discussion after L359-366.

*Baracchini, T., Hummel, S., Verlaan, M., Cimatoribus, A., Wüest, A., & Bouffard, D. (2020). An automated calibration framework and open source tools for 3D lake hydrodynamic models. Environmental Modelling & Software, 134, 104787.*

**p.13 L395:** Missing end '.' of sentence?

Response: We will add the missing end ".".

**p.13 L395 - 399:** This sentence is confusing and might require rephrasing.

Response: We will rephrase the sentence to be: "Figure 3 illustrates that calibrating to temperature data only (red scatter plot) results in larger velocity errors $\Delta\overrightarrow{Vel}$ , relative to velocity errors when calibrating to velocity data only (Cali-Vel scenario, i.e., black scatter plot) or to both velocity and temperature data (Cali-Both scenario, i.e., green scatter plot)."

**p.14 L406 - 416**: I had the impression that many articles were missing here. This paragraph is hard to read in general and could potentially be revised.

Response: We will revise this paragraph to be: "Figure 4 shows the temperature error of solutions from three different calibration scenarios: Cali-Tem (red time-series), Cali-Vel (black time-series) and Cali-Both scenarios (green time-series). The errors between simulated and observed water temperature at

the surface, middle and bottom layers of two stations (STN. A1 and STN B1) are plotted. In general, the temperature error of the solution in the Cali-Both scenario is generally close to zero °C for all the layers and stations shown. The solution in the Cali-Tem scenario also got temperature error close to zero °C at the middle and bottom layer at STN. A1, but it has larger temperature error than solution in the Cali-Both at surface layer of STN. A1 and all layers of STN. B1. The solution in the Cali-Vel scenario generally overestimated the water temperature in all locations (i.e., all the surface, middle and bottom layers at both stations). The temperature error of solution in the Cali-Vel scenario is much larger than solution in the Cali-Tem and Cali-Both scenarios in the middle and bottom layer of both stations. The temperature error at most times, for the Cali-Vel scenario, is greater than 0.1 °C."

p.15 L431: *the* calibration...

Response: We will add "the".

**Figure caption Fig. 5 L451:** in term*s* of...

Response: We will change "term" to "terms".

**Figure 6:** Values in the darkest hex tiles are almost not readable.

Response: We will change the color of values in the darkest hex tiles to make it readable.

**p.19 L536 - 539:** This sentence was not clear to me and might require revision.

Response: We will revised the sentence to be: "Outlier or extremely bad solutions are also likely happen for calibration problems where the model output is very sensitive to the calibration parameters (i.e., a small change in model parameters can cause huge changes in the model output that leads to much worse solutions)."

**p.20 L553 - 566:** Again, the argumentation in this paragraph could be affected by the affect of synthetic observation data as outlined in the first major remark.

Response: We have responded the first major remark in response to R3-1. These hydrodynamic models are built based on physics and knowledge human learned in the past hundreds of years. Hydrodynamic models are not like most of hydrology models that with lots of uncertainty. They of course cannot be try the same as real world situation but they are close representations of the real world situation. We think the investigation on models can provide at least some implications for the investigation people could do with real world data. We will revise L553-566 to limit our language by not making universal statement.

**p.20 L573**: Remove *these*

Response: We will remove "these".

**p.20 L574:** suggest *to* have

Response: We will change "suggest" to "suggest to".

**p.21 L613:** We conclude that the Dynamically Normalized Objective Function *that* we propose

Response: We will revise the sentence to be: "We conclude that the DYNO objective function that we propose"

---

## Author Response (AR1)

**Response to Reviews**

We appreciate and would like to thank three reviewers for taking the time and effort to read our manuscript and expressing the generally positive impression of our work. We will improve our manuscript based on reviewer's helpful comments. Our point-to-point response are below (comments of the referee are in black, our responses are in purple, texts quoted from the revised manuscript are in italic).

**Point-to-point response**

**Response to referee comment Referee #1**

**R1-1:** This paper presents a new objective function for automatic calibration of 3D hydrodynamic models based on water temperature and velocity data. The case study is a tropical lake, where the authors tested their DYNO+PODS for calibrating a Delft3D model against a subset of previously simulated 3D flow and temperature fields from a run assumed to be the "truth". I strongly appreciate this approach as it provides the full control of the optimization. Moreover, as the authors stressed many times (maybe too many, a lot of repetitions could be avoided), the use of velocity data together with water temperature is not so diffused yet in the field of lakes hydrodynamic modeling but it is crucial to have consistent results and realistic flow fields. Hence I welcomed the author's effort in quantifying how important it is.

Response: Thanks for reviewer's comments.

**R1-2:** Optimization algorithms as well as suitable objective functions for calibrating complex models are needed in the wide environment of hydrological numerical applications. I enjoyed the reading of the manuscript, which is well written and clearly structured. I appreciated how the authors describe their DYNO and tested its performances. Some clarifications are required, in my opinion, to make the optimization part (which is not the focus of the manuscript but still a key part of it) more accessible to the wide public of HESS, but in general I believe this work is worthy for publication on HESS after some minor revisions.

Response: Thanks for the reviewer's comments. We have revised the optimization part in the revised manuscript (based on this comment and comment at R1-29 and R1-30). Detailed response refers to the response to the comment R1-29 and R1-30.

**R1-3:** I'd like the authors to consider deepening the analysis on two more aspects which I believe worth a little discussion:

Computational costs: Addressing this aspect is mandatory in a paper on optimization algorithms. The authors make some general considerations here and there, but maybe a dedicated paragraph would be more appropriate. My questions: How many (real) runs of the hydrodynamic model were necessary to get the final solution for e.g. each trial/each configuration of Dyno (temp, vel, both)?

What is the computational cost (wall clock time) of these tests? e.g., how much time compared to the error?

Response: Our study set the same evaluation budget (i.e., the maximum number of hydrodynamic model runs) for each trial and calibration scenario (i.e., Cali-Tem, Cali-Vel, and Cali-Both). The maximum number of hydrodynamic model runs in each trial is 192, which is about 8 iterations with 24 evaluations in each iteration. The result indicates that 8-iterations is a sufficient budget as the calibration progress plot in Figure R1-1 shows the optimization experiments almost converged in the last few iterations.

The computational time of one simulation is approximately 5 hours on a windows desktop with CPU Intel Core i7-4790 processor. However, when running 24 simulations simultaneously on a multi-core platform, the computational time gets longer because of the limited cache memory resources (as discussed in Xia and Shoemaker (2022)). Cache memory is a small amount of much faster memory than main memory. The wall-clock time for one iteration with 24 cores simultaneously running is about 12 hours if using the default process scheduling of the nonuniform memory access (NUMA) multi-core system. We used the mixed affinity scheduling proposed by Xia and Shoemaker (2022), and the wall-clock time is reduced to about 8 hours per iteration. The mixed affinity scheduling changed the default affinity setting by setting a hard affinity on the simulation of each PDE model (i.e., fixing the process of each PDE simulation to one core). This approach proved to be efficient for memory usage and reduced the simulation time. More details about the mixed affinity scheduling and the NUMA system can be found in the study of Xia and Shoemaker (2022). Hence, the wall-clock time of each trial takes about 64 hours (8 iteration×8 hours/iteration).

*Xia, W., & Shoemaker, C. A. (2022). Improving the speed of global parallel optimization on PDE models with processor affinity scheduling. Computer-Aided Civil and Infrastructure Engineering, 37(3), 279-299.*

We have added a separate paragraph in the end of section 2.7 of the revised manuscript as suggested by reviewers to discuss the computational cost of the experiments and we have also added the calibration progress plot (Figure R1-1) in the supplementary materials (as Figure S1).

[Figure]

Figure R1-1. Calibration progress plot of PODS in Cali-Tem, Cali-Vel, and Cali-Both scenarios. The best solution found so far in average of three trials is plotted with the number of evaluations. Calibration progress of PODS in the Cali-Tem scenario is plotted in (a), where the temperature error ($f_{Tem}(X)$ in Eq. (7)) of the best solution found so far is plotted. Calibration progress of the Cali-Vel scenario is plotted in (b), where the velocity error ($f_{\overline{Vel}}(X)$ in Eq. (10)) is plotted. The calibration progress of PODS in the Cali-Both scenario is plotted in (a) and (b) in terms of temperature error and velocity error, respectively, of the best solution found so far based on Eq. (12).

**R1-4:** Application to real data from observations: The authors auspicate that future users will test the DYNO against observations and so do I. So my questions are: Are there any constraints in the time/space frequency of the observations? In order to calibrate e.g. their Delft3D lake model to some temperature profiles and some current measurements, how should this data be? E.g. should temp and vel be simultaneous/in the same locations/same depths? As far as I understood, this is indeed the case of the data used in the authors' application, but this is not that common in standard monitoring schemes, where data are sparse and often not simultaneous. So basically, does this optimization (I guess this applies to PODS rather than DYNO) handle sparse observation? Did the authors test their DYNO+PODS by changing e.g. the sampling time or the number of locations of their "truth"? I guess the more data the better, but I'd greatly appreciate some discussion on this. Is there an optimal number of locations/time frequency which gives a satisfactory calibration?

Response: The reviewer is right that we used the temperature and velocity data at the same locations and depths. We are not sure what the result would be if the temperature and velocity data were collected at different places. In reality, we think it is possible that people do measurements of temperature and velocity at the same locations (or locations sufficiently close to each other). We agree that people might measure temperature and velocity at different temporal frequencies

(for example, the observation data of one variable might be sparser than that of another). In general, there is no problem in using PODS on problems with sparse observation in the technical aspect. PODS can handle sparse observation. And we don't require the observation of temperature or velocity to be at the same location or with the same frequency since the error of temperature and velocity are calculated separately.

DYNO also does not require the same number of locations / same time frequency for different calibrations constituents. Regardless of number of locations / sampling frequencies, DYNO normalizes the objective function of each constituent dynamically, to allow equal weight to each constituent in the calibration process. In the calibration setup of this study, we believe DYNO has an advantage since it can dynamically adjust the weights between the error of temperature and velocity. In cases when the locations and time-frequencies are vastly different for the two variables, it may be reasonable to introduce custom weights to give more weightage to one constituent. For instance, if velocity observations are limited in both space and time, a custom weight could be introduced in DYNO to reduce overall weightage of velocity in calibration.

We don't think there is an answer to the optimal number of locations and time-frequency. The frequency needs to be small enough to capture the time variation (e.g., diurnal variation or seasonal variation). The number of locations seems dependent on the geography or the reservoir's shape. The location should be enough to capture the spatial variations. The number of locations might also depend on the budget available to do these measures in real practice.

We have added the above discussion into the revised manuscript in the end of Section 3.5.

**Attached are few minor comments/suggestions to improve the paper.**

**R1-5:** Below few minor comments/suggestions to improve the paper. I strongly recommend the authors to revise the English language as I found some typos (missing spaces, wrong singular/plural verbs) and some passages to be improved, especially in the Abstract, Introduction and Methods. Figures are ok but some "transparency" boxes from the png (my guess) are visible so please consider improving the quality or changing the figure format.

Response: Thanks for the reviewer's comments. We have revised the English language based on reviewer's detailed comments below. We also did a careful proofreading for the revised manuscript. We have improved the quality of the figures by removing these "transparency" boxes.

**R1-6:** Title: Please synthetize the title: my suggestion: A Novel Objective Function DYNO for Automatic Multivariable Calibration of 3D Lake Models

Response: Thanks for the reviewer's suggestion. We have changed the title to be: "A Novel Objective Function DYNO for Automatic Multivariable Calibration of 3D Lake Models."

**R1-7:** L19-20: "by comparing the result of using DYNO to results of calibrating to either temperature or velocity observation only" please rephrase

Response: We have revised the sentence to be: "*We also investigated the efficiency of DYNO by comparing the calibration result obtained with DYNO to the result obtained through calibrating to only temperature and to the result obtained through calibrating to only velocity.*" (in line 17-19 of the revised manuscript)

**R1-8:** L27-33: please revise the English form and make it less general. An example: "Hydrodynamic models simulate the hydrodynamic and thermodynamic processes in lakes and reservoirs": not really, "hydrodynamic models" is a very wide definition for models that can be used to simulate either hydrodynamics only or hydro-thermodynamics (as for Delft3D), and to different water environments, not only lakes. This is just a formal comment and applies to the entire Introduction, please avoid generalized and rough statements as well as repetitions.

Response: We have revised the sentence to be "*Lake hydrodynamic models simulate the hydrodynamic or thermodynamic processes in lakes and reservoirs …*" (in line 27 of the revised manuscript). We have also changed "hydrodynamic models" to "lake hydrodynamic models" in the rest part of the Introduction section.

**R1-9:** L30: The authors say that hydrodynamic models simulate specific water quality variables. What do they mean with "specific water quality variables"?

Response: We want to say hydrodynamic models are often built to support water quality modelling of variables such as nutrients, toxins. We have rephrased the sentence to be: "*These simulation models (e.g., hydrodynamic modelling) play a critical role in managing water bodies, as they are built to support the simulation of the spatial and temporal distributions of specific water quality variables (e.g., nutrients, chlorophyll-a), and to study the response of a water body to different future management scenarios.*" (in line 28-31 of the revised manuscript)

**R1-10:** L46: "some" water variables. I'd say all of them! If the model is 1D, all variables will be 1D. Also in this case, please be more precise: "spatial" is very general, 1D models typically consider the vertical dimension, so what they don't provide is the horizontal spatial distribution and in general they can't capture the 3D processes (e.g. circulations, 2D waves…).

Response: We have revised the sentence to be: "*However, one-dimensional models are unable to simulate the horizontal spatial distribution and cannot capture the 3D processes, and thus may not be suitable for certain studies.*" (in line 46-47 of the revised manuscript)

**R1-11:** L66-75: The authors could be interested in reading this work https://doi.org/10.1016/j.envsoft.2021.105017 and references therein where some of the issues mentioned in this paragraph are addressed in a manual calibration of Delft3D in a lake.

Response: Thanks for reviewer's recommendation. The study mentioned by the reviewer is interesting and relevant. The study used different sources of temperature data (from in situ observations, multi-site high-resolution profiles and remote sensing data) to compensate for unavailability/scarcity of velocity measurements. This is a practicable approach when there is no velocity data available and there are such different sources of temperature data available. In cases, there is no high-quality remote sensing data (for example because of cloud) or a large amount of high-resolution profiles of temperature measurement it is still challenging to verify the spatial simulation of hydrodynamic quantities. We have discussed this study in our revised manuscript (line 85-90).

**R1-12:** L85-90: a little confused, please revise English form.

Response: We have revised the sentence to be: "*Lake hydrodynamic models predict the velocities throughout a water body. Accurate velocity simulations are thus important to understand the spatial distribution of water quality problems (e.g., algal blooms) in sizeable lakes. Hence, during*

*calibration of these models it is useful to know whether efforts to measure velocity directly are justifiable even if temperature data is already available. We will examine the extent to which direct measurement of velocities justify the extra effort by giving more accurate results for hydrodynamics models. We will also look at the error of the spatial distribution of hydrodynamics associated with calibrating to temperature only, which is rarely studied in literature.*" (lines 91-97 in the revised manuscript)

**R1-13:** L100: etc → among others; desire → require

Response: We have replaced "etc" with "among others" and replace "desire" with "require" (in line 107-108 in the revised manuscript).

**R1-14:** L108: "A key challenge for automatic calibration of multi-variable calibration problems is in defining a suitable objective function to calibrate multiple variables simultaneously": please remove some "calibrations", e.g.: A key challenge for automatic calibration of multi-variable problems is in defining a suitable objective function.

Response: We have revised the sentence to be: "*A key challenge for automatic calibration of multi-variable calibration problems is in defining a suitable objective function*" (in line 113-114 in the revised manuscript).

**R1-15:** L110: varying → vary

Response: We have replaced "varying" with "vary". (in line 118 in the revised manuscript).

**R1-16:** L118: Anticipate MOO to the first time it is mentioned (L113). Does SOO refer to the optimization methods mentioned in lines (105-112)? If yes, please anticipate SOO as well.

Response: We have anticipated MOO to the first time it is mentioned. We have also anticipated SOO at line 114-117: "*Traditional approaches typically formulate the goodness-of-fit of multiple variables into a single objective function by adding weights between the goodness-of-fit of multiple variables and solve the problem with single objective optimization (SOO) techniques*"

**R1-17:** Tab.1 check spaces

Response: We have corrected the spaces in Table 1.

**R1-18:** L164: (e.g. calibrating temperature and…) we got that the authors are dealing with multi-variables problems and in particular with temp and vel. Please revise the paper critically and remove repetitions.

Response: Thanks for reviewer's comments. We have revised the paper critically and removed repetitions.

**R1-19:** Sect.2.3: I see that the point of this work is not the simulation of one specific case study, but since the name of the section is "Study site and data" the authors could at least include the name of the lake and a few morphological characteristics (e.g. where it is located, how deep and large it is) and then refer to Xia et al. 2021 for all other details. Also what year was simulated should be reported for completeness (in the text and in fig. 1).

Response: We agree with the reviewer that it would be better to provide the name of the lake. However, we are constrained by the local agency, which provided us with data, from releasing the

lake name, including other confidential information such as reservoir locations and inflow and outflows. We appreciate if the reviewer understands our situation. But we are permitted to mention the depth and the surface of the reservoir. Hence, we added these information (e.g., reservoir size, depths, and the simulation year) in section 2.3 (line 189-192) of the revised manuscript.

**R1-20:** L189: just a curiosity, why are the names of the station A1, B1-4? Does this A1 station mean something different than the others?

Response: Station A1 is the station where the real measured temperature data was used for the calibration of the hydrodynamic model. While there is no real observation data at other stations.

**R1-21:** L211-212: "the water utilities' employees and consultants": some specification is missing here... maybe Singapore?

Response: As responded to the comment above, we are not allowed to disclose the confidential information such as reservoir locations. So we did not mention the name of the local agency.

**R1-22:** L234: Secchi depth: what about the space-time variability of this parameter? Delft3D allows to consider both variations, as transparency is not a constant and uniform feature of water. Did the authors consider this possibility?

Response: The reviewer is right that Delft3D allows considering space-time variability of these parameters. We consider time-varying parameters such as Secchi depth, Ozmidov length scale, Dalton number, and Stanton number to be constant mainly because it would be very challenging to calibrate them as time-varying variables. Considering these parameters as time or space-varying parameters will substantially increase the number of decision variables in the optimization. In addition, the reservoir we study is relatively small (with maximum depths of 22 m and a surface of 250 hectares), and it is located in a tropical region where there is no significant seasonal variation. Hence we think it is acceptable to consider them as constant. But we think that considering these parameters' space-time variability into optimization calibration would be an interesting future topic. In that case, new methods for reducing the parameter dimensions are needed (e.g., design some low dimensional controlling parameters to represent the high dimensional space-time variability of these parameters). We have added these discussions in the end of section 2.4 (line 252-257). We also discussed in the end of section 3.6 that the consideration of the spatial-temporal variability of the calibration could be a possible future work.

**R1-23:** L237: please consider moving here the sentence in lines 228-230.

Response: We have moved the sentence in lines 228-230 to Line 247-248 and revised it to be: "*The last parameter is the manning coefficient, which affects the roughness of the bottom of the lake and has a direct impact on velocity.*"

**R1-24:** L241: Model parameter(s) in table caption

Response: We have replaced "Model parameter" with "Model parameters" in Table 2.

**R1-25:** Tab.2: Why didn't the authors include the coefficient of free convection, whose value greatly affects the modeled temperature? The default value in Delft3D is 0.14 but it strongly modifies the thermal profile when tuned.

Response: We chose the calibration parameter set as suggested by the local experts on lake modeling (for the region of the study site) and they did not suggest the coefficient of free

convection as an optimization parameter. As the reviewer pointed out, this parameter affects the thermal profile, and could be considered as a parameter to be optimized in future.

**R1-26:** L261: Have the authors considered normalization of RMSE by the standard deviation instead of the mean and why did they eventually chose the mean as normalization factor?

Response: We did not considered normalization of RMSE by standard deviation. RMSE could be normalized with standard deviation or mean. We just chose one of them (in our case we use the mean).

**R1-27:** L272-273 and Eq.11: It looks to me that there is a power of 2 missing in eq.11. The referred papers (starting from Beletsky et al. 2006) present this formula with the module of the difference between observations and models (at the numerator) and the module of the observations vector (at the denominator) both to the power of 2, and then everything under the square root. If the authors prefer to use the Euclidean norm instead of the module it is fine with me, as ||x||_2 = |x| in R^2, but the power of 2 should be maintained anyway, right? I checked the codes uploaded on github and I see a **2 in the computation of the Fourier norm, but please double check the equation, the code and the references.

Response: The reviewer is right that there is a power of 2 missing in eq.11. We have revised eq.11 in the revised manuscript.

**R1-28:** L280: "being summed them" remove them

Response: We have removed "them".

**R1-29:** Sect.2.6.1. This paragraph seems like an advertisement of PODS and is highly technical. I'm not sure it is adequate to the wide audience of HESS. The authors could consider limiting the acronyms to those that are really needed (e.g. do we really need to know that "DYCORS inherits the dynamic coordinate search idea from DDS (Tolson and Shoemaker, 2007) to improve its effectiveness and efficiency for high dimensional problems"?) and try to clarify a bit. What does RBF surrogate mean? Maybe the authors could consider merging sect 2.6.1. and 2.6.2. and try to explain things in easier terms (and referring to publications for high-technical details), eventually splitting some very long sentences.

Response: Thanks for reviewer's comments. We have revised the section 2.6 by merging section 2.6.1 and 2.6.2 as suggested by the reviewer. We rewrote the text by explaining things in easier terms and added more details than previous version. These changes include more details about RBF surrogates and how they are used in the optimization.

**R1-30:** Figure2 and commenting text: two aspects are not completely clear to me: 1) how does the RBF surrogate model interact with the new fitness F and how does it then communicate to worker_i such that the latter can run a new hydrodynamic simulation? I believe that lines 318-320 are crucial here. The authors could consider expanding these two lines as it seems to me they are taking for granted too many things. 2) how are workers-p related one another? Are all steps M1-M6 performed independently on each processor? If yes, how do they communicate at the end to define the best solution found? If not, do M1-M6 steps consider all trials from workers-p within ψ? And how does M6 discriminate to which worker-i it should communicate the new X?

Response: After one iteration is finished (e.g., the simulation of the $P$ hydrodynamic simulations). The objective function value $F(X|K)$ of all evaluations in $\psi$ (including the newly finished

evaluations in this iteration and previous iterations) are recalculated. The RBF surrogate model is rebuilt with the new objective function value $F(X|K)$ that are calculated based on Eq. (3). The newly built RBF surrogate model is then used for selecting the evaluation points for the next iteration. In PODS, the algorithm first generates a large number of candidate points around the best evaluation found so far. Then $P$ evaluation points are selected from the candidate points using a Surrogate-Distance Metrics discussed in the PODS paper. The metrics consider the approximated objective function value based on the surrogate model and the distance of the evaluation points from the evaluated points. We have revised lines 318-320 (in original manuscript) by adding more explanations.

To reviewer's question 2), the tasks of $P$ workers are independent of each other (i.e., each worker evaluates one hydrodynamic model). The steps M1-M6 are master's tasks, performed after the tasks of $P$ workers are finished in each iteration. So each worker sends back to the master the results of one simulation (i.e., temperature and velocity error of one evaluation). Then the master adds all these newly obtained results to the history list $\psi$, which contains the results of all evaluated points in previous iterations. M1-M7 only need to be performed on one processor. The jobs of M1-M7 is to generate $P$ evaluation points and then distribute the $P$ evaluation points to $P$ processors for the workers' tasks. The master M7 distributes the $P$ evaluation points randomly to the P processors so each processor will get an evaluation point and send back the evaluated results to master. We have modified the Figure 2 in the original manuscript as below Figure R1-2 to make it clear.

[Figure]

**Figure R1-2 (Revised from Figure 2 in original manuscript).** Diagram of the implementation of DYNO with the parallel algorithm PODS. $P$ is the number of processors available. The green texts (i.e., steps W3, M1-5) are changes made on PODS to incorporate DYNO. The rest part follows the original PODS method.

**R1-31:** Table 4 and commenting text: a "good model" range for Fn should be between 0 (perfection) and 1. How do the authors comment such large values of Fn? Their best solution Cali-borh gives almost 2, while the Cali-Vel, which should minimize only Fn, gives almost 3!

Response: The value in Table 4 is the sum of Fn value at multiple stations (in total 12 locations). So the Fn value at one location in average is about 0.167 for the best solution in Cali-Both scenario and 0.25 for best solution in Cali-Vel scenario. Hence it is not a large value of Fn. We have added these explanations in the manuscript: "*It also worth mentioning that the value in Table 4 is a sum of temperature or velocity error at multiple (in total 12) locations. Hence the error at each location is smaller than the value in the table.*" (line 415-417 in the revised manuscript)

**R1-32:** L395: missing dot before Figure3?

Response: We have added the missing dot. (in line 466 of the revised manuscript)

**R1-33:** L421: overestimated ◊ overestimation

Response: We have replaced "overestimated" with "overestimation". (line 495 in the revised manuscript)

**R1-34:** L433-434: Latin Hypercube Designs (LHD) is mentioned here for the first time. Please specify what it is used for in the methods section.

Response: We have added the introduction of the Latin Hypercube Designs (LHD) in the method section (i.e., section 2.6.2, line 327-328 in the revised manuscript).

**R1-35:** L456: "only the best solution in each of the optimization iterations are plotted" -> is plotted

Response: We will replace "are" with "is". (in line 563 of the revised manuscript)

**R1-36:** L462 please change the asterisk with standard math notation

Response: We have changed "$X_i, i = 1, ..., 3 * N_{max}$" to "$X_i, i = 1, ..., 3 \times N_{max} ...$" (in line 569 of the revised manuscript)

**R1-37:** Figure5 and 6 Please make the axis labels consistent between the two figs. fvel(X) and ftem(X) should be fine for fig. 5 (if I got correctly that only the best K is plotted), while ftem(X|K) and fvel(X|K) should be fine in fig. 6

Response: Thanks for reviewer's suggestion. We have revised the Figure 5 as the reviewer suggested, which is Figure 6 in revised manuscript.

**R1-38:** Figure 6: please improve the readability of the number inside the darkest hexagon by either changing the darkest color or modifying the color of the number (e.g.) yellow. Please note that there is a missing C in (c) in the figure legend.

Response: Thanks for reviewer's comment. We have modified the color of the number inside the darkest hexagon and also add the missing "C" in the figure legend (Figure 6 in the original manuscript is now Figure 7).

**R1-39:** I would have expected the darkest hexagons to be the closest to the origin. Why isn't it like that? What is the hexagon containing the final solution? The authors could consider highlighting it e.g. with a bold colored contour line.

Response: The figure the reviewer is mentioning is Figure 6 in the original manuscript. It is now Figure 7 in the revised manuscript (we refer it to Figure 7 in blow text). We would like clarify here that Figure 7 represents the joint distribution of the i) Velocity Error and the ii) Temperature error components of DYNO for "all simulations evaluated" during the different calibration scenarios. Hence, Figure 7 is a representation of the optimization search dynamics for the three different objectives analyzed (i.e., Cali-Tem, Cali-Vel and Cali-Both). Figure 7 shows that when calibrating to only temperature or velocity, the search of the optimization only considered the error of one variable and ignored the error of another variable. Hence the darkest hexagon (i.e., the concentration of error distribution) is expected to be close to one of the coordinate axes instead of the origin (e.g., in Figure 7(a) more solutions are found with only better Temperature error, hence darker hexagons are close to vertical axis). When calibrating to both temperature and velocity, the darkest hexagons are close to the origin but not necessarily the closest. This could happen for many reasons. For example, the solution space is multi-modal, and many solutions have the same error around the value of the darkest hexagons. It could also happen that the algorithm searched more

around the region that is not close to the best solution. In general, we think it is true that the best solution is the closest to the origin when calibrating to both temperature and velocity, but we don't think it is true that the darkest hexagons must be the closest to the origin.

In Figure 7 of the revised manuscript, we have highlighted the hexagon containing the final solution in each of the three random trials.

**R1-40:** L485: represent (remove s) (in line 592 of the revised manuscript)

Response: We have removed "s".

**R1-41:** L534: please correct, which one is better than the other? N2 better than N1?

Response: We have corrected the sentence to be "…*with DYNO-N2 might be better than with DYNO-N1*" (in line 641 of the revised manuscript)

**R1-42:** L537: like→ likely?

Response: We have revised the sentence to be: "… *also likely happen for*…" (in line 644 of the revised manuscript)

**R1-43:** L565: helps to improve the calibrate of → calibration

Response: We have replaced "calibrate" with "calibration". (in line 671 of the revised manuscript)

**Response to referee comment Referee #2**

Overview:
**R2-1:** The manuscript is well-written and clear in its intent and results. It is close to publishable if the authors place their approach to calibration in the appropriate context. I am classifying this as "major revisions" because I think the issues are important, but they should not necessarily take a lot of work to implement.

Response: Thanks for reviewer's comment. We have carefully considered the reviewer's comments and will revise the manuscript accordingly based on the comments.

Specific Comments

**R2-2:** 1. I would have liked to see the introduction have a little that explains the context for the authors choice to test their calibration against a calibrated model rather than against observations. This choice gives them more data to compare against, but with the drawback that the "true" data are biased in exactly the same way as their calibrated results. This is an acceptable, but limited approach -- acceptable because it is useful in understanding and illustrating the new calibration method -- but limited in that it cannot be used to say anything about what might occur when compared to real world data. This idea is emphasized in comment 2 below.

Response: We don't have real observation for the velocity data, and we found in literature it is also rare that people calibrate to both temperature and velocity. This is why we did our analysis with the hypothetical dataset. Using a calibrated model to generate synthetic data is thus the plausible

alternative we used to generate observation data. We agree that this approach has limitations. However, since, hydrodynamic models are built based on physics and human-acquired knowledge, they are close representations of real-world situations. Hence, we think that synthetic data generated from calibrated models is a reasonable alternative for real-world data, for this study.

We have revised our introduction and text to explain the context why we do investigation, not on real observations, but on synthetic data. We agree that any conclusions drawn from this study should be considered cautiously and hence we have also soften our statements about the value of this analysis in a real-world context (we have emphasized in response to comment R2-3 below).

In the end of introduction (line 139-140), we have added: "*Since velocity measurements are usually not included in standard lake monitoring systems (whereas temperature measurements are included), real velocity observations are seldom available (Amadori, et al, 2021). We don't have real observation for the velocity data in our case as well. Hence, we did the investigation based on synthetic observations generated from a calibrated model. It is worthwhile to revisit and validate this analysis with real velocity measurements if they are available in future.*"

**R2-3:** 2. I strongly disagree with the penultimate sentence of the abstract, that the study "suggests that both temperature and velocity measures should be used for hydrodynamic model calibration in real practice."  Similar language is found elsewhere in the paper. The model "suggests" nothing and we must remain skeptical about the value of calibration in a real world context without direct illustration of its importance. Unfortunately, the authors' methodology does not support this suggestion or any suggestion about real-world practice. The authors are testing their calibration against results of a calibrated model -- not the real world. The ability to more precisely capture the calibrated model (by a variant of the same model) cannot be used to imply that real world will be also represented more accurately.  This is a fundamental confusion of "precision" -- how close are my answers to grouped together, with "accuracy" -- how close do my answers reflect the real world. The authors have not included any comparisons of their model to real-world data hence they cannot make any statements or suggestions about likely accuracy in representing real-world phenomena.

Response: As responded to comment R2-2, we don't have real observation for the velocity data, and we found in literature it is also rare people calibrate to both temperature and velocity. That's why we did our analysis with the hypothetical dataset. Reviewer holds the view that the model "suggests" nothing. However, the physical model we use is established with scientific knowledge and the laws of physics. Hence, it is reasonable to do analysis on synthetic observation data generated from a hydrodynamic model when we do not have real observations available. We agree with the reviewer that we should be aware of the limitations of the model results.

We have revised the sentence in abstract to be "*Our study implies that for practical application, an accurate spatially distributed hydrodynamic quantification, including direct velocity measurements are likely to be more effective than using only temperature measurements for calibrating a 3D hydrodynamic model.*" (in line 21-23 of the revised manuscript)

We added a new paragraph in the new section 3.6 Future Work: "*Our analysis of the role of temperature and velocity measurements in 3D hydrodynamic model calibration is based on synthetic observation data generated from models. We do think it is worthwhile to further*

*investigate the role of temperature and velocity measurements in hydrodynamic model calibration if there are velocity measurements available in future.* ”

We have also revised other similar statements in our manuscripts. For example in the conclusion, we have added the sentence: “*Our analysis suggests that for practical applications, both temperature and velocity data might need to be considered for model calibration. The common practice of calibrating only to temperature data might not be sufficient to reproduce the flow dynamics accurately and extra effort and expense to collect velocity data is expected to give a beneficial effect. However, our analysis are based on synthetical data from models, hence it is worthwhile to further investigate the role of temperature and velocity in model calibration with real temperature and velocity measurements.*” (in line 764-769 of the revised manuscript)

**R2-4:** 2. If the authors want a stronger paper, they can either compare to real-world observations in a different time period as a classic validation test, or they can examine some important phenomena that are not directly included in the observational data set. For example, the timing of global overturns of the lake are arguably an important phenomena that can be computed from real-world observed data -- the model results for the different calibrations in predicting timing of overturns could be analyzed. I may be wrong, but I suspect that the differences between the various models may not be as significant when compared in their prediction of a large-scale phenomena. The paper would still be publishable, but future researchers would be able to see that control of model biases are likely more important than calibration in capturing real world behaviors.

Response: We don't have data at a different time period to do this validation test. We don't have real observation data for temperature and velocity either. This is why we use the hypothetical dataset for the investigation. The reviewer mentioned the examination of important large-scale phenomena such as overturn. First, we have to mention that this is out of the scope of our study. The model we built for the tropical reservoir is not used for studying overturn. Our data is based on shallow lake in a location with almost no seasonal variation in ambient temperature. Hence, we would not expect overturn to be a significant event in our data set. Lake overturn (e.g., enhanced mixing of upper and lower water levels due to a temperature gradient caused by seasonal air temperature changes) is a complex phenomenon, so one would expect there is an advantage to having both temperature and velocity data information to model this overturn situation. Since the focus of our argument is on obtaining some velocity data rather than entirely relying on temperature data, this argument obviously is valid for lake overturn. There does not seem to be a need to lengthen the paper by adding a section on lake overturn. In addition, we think, to study such large-scale phenomena, it might not be necessary to build a complex 3D model since a 1D or 2D model can do the same job. There are many other purposes of building a 3D model, such as supporting water quality modelling (e.g., the horizontal and spatial distribution of temperature or other water quality parameters like nutrients or toxins). In such a case, the correct temperature and velocity modeling is very important.

**R2-5:** 3. The authors should include a comparison to an uncalibrated run (using conventional default values).

Response: We have included the comparison to an initial uncalibrated simulation in the revised manuscript (for example in Figure 3 and Figure 4). The parameter value of the uncalibrated

solution (in Table S1) is set be the middle of the calibration range in Table 2. Both Figure 3 and Figure 4 shows that solutions of all the three scenarios improved the temperature fit compared with the initial solution, which demonstrate the effectiveness of the optimization calibration.

**R2-6:** 4. The authors should discuss the choice of calibration parameters -- how were they chosen, why were they chosen, and what does it mean to hold these as constant parameters. Arguably, a sensitivity study should have been done prior to choosing the calibration parameters. I am somewhat concerned about whether it is physically meaningful to calibrate as fixed parameters values that arguably depend on time-varying physics (e.g., Secchi depth, Ozmidov length scale, Dalton number, Stanton number).

Response: We selected the parameter values by discussing with an expert who is familiar with the Delft 3D simulation model, and he suggested the parameters we optimize. We did not carry out a sensitivity analysis because it requires many simulations. For an expensive simulation model, this means you need to reduce the number of optimization evaluations given a limited total budget for sensitivity analysis and optimization evaluation. Sensitivity analysis (especially global sensitivity) is expensive to do for our problems. We have nine parameters, and each simulation takes about 5 hours to run. Cheaper sensitivity analysis like OAT (one at a time perturbation of parameter value) depends on one's guess of the probable best solution, around which the sensitivity should be done. The sensitivity is the first partial derivative of f($X$) with respect to the parameter evaluated at that probable best solution. For example let the function be f($x_1$, $x_2$) = $x_1$ +$(x_2)^2$. If you assume the best solution is $x_1$=4 and $x_2$=2, then the most sensitive parameter is $x_2$. If you assume the best solution is $x_1$=1 and $x_2$=1/4, then the most sensitive parameter is $x_1$. So this shows OAT is not necessarily reliable for this kind of problem with lots of parameters and a limited computational budget. Discussion of sensitivity analysis methods would add many pages to the manuscript, and we are trying to shorten it. Sensitivity analysis is also not the topic that we are focused on. So here we are using the reasonable practice of getting estimates of sensitive parameters by talking to an expert who has used the model on similar lakes.

In terms of time-varying parameters such as Secchi depth, Ozmidov length scale, Dalton number and Stanton number, we consider them as constant mainly because it would be very challenging to calibrate them as time-varying variables. Considering these parameters as time or space-varying parameters will substantially increase the number of decision variables in the optimization. In addition, the reservoir we study is relatively small (with maximum depths 22 m and the surface 250 hectare) and is located in a tropical region where there is no significant seasonal variation. Hence, we think it is reasonable to considering them as constant. But we do think that considering the space-time variability of these parameters into optimization calibration would be an interesting future topic. In that case, new methods on how to reduce the parameter dimensions are needed (e.g., designing some low dimensional controlling parameters, like curve number in hydrology (Bartlett et al. 2016), to represent the high dimensional space-time variability of these parameters).

*Bartlett, M. S., Parolari, A. J., McDonnell, J. J., & Porporato, A. (2016). Beyond the SCS‑CN method: A theoretical framework for spatially lumped rainfall‑runoff response. Water Resources Research, 52(6), 4608-4627.*

We have added these discussions in the revised manuscript. In section2.4 we added: "… *These are also the parameters included in the routine model calibration by local experts and thus*, .... *Some*

*of these parameters might be spatio-temporal variant (such as Secchi depth, Ozmidov length scale, Dalton number, and Stanton number). Considering these parameters as time or space-varying parameters will substantially increase the number of decision variables in optimization. Considering that the reservoir in our study is relatively small and it is located in a tropical region where there is no significant seasonal variation. Hence to simplify the problem, we consider these parameters as constant.*"

We also have added some discussion in the new section 3.6 Future Work: "*Another possible future work is the consideration of the spatial-temporal variability of calibration parameters (such as Secchi depth, Ozmidov length scale, Dalton number, and Stanton number). We considered them as constant parameters in our study to simplify the problem. This is reasonable since our study area is relatively small and there is not much seasonal variation. In cases where the study areas are large and there is significant seasonal variation, there might be a need to consider these parameters as space and time-varying calibration parameters. The consideration of space and time variability will of course increase the number of decision variables for the optimization problems, which will bring more challenges. In that case, new methods on how to reduce the parameter dimensions might be needed (e.g., designing some low dimensional controlling parameters, like curve number in hydrology (Bartlett et al., 2016), to represent the high dimensional space-time variability of these parameters).* " (in line 738-747 of the revised manuscript)

**R2-7:** 5. The authors should provide some discussion about the physical meaning of the errors in the results. For example, the Ozmidov length scale in the "true solution" is 0.015, but the "best" calibration has more than double this value; at the same time, the background vertical diffusivity in the calibration is about 3/5 of the true value and the Secchi depth is overestimated by 13%. These all exert controls on vertical mixing, and the wide disparities for different calibration methods makes me question as to whether calibration can really be effective to capture the complex, time-varying behaviors of vertical mixing in a stratified system with the given turbulence model. I do not expect the authors to solve such a problem, but I think they should discuss the implications. Replacing Table 5 with a well-constructed bar graph (with appropriate normalization) would help the readers see which parameters are doing all the work.

Response: It is always challenging to explain the physical effect of parameter error when multiple parameters are being calibrated simultaneously. The relation between parameter value and the simulation results is nonlinear, and it becomes more complex when multiple parameters interact with each other. The reviewer has quantified the difference in "actual" parameter value and "best" parameter value obtained from calibration as a percentage. This might give the impression that the solution is very inaccurate. We are not sure if it is meaningful to look at the parameter error in percentage since some of the parameters value are very small; hence it might lead to a large percentage value when dividing the parameter error by the parameter value. In addition, the solution is obtained with a limited computing budget. Hence it is difficult to find the exactly true solutions in such a limited computing budget. We think it might be more meaningful to visually compared parameter sets values of different calibration scenarios with the true parameter set. Our goal is to show that calibrating to both temperature and velocity gives a better solution than calibrating to one of them. Figure R2-1 below shows the parameter value of solutions under different scenarios. The parameter value of the solution in calibrating to both is generally closer to the true solution than the other two cases.

The reviewer also questioned whether calibration could really be effective in capturing the complex, time-varying behaviors of vertical mixing. We have analyzed the vertical temperature profiles (Figure R2-2), the solution in the Cali-Both scenario can almost capture the vertical time-varying temperature profiles of the true solution. In contrast, calibrating to one variable did not fully capture the vertical time-varying temperature profiles. (For example, in April-May for Cali-Tem scenario, in Mar-May, Aug-Sep for the Cali-Both scenario.) We also analyzed the vertical time-varying velocity, eddy diffusion and viscosity (Figure R2-3, Figure R2-4, Figure R2-5 ). The result also indicated that the result in the Cali-Both scenario captures better vertical mixing than the other two scenarios.

In our revised manuscript, We have replace Table 5 with Figure R2-1 below. We have put Table 5 into supplement since some reader might prefer to see table than figure. We have added the discussion of the parameter error in the section 3.1.2: "*The objective function value in terms of temperature and velocity composite error (over multiple locations) ($f_{Tem}(X)$ and $f_{\overline{Vel}}(X)$, as formulated in Eq. (7) and (10), respectively) and the corresponding parameter configuration ($X^*$) of the selected solution (among three trials) are plotted in Fig. 5 and reported in Table S1. In general, the solution in the Cali-Both scenario is closer to the True solution than solutions in other two scenarios. The solution in the Cali-Both scenario is closest to the True solution at four parameters (i.e., $D_H^{back}, D_V^{back}, H_{Secchi}, c_H$). In addition, besides the Manning coefficients, the solution in the Cali-Both scenario is not the worst at any parameter. In contrast, solution in the Cali-Tem in worst at five parameters (i.e., $v_H^{back}, D_H^{back} v_V^{back}, D_V^{back}, c_e$) and solution in the Cali-Vel is worst at $L_{oz}, H_{Secchi}, c_H$. This indicates that calibrating to both temperature and velocity can help to prevent very bad value of the 9 calibration parameters.* " (line 444-452 in the revised manuscript)

We have added the results about the vertical time-varying temperature, diffusion, viscosity profiles into the supplementary document. In the revised manuscript, we have added a paragraph discussing the time-varying vertical mixing simulation: "*Simulation of vertical temperature, vertical velocity, vertical eddy diffusivity, vertical eddy viscosity (Fig. S2-S5) also shows that the solution in the Cali-Both scenario is much better than solutions in the Cali-Tem and Cali-Vel scenario. For example, the solution in the Cali-Both scenario can almost capture the vertical time-varying temperature profiles of the true solution. In contrast, calibrating to one variable did not fully capture the vertical time-varying temperature profiles (e.g., April-May for Cali-Tem scenario; Mar-May and Aug-Sep for the Cali-Both scenario in Fig. S2.) The solution in the Cali-Both scenario also give much smaller vertical velocity, eddy diffusivity, and eddy viscosity error than solutions in other two scenarios (in Fig. S3-S5). The result indicates that using both temperature and velocity data in model calibration also helps to improve the complex time-varying vertical mixing behavior.*" (line 498-506 in the revised manuscript)

[Figure]

Figure R2-1. The parallel axis plot for the parameter value and the composite error of temperature and velocity of calibration solutions under different scenarios (Cali-Tem, Cali-Vel, and Cali-Both). True solution defined in Table 2 is given for reference.

[Figure]

Figure R2-2 The change of vertical temperature profiles at STN. A1 with the change of time. The result of three calibration scenarios (Cali-Tem, Cali-Vel, and Cali-Both) and the True solution are plotted.

[Figure]

Figure R2-3 The absolute vertical velocity error at STN. A1 between the calibrated results (in the Cali-Tem, Cali-Vel, and Cali-Both scenarios) and the true solution. The change of absolute vertical velocity error is plotted with the change of time.

[Figure]

Figure R2-4 The absolute vertical eddy diffusivity error at STN. A1 between the calibrated results (in the Cali-Tem, Cali-Vel, and Cali-Both scenarios) and the true solution. The change of absolute vertical eddy diffusivity error is plotted with the change of time.

[Figure]

Figure R2-5 The absolute vertical eddy viscosity error at STN. A1 between the calibrated results (in the Cali-Tem, Cali-Vel, and Cali-Both scenarios) and the true solution. The change of absolute vertical viscosity error is plotted with the change of time.

Path to acceptance:

**R2-8:** The authors need to carefully limit their language so that the manuscript reflects what can truly be understood from the model and refrain from speculation about real-world behaviors unless they specifically bring new real-world comparisons into the manuscript.

Response: We have revised our language and been skeptical about the model results (as we have responded to R2-1, R2-2).

**Response to referee comment Referee #3**

**General evaluation and major comments**

**R3-1:** Overall, I think the manuscript is well structured and the methodology is well prepared. The presented tables and figures support the findings that were presented in the text and provide good insights in the functionality of the presented algorithm. While I think the overall quality of the

manuscript in its present form is already good, I see some crucial points that require clarification. Other less significant points should also be improved to improve the quality and readability of the manuscript. I will outline my major concerns in the following and will address smaller issues in a line-by-line notation in the following section.

Response: Thanks for reviewer's comment. We have revised the manuscript based on the below comments raised by reviewer.

1. *Synthetic study design:*

**R3-2:** The observation data were generated with the same model structure that was later on investigated in the case study. Which means that "observed" variables (in this case velocity and temperature) were calculated with the exact same set of equations that in the following calculated the simulated variables. I think this property of the observation data could potentially impair the entire analysis and favour the simultaneous calibration of velocity and temperature.

As the authors outline in section 3.5 of the manuscript, multi-variable calibration problems are often affected by trade-offs between the variables for which performance metrics should be optimized (minimized). One reason for that is that models are simplification of the represented reality and parameters can affect multiple processes, sometimes in the opposite directions. Thus a change of a parameter value into one direction could improve the performance of one metric, while deteriorating the performance of another metric at the same time. This is different in the synthetic example where in fact the reality and the model are the same thing. Thus the "observed" time series of velocity and temperature perfectly agree with the model simplifications and assumptions.

Thus, in such a case providing both, temperature and velocity, to the search algorithm should better constrain the parameter response surface than only providing one of the two variables. If this hypothesis is sound, then the presented results would by affected by this effect. In this case only a real case scenario could provide an honest comparison of the three cases.

Response: The reviewer questioned the rationality of using data generated from the model as observations. We fully agree that using real observation data for such investigation is better. However, real observations for velocity are not available for this study, and are usually difficult to find. Due to scarce availability of observed velocity data, the use of both temperature and velocity for calibration of 3D lake hydrodynamic models is also rare both in practice and research. However, understanding of the importance of velocity data in calibrating such models is important and could be possible by generating synthetic observed data from lake hydrodynamic model simulations. These underlying hydrodynamic models are physically-based and precise. Hence, we believe that generating synthetic observation data from such models is reasonable. We agree with the reviewer that generating synthetic data from conceptual environmental simulation models may not be reasonable though (e.g., conceptual hydrologic models that have intrinsic structural errors and uncertain conceptual parameters). Of course, even in the case of 3D lake hydrodynamic models, synthetic model scenarios cannot be considered the same as real-world situations, but they are close representations of real-world situations. We think that our analysis on model-based synthetically generated observed data can provide at least some implications for the investigation people could do with real-world data. Our research implication of the use of velocity observation

is also in line with the study of Baracchini et al. (2020), where they also suggest having both temperature and current velocity for complete system calibration.

*Baracchini, T., Hummel, S., Verlaan, M., Cimatoribus, A., Wüest, A., & Bouffard, D. (2020). An automated calibration framework and open source tools for 3D lake hydrodynamic models. Environmental Modelling & Software, 134, 104787.*

The reviewer also highlighted the discussion of trade-offs in section 3.5. We need to clarify that section 3.5 discusses other possible applications for which our new method, DYNO, could be used. But we don't think the multi-variable hydrodynamic calibration problem we are dealing with is a trade-off problem. As we have discussed in the introduction (lines 113-122 of the original manuscript), it is not apparent that there is usually a trade-off between the fit of multiple variables / constituents. This might be true for conceptual models, e.g., hydrologic models, where equations are experimental, and there is lots of uncertainty embedded in parameterization. In such models, trade-offs typically exist between different sub-objectives (e.g., using different performance metrics). But hydrodynamic models are physically based. Moreover, lake water velocity is important for determining distribution of water temperature. Hence, we believe that simulating velocity more accurately in hydrodynamic models, is expected to improve accuracy of water temperature simulation and distribution. Consequently, a trade-off may not exist between the fits of temperature and velocity. We agree that investigation on real data might be more convincing in ensuring the above hypothesis as well, and this could be part a future study where observed data for both temperature and velocity is available. We have revised our statements in the manuscript to be more cautious about implications for real-world practice based on modeling results.

We added a new paragraph in the new section 3.6 Future Work: "*Our analysis of the role of temperature and velocity measurements in 3D hydrodynamic model calibration is based on synthetic observation data generated from models. We do think it is worthwhile to further investigate the role of temperature and velocity measurements in hydrodynamic model calibration if there are velocity measurements available in future.*"

We have also revised other similar statements in our manuscripts. For example in the conclusion, we have added the sentence: "*Our analysis suggests that for practical applications, both temperature and velocity data might need to be considered for model calibration. The common practice of calibrating only to temperature data might not be sufficient to reproduce the flow dynamics accurately and extra effort and expense to collect velocity data is expected to give a beneficial effect. However, our analysis are based on synthetical data from models, hence it is worthwhile to further investigate the role of temperature and velocity in model calibration with real temperature and velocity measurements.*" (in line 764-769 of the revised manuscript)

*2. Number of iterations of the search algorithm*

**R3-3:** This comment is somehow related to the previous one. In theory it should be possible for the search algorithm to identify the "true" parameter set that was used to generate the synthetic observation data. This global minimum is present on the parameter response surfaces of all three calibration scenarios and has a value of 0 independent of the used metric (DYNO or the single performance metrics). Thus, when not ending up in a local minimum all three cases should converge towards this

global minimum. As outlined above, I simply think that the Cal-Both scenario does this quicker due to the given reasons.

In the presented results all "best" solutions did not find the global minimum. As far as I got it right from the text, each experiment involved 8 iterations with 24 parallel evaluations in each iteration step. Given a computation time of 5 hours per simulation run (according to the text and thus 5 hours per parallel iteration) one experiment takes 1.6 days. I am wondering if experiments with larger numbers of iterations were performed (that are maybe just not shown). I would be interested how the convergence of the three calibration experiments develops with larger number of iterations.

Response: The reviewer brings up an interesting aspect to discuss. In theory, the global minimum has a value of 0 for all three calibration scenarios. However, it might not be correct that having zero temperature error at these observation locations would mean zero velocity error at these observation locations. This is because only the observation at part of the simulation space is used for calibration (not the observation data at each grid and each time step of the simulation space are used to calculate the temperature error). So the temperature error may be 0 at these observation locations, while the temperature error is not 0 at other locations where observation is not used in calibration. In this case, getting a temperature error 0 at observation locations cannot guarantee the velocity error is 0. Hence it is possible that the optimization ends up in a local minimum and does not converge towards the global minimum (both temperature and velocity error are zero) in cases calibrating to only temperature or velocity.

In optimization search, the algorithm might not even reach the solution with zero temperature or velocity error when calibrating to only one variable. This is because the optimization only focused on one variable (temperature or velocity) and ignored the calibration on another. For example, if only calibrated to temperature, the velocity error might still be large. In this case, the algorithm is less likely to find the solution with temperature error 0 while the velocity error is large. This is exactly what Figure R3-1 showed (We copied Figure 6 (a) in the revised manuscript below). In Figure 6 (a), when calibrating to temperature only, the best solution found after each iteration is converged at places where the velocity error is large, and temperature error is not reduced to 0. In contrast, the optimization search considered both temperature and velocity error when calibrating to temperature and velocity. Hence the best solution found after each iteration is improved in both temperature error and velocity error.

The reviewer asks for experiments with a large number of iterations. We did not increase the number of iterations because, 1) we found the optimization search almost converged at the last few iterations (As shown in the calibration progress plot Figure R3-2 below) and 2) our problem is computationally very expensive; one single simulation takes about 5 hours; 24 simultaneous simulations take about 8-12 hours because of limited memory resources. It takes more than 64 hours to conduct one calibration experiment, and we have three scenarios, and each scenario is repeated in three trials. This is a huge amount of computing. Meanwhile, in real practice, users are also unwilling to wait too long for the calibration process. That is why we compared the accuracy of the solutions within a limited computing budget.

In the section 3.2 of revised manuscript, we have added: "… *As discussed in experiment setup section, we conducted 8 iterations of the optimization search. This is a reasonable number of iterations for our case given that 1) the problem is computationally expensive (one experiment takes about 64 hours to run and there are 9 experiments) and 2) the calibration progress plot in iterations (Figure S1) indicates that the optimization search almost converged in 8 iterations*." and "… *Figure 6 also shows that when*

*calibrating to one variable, the optimization search is easily convergent (i.e., the best solution does not continue improving after few iterations even in terms of the fit of the variable considered in calibration). For example, in Fig. 6(a), when calibrating to temperature only, the best solution in in terms of temperature error does not improve much (in the last few iterations). The reason might be that when the velocity error is large, it is less likely that the temperature fit would be improved further. In contrast, when calibrating to both temperature and velocity, the optimization search continues improving in both the temperature and velocity fit. Hence only considering one variable in the calibration, it is difficult to get a solution that have small (or close to zero) error of the variable considered. We should also highlight that even though the optimization get a solution that has zero error in one variable, it does not mean that the error of another variable would be zero. The reason is that only the observation in part of the simulation space is used for calibration (not the observation data at each grid and each time step of the simulation space are used to calculate the temperature error). So the temperature error may be 0 at these observation locations, while the temperature error is not 0 at other locations where observation is not used in calibration. In this case, getting a temperature error 0 at observation locations cannot guarantee the velocity error is 0…"* (ine line 543-556 of the revised manuscript)

[Figure]

Figure R3-1 (Copied from Figure 6 in the manuscript). Calibration progress plot of the best solution found (in terms of objective function value) during optimization search by PODS when calibrating to temperature only (Cali-Tem), calibrating to velocity only (Cali-Vel), and calibrating to both temperature and velocity (Cali-Both). Three random trials (i.e., T1, T2, and T3) are plotted in (a), (b), and (c). Lower velocity and temperature error are better. The yellow makers are evaluation point in initial experiment design using LHD. Besides solutions in LHD, only the best solution in each of the optimization iterations is plotted (i.e., makers lined with lines). The line links the best previous solution in one iteration to the best solution in next iteration. The arrow indicates the direction from the previous solution to the next solution.

[Figure]

Figure R3-2. Calibration progress plot of PODS in Cali-Tem, Cali-Vel, and Cali-Both scenarios. The best solution found so far in average of three trials is plotted with the number of evaluations. Calibration progress of PODS in the Cali-Tem scenario is plotted in (a), where the temperature error ($f_{Tem}(X)$ in Eq. (7)) of the best solution found so far is plotted. Calibration progress of the Cali-Vel scenario is plotted in (b), where the velocity error ($f_{\overline{Vel}}(X)$ in Eq. (10)) is plotted. The calibration progress of PODS in the Cali-Both scenario is plotted in (a) and (b) in terms of temperature error and velocity error, respectively, of the best solution found so far based on Eq. (12).

**3. Consistency in the nomenclature**

**R3-4:** Overall I think the outline of DYNO and the explanation of variables is well done. Yet, I found some inconsistencies in the nomenclature and some mathematical definitions that must be improved. I will address these (at least the ones that I found) in the line-by-line comments.

Response: Thanks for the reviewer's comments. We have carefully revised the nomenclature and mathematical definitions. We also have a detailed response to the line-by-line comment below.

**4. English language**

**R3-5:** Overall I think the manuscript is well written. In some sections of the manuscript I found that the same errors repeat in every second line (e.g. singular/plural, missing articles). Although I am not a native speaker myself I had the impression that the manuscript could require some proof reading. Some sentences I had to read over and over again, but they still do not make sense to me. I addressed them as well in the line-by-line comments.

Response: Thanks for the reviewer's comments. We have carefully looked at the reviewer's comments about these language issues. We have revised them as the reviewer suggested in the revised manuscript.

5. *Non-color blind color theme in figures*

**R3-6:** Although this does not affect me, I would advise to change the color theme in the figures and refrain to use red and green at the same time (8% of males are affected by red-green color blindness).

Response: Thanks for the reviewer's comments. We agree with the reviewer's suggestion, which is very important. We have changed the "red and green" to "red and blue" colors for all related figures.

**Line-by-line comments**

**p.1 L13, p.4 L123 and further**: Please use a consistent naming for Dynamically Normalized Objective function. Either all first letters caps or none. It differs throughout the text. Also only use the acronym after it was first introduced in the text.

Response: We have revised the manuscript with a consistent naming for Dynamically Normalized Objective Function and used acronym after the first introduction in the text.

**p.1 L21 - 22:** Please rephrase this sentence. Further, I think the statement is not universally true how it is formulated, that calibrating only one variable does not improve the calibration of another variable.

Response: Our statement is not to say that calibrating one variable does not "improve" the calibration of another. Hence, we are saying calibrating one variable cannot "guarantee" an improved fit of another. Our results support our statement. When calibrating to temperature only, the velocity error is large, and vice versa. It is true in our case. Of course, it might not be universally true for other cases. We will rephrase the sentence to be: "*The result indicates that DYNO can balance the calibration in terms of water temperature and velocity and that calibrating to only one variable (e.g., temperature or velocity) cannot guarantee the goodness-of-fit of another variable (e.g., velocity or temperature) in our case*." (in line 19-21 of the revised manuscript)

**Eq. 2, Table 1:** Consistent notation of *Sim* and *Obs* necessary. Either capital first letter or lower case.

Response : We will capitalize the notation of Sim and Obs in Table 1 and throughout the manuscript.

**Table 1**: Please rephrase the sentence: "The set of variables the observation data of which is used in calibration"

Response: We have rephrased the sentence to be: "The set of variables whose observation data is used in calibration." In Table 1

**Table 1:** I would suggest to add the vector $\{x_1, ..., x_d\}$ to the definition of $X$ as e.g. Table 2 refers to them.

Response: We have added following text into the definition of X in Table 1: "$X = (x_1, x_2, \ldots, x_d)$."

**Table1 and throughout the text:** I would suggest to use the nomenclature *Sim$^k_{j,t}$* (same for *Obs*) in the definition and throughout the text to indicate the time dependency. This nomenclature is in fact introduced later in the manuscript. Also the definition refers to the time *t* which is not indicated in the variable.

Response: We will add the definition of $Sim^k_{t,j}(X)$ and $Obs^k_{t,j}$ in Table 1. We will bold the definition of $\boldsymbol{Sim}^k_j(X)$ and $\boldsymbol{Obs}^k_j$ to denote the time series of simulation output and observations data at all time steps ($\boldsymbol{Sim}^k_j(X) = \left(Sim^k_{1,j}(X), \dots, Sim^k_{N,j}(X)\right)$ and $\boldsymbol{Obs}^k_j = (Obs^k_{1,j}, \dots, Obs^k_{N,j})$). We keep the definition of $\boldsymbol{Sim}^k_j(X)$ and $\boldsymbol{Obs}^k_j$ for the simplicity of Eq. (2) (since the goodness of fit function takes the simulation output and observation time series as input).

**p.5 L153**: ...multiple variables *with* a single objective function.

Response: We have revised the sentence to be: "… *formulate the error of multiple variables with a single objective function*". (in line 165 of the revised manuscript)

**p.5 L159:** Please remove \textit{value} as the distribution of NRMSE is not a single value.

Response: We are trying to make two comparisons: one is the highest attainable NRMSE value another is the distribution of the NRMSE value. We have revised the sentence to be: "*However, it is still possible that the highest attainable value (or distribution) of NRMSE (or KGE) across the parameter space for one variable maybe be much higher than the highest attainable value (or distribution) of NRMSE (or KGE) of another variable.*" (in line 170-172 of the revised manuscript)

**p.5 L160:** Same as above.

Response: We have resolved this comment in response to the comment above.

**p.5 L160:** Hence **,** ...

Response: We have revised the sentence to be: "Hence, … ." (in line 172 of the revised manuscript)

**p.5 L163:** Again consistent naming of DYNO or removing the full name as the acronym was already introduced.

Response: We have removed the full name of DYNO. (in line 173 of the revised manuscript)

**p.5 L166:** ...all evaluations in $\boldsymbol{\psi}$ found so far?

Response: $\boldsymbol{\psi}$ is the set of evaluations found so far by the optimization. We have revised the sentence to be: "…. *Let $\boldsymbol{\psi}$ be the set of evaluations found so far by the optimization, DYNO (as shown in Eq. (3)) normalizes the error of each variable $f_k(X)$ with its upper and lower bound, $f^{max}_k$ and $f^{min}_k$ of all evaluations in $\boldsymbol{\psi}$*" (in line 174-175 of the revised manuscript)

**p.5 L166:** The variable $\boldsymbol{\psi}$ was not introduced at this point and is I think not defined in the manuscript.

Response: We have added the definition of $\boldsymbol{\psi}$ in response to the comment above.

**p.5 L167:** *The* mathematical formulation...

Response: We have revised the sentence to be: "*The mathematical formulation ...*" (in line 178 of the revised manuscript)

**p.6 L172**: ...all evaluatio*ns* ...

Response: We have revised the sentence to be: "… all evaluations …" (in line 182 of the revised manuscript)

**p.6 L172 - 174:** Please rephrase this sentence. It is in my opinion not clear what is meant.

Response: We have split the sentence to two short sentence: "*where $f_k^{max}(\boldsymbol{X})$ and $f_k^{min}(\boldsymbol{X})$ are the maximum and minimum values of $f_k(\boldsymbol{X})$ for all evaluations in . $f_k^{max}(\boldsymbol{X})$ and $f_k^{min}(\boldsymbol{X})$ have to be updated dynamically in each iteration during optimization.*" (in line 182-183 of the revised manuscript)

**p.6 L180 - 185:** The procedure as described is confusing and does not really make sense to me. A calibrated model exists from a previous study. This calibrated model was taken, but different parameters based on "expert guessing" were then used to run the model. So it is not the calibrated model anymore?

Response: We have revised the words "expert guessing". The set of "true" value we use is the manual calibration results from expert. The words "expert guessing" might cause confusion. We have replaced it with "manual calibration by experts". (in line 195 of the revised manuscript)

**p.6 L186:** Replace *a* by *the* as you did not use a parameter set but the parameter set $\mathbf{X^R}$.

Response: We have revised the sentence to be: "the vector of model parameters $X^R$" (in line 198 of the revised manuscript)

**p.6 L193:** ...the true value*s* of the parameter vector $\mathbf{X^R}$ *are*...

Response: We have revised the sentence to be: "...*the true values of the parameter vector $X^R$ are...*". (in line 205 of the revised manuscript)

**p.6 L194**: I would replace *value* with *set*.

Response: We have replaced value with set. (in line 206 of the revised manuscript)

**p.6 L202 - 203**: This sentence is not clear to me.

Response: We have revised the sentence to be: "*We have the five sampling locations across the reservoir. The observations data at these five locations are used to calibrate the model parameters.*" (in line 214-215 of the revised manuscript)

**p.7 L226**: coefficien*ts*

Response: We have changed coefficients to coefficient: "*The vertical exchange of horizontal momentum and mass is affected by vertical eddy viscosity and eddy diffusivity coefficients.*" (in line 235-236 of the revised manuscript)

**p.8 L229**: *the* Manning formulation

Response: We have added "the". (in line 247 of the revised manuscript)

**p.8 L229:** ...which is a parameter that should also be calibrated.

Response: In the revised manuscript we have removed this sentence.

**p.8 L232:** ...*are* parameterized...

Response: We have revised the sentence to be: "...*are* parameterized..." (in line 242 of the revised manuscript)

**p.8 L237:** *M*anning coefficient

Response: We have capitalized the word "Manning". (in line 247 of the revised manuscript)

**Section 2.5:** This section is actually a repetition of L146 – 150

Response: We have removed the repeated information. Section 2.5 gives the detailed calibration formulations. We have revised the first paragraph of section 2.5 to be: "*Three scenarios are considered to investigate the impact of model calibration against temperature and/or velocity observations (as discussed in section 2.1). The first two scenarios calibrate to only one variable, and the last scenario calibrates both variables simultaneously. This section give the detailed calibration formulations of these three scenarios.*"

**p.9 L280:** Remove *them*

Response: We have removed "them". (in line 296 of the revised manuscript)

**p.9 L287:** I would suggest to replace *a roughly* by *an approximately*

Response: We have replaced "a roughly" by "an approximately". (in line 303 of the revised manuscript)

**Caption Table 3:** formulation*s*

Response: We have changed "formulation" to "formulations". (In Table 3)

**p.10 L309:** Use the acronym DYNO instead

Response: We have replaced "the Dynamically Normalized Objective Function" with "DYNO". (in line 308 of the revised manuscript)

**p.10 L317:** Replace *is* with *are*

Response: In the revised manuscript, we have revised this sentence to be: "The objective function value $F(X|K)$ for all evaluations found in current and previous iterations need to be recalculated because of the update of $f_k^{max}(X)$ and $f_k^{min}(X)$." (in line 337-338 of the revised manuscript)

**Figure 2**: The positions of the labels *Yes* and *No* in the stopping criteria are not clear to me.

Response: We have corrected the position of "Yes" and "No" in Figure 2.

**p.12 L359 - 366:** The argumentation in this paragraph could be affected by the affect of synthetic observation data as outlined in the first major remark.

Response: We have responded the first major remark in response to R3-1. The discussion in line 359-366 of the original manuscript is stating the results based on physical models. These hydrodynamic models are built based on physics and knowledge human learned in the past hundreds of years. Hydrodynamic models are not like most of hydrology models that with lots of uncertainty. They of course cannot be try the same as real world situation but they are close representations of the real world situation. We think the investigation on models can provide at least some implications for the investigation people could do with real world data. Our research implication of the use of velocity observation is also in line with the study of Baracchini et al (2020), where they also suggest have both temperature and current velocity for a complete system calibration.

*Baracchini, T., Hummel, S., Verlaan, M., Cimatoribus, A., Wüest, A., & Bouffard, D. (2020). An automated calibration framework and open source tools for 3D lake hydrodynamic models. Environmental Modelling & Software, 134, 104787.*

We have added the discussion above in the end of Section 3.1.1 : "… *Our analyses here are based on physical models, which are built based on physics laws and knowledge human have learned over hundreds of years. Our findings here are in line with the study of Baracchini et al (2020), where they also suggested have both temperature and velocity for a complete system calibration*." (in line 427-430 of the revised manuscript)

**p.13 L395:** Missing end '.' of sentence?

Response: We have added the missing end "." (in line 466 of the revised manuscript)

**p.13 L395 - 399:** This sentence is confusing and might require rephrasing.

Response: We will rephrase the sentence to be: "*Figure 3 illustrates that calibrating to temperature data only (red scatter plot) results in larger velocity error $\Delta\overrightarrow{Vel}$, relative to velocity error when calibrating to velocity data only (Cali-Vel scenario, i.e., black scatter plot) or to both velocity and temperature data (Cali-Both scenario, i.e., blue scatter plot).*" (in line 466-468 of the revised manuscript)

**p.14 L406 - 416**: I had the impression that many articles were missing here. This paragraph is hard to read in general and could potentially be revised.

Response: We have revise this paragraph to be: "*Figure 4 shows the temperature error of solutions from three different calibration scenarios: Cali-Tem (red time-series), Cali-Vel (black time-series) and Cali-Both scenarios (blue time-series). The errors between simulated and observed water temperature at the surface, middle and bottom layers of two stations (STN. A1 and STN B1) are plotted.In general, the temperature error of the solution in Cali-Both scenario is generally close to zero °C for all the layers and stations shown. The solution in Cali-Tem scenario also got temperature error close to zero °C at the middle and bottom layer at STN. A1, but it has larger temperature error than solution in the Cali-Both at the surface layer of STN. A1 and all layers of STN. B1. The solution in the Cali-Vel scenario generally overestimated the water temperature in all locations (i.e., all the surface, middle and bottom layers at both stations). The temperature error of the solution in the Cali-Vel scenario is much larger than that of the solution in Cali-Tem and Cali-Both scenarios in the middle and bottom layer of both stations. The temperature error at most times, for the Cali-Vel scenario, is greater than 0.1 °C.*" (in line 480-490 of the revised manuscript)

p.15 L431: *the* calibration...

Response: We have added "the". (in line 521-522 of the revised manuscript)

**Figure caption Fig. 5 L451:** in term*s* of...

Response: We have changed "term" to "terms". (in Figure 6 caption of the revised manuscript)

**Figure 6:** Values in the darkest hex tiles are almost not readable.

Response: We have changed the color of values in the darkest hex tiles to make it readable. (in Figure 7 of the revised manuscript)

**p.19 L536 - 539:** This sentence was not clear to me and might require revision.

Response: We will revised the sentence to be: "*Outlier or extremely bad solutions are also likely happen for calibration problems where the model output is very sensitive to the calibration parameters (i.e., a small change in model parameters can cause huge changes in the model output that leads to much worse solutions).*" (in line 643-644 of the revised manuscript)

**p.20 L553 - 566:** Again, the argumentation in this paragraph could be affected by the affect of synthetic observation data as outlined in the first major remark.

Response: We have responded the first major remark in response to R3-1. These hydrodynamic models are built based on physics and knowledge human learned in the past hundreds of years. Hydrodynamic models are not like most of hydrology models that with lots of uncertainty. They of course cannot be try the same as real world situation but they are close representations of the real world situation. We think the investigation on models can provide at least some implications for the investigation people could do with real world data. We have added the following text after this paragraph to limit our language by not making universal statement: "… *Our analysis is based on synthetic observation data from the physical model since we don't have real velocity measurements. These physical models are*

*based on physics laws and knowledge from human's observation and understanding of environment. The analysis from modelling can provide some implications for the real world situation. Hence, it is worthwhile to repeat the analysis based on real data if there are real velocity measurements available in future.*" (in line 671-675 of the revised manuscript)

**p.20 L573**: Remove *these*

Response: We have removed "these". (in line 682 of the revised manuscript)

**p.20 L574:** suggest *to* have

Response: We have changed "suggest" to "suggest to". (in line 683 of the revised manuscript)

**p.21 L613:** We conclude that the Dynamically Normalized Objective Function *that* we propose

Response: We have revised the sentence to be: "*We conclude that the DYNO objective function that we propose …*" (in line 749 of the revised manuscript)

---

## Referee Report (RR1)

**Referee report for the manuscript HESS-2021-601: 'A Novel Objective Function DYNO for Automatic Multi-variable Calibration of 3D Lake Models'**

Christoph Schürz, christoph.schuerz@ufz.de

June 7 2022

**General evaluation and major comments**

I want to thank the authors for putting great effort in the revision of the manuscript. Although I think that the authors addressed most of the technical remarks well, the major issue that is in my opinion present in this study requires further discussion. I am afraid that the authors reply does not fully dispel my doubts in the study design. I will again address this issue in the next section. In the following section I address minor line-by-line comments.

**Continuing the discussion on the synthetic study design**

I appreciate that the authors stress the fact of using only synthetic data in their case study in the revised version of the manuscript. Yet, I would like pick up the discussion on the synthetic case study and come back to the arguments that were in my opinion not fully addressed.

I raised two major concerns in my previous review: i) synthetic observation data that were generated with the same set of equations that are then fitted to these date would favour a combined calibration (velocity and temperature), as both data would constrain the function space stronger than each of the variables individually; ii) real world settings are usually trade-off problems in the calibration of multiple variables.

While I can accept the reply to argument ii), I think the reply to argument i) misses the main issue. The authors argue that the model is physically based and should therefore be capable of reproducing observations of physical variables. Further, as the model is physically based it is capable of generating realistic data that can be used as observation data in a test case. The authors also refer to Baracchini et al. (2020) as this study supports the use of velocity data and temperature data for a 3D lake hydrodynamic model calibration.

The theoretical argument may be reasonable, but it misses the fact the measurement uncertainty in real observation data can be substantial. The study of Baracchini et al. (2020) used measured observation data for velocity and temperature. Baracchini et al. (2020) mention that measured flow velocities are often close to their measurement accuracy in one of the case studies while the difference between the variance in temperature measurements and the measurement accuracy is substantially lower. A synthetic example that uses simulated data as observation data cannot account for this property of the observation data. To make the synthetic study more comparable to a real case setting overlaying the synthetic data with an error model may be a probable solution. Although I fully understand that this is likely infeasible to be still considered in this study this aspect should be addressed. I think the distributions of the observation data and their uncertainties can strongly influence the performance of the simultaneous calibration.

**Line-by-line comments**

**p.4 L141** - **L143** Please rephrase the sentence. The formulation (e.g. don't) is highly informal.

**p.8 L256** Remove *assume*.

**p.9 L266** This section give**s**...

**p.13 L382** Remove *about*. It is exactly 8 times 24.

**p.14 L427** Please revise or remove this sentence. It sounds unnecessary and very vague to me.

**p.14 L443** Please rephrase 'is set be the middle'.

**p.15 L447** ...in *the* other two scenarios...

**p.15 L450** - **L452** Please revise the two sentences. The wording sounds odd. Avoid using vague wording such as 'good' and 'bad'.

**p.23 L671** - **L675** Please revise this section. The formulation (e.g. don't) is highly informal. Further, either it is a physical law or some empirical observation by a human. I would avoid this formulation.

**References**

Baracchini, T., S. Hummel, M. Verlaan, A. Cimatoribus, A. Wüest, and D. Bouffard (2020). "An automated calibration framework and open source tools for 3D lake hydrodynamic models". In: *Environmental Modelling & Software* 134, p. 104787. DOI: https://doi.org/10.1016/j.envsoft.2020.104787.

---

## Author Response (AR2)

Manuscript ID: hess-2021-601

**Response to Reviews**

We appreciate and would like to thank two reviewers for taking the time and effort to review our revised manuscript. We have improved our manuscript based on reviewer's helpful comments. Our point-to-point response are below (comments of the referee are in black, our responses are in purple, texts quoted from the revised manuscript are in italic).

**Point-to-point response**

**Response to referee comment Referee #1**

**R1-1:** The new version of the manuscript by Xia et al. has addressed all comments I raised. The authors included several new paragraphs based on my questions for clarification as well as those from the other reviewers. I see this made their paper way more accessible to a wider audience. I have no concerns in recommending the publication of the paper in its present form, except for a few language corrections which should be accomplished. I list some of them below but I recommend the authors to double check the manuscript and pay some more attention to the new paragraphs as they include several typos.

Response: Thanks for the reviewer's comments. We have revised the English language based on reviewer's detailed comments below. We also did a careful proofreading for the revised manuscript.

**R1-2:** L343-344: "Because the change of objective function value for all evaluations in $\psi$," a verb is missing here

Response: We have revised the sentence to be: "*Because the objective function value for all evaluations in $\psi$ changed after each iteration, …*" (In L343-344 of the revised manuscript)

**R1-3:** L351: missing "is" in "since it more likely"?

Response: We have added the missing "is". (In L351 of the revised manuscript)

**R1-4:** L353: missing "were" in "that not explored"?

Response: We have added the missing "were" (In L353 of the revised manuscript)

**R1-5:** L362: missing space in "thefitting"

Response: We have added the missing space between "the fitting" (In L362 of the revised manuscript)

**R1-6:** L447-452: please revise the English of the entire comment to Figure 5. What do the authors mean with "is worst"? Maybe they meant "worse"? Still it's not clear what they mean with worse/worst. Please comment the figure in a more clear way. In line 450 "Cali-Tem in worst" maybe that "in" was a "is"?

Response: We have added the explanation of what "the worst" mean. We "the worst" is defined in terms of how far the parameter value of the optimization solution is from the true solution. We have revised the entire comment to Figure 5 to be: "*In general, the solution in the Cali-Both scenario is closer to the True solution than solutions in the other two scenarios in terms of parameter values. Calibrated values proposed by the Cali-Both scenario are closest (relative to other scenarios) to the True values for four parameters (i.e., $D_H^{back}$, $D_V^{back}$, $H_{Secchi}$, $c_H$). Moreover, besides the Manning coefficients, the calibrated values proposed by the Cali-Both scenario are not the worst (relative to the other two scenarios) for any other parameter. In contrast, calibrated values proposed by the Cali-Tem scenario are worst (i.e., the parameter values are farthest from the true solution, relative to other scenarios) for five parameters (i.e., $v_H^{back}$, $D_H^{back}$ $v_V^{back}$, $D_V^{back}$, $c_e$ ) and calibrated parameter values for the Cali-Vel scenarios are worst for $L_{oz}$, $H_{Secchi}$, $c_H$. This indicates that calibrating to both temperature and velocity can help to prevent the value of the 9 calibration parameters from being very far from the corresponding value for the True solution.*" (In L445-L454 of the revised manuscript)

**R1-7:** Figure 5: Please add the legend for the lines

Response: Thanks for reviewer's comments. We have added the legend for the lines in Figure 5.

**R1-8:** L551: getS

Response: We have added the missing "s" after "get" (In L553 of the revised manuscript)

**R1-9:** Figure 7: missing "is" in the description of the final solution hexagon

Response: We have added the missing "is" in the caption of Figure 7. (In L581 of the revised manuscript)

**R1-10:** L727: it worthS

Response: We have changed it to be "it is worth mentioning that …"(In L729 of the revised manuscript)

**Response to referee comment Referee #2**

**R2-1:** I want to thank the authors for putting great effort in the revision of the manuscript. Although I think that the authors addressed most of the technical remarks well, the major issue that is in my opinion present in this study requires further discussion. I am afraid that the authors reply does not fully dispel my doubts in the study design. I will again address this issue in the next section. In the following section I address minor line-by-line comments.

Response: Thanks for reviewer's time and careful review. We have revised the manuscript based on reviewer's comments on the study design. We have responded in the next comment. We have also revised manuscript based on reviewer's minor line-by-line comments below.

**R2-2:** I appreciate that the authors stress the fact of using only synthetic data in their case study in the revised version of the manuscript. Yet, I would like pick up the discussion on the synthetic case study and come back to the arguments that were in my opinion not fully addressed.

I raised two major concerns in my previous review: i) synthetic observation data that were generated with the same set of equations that are then fitted to these date would favour a combined calibration (velocity and temperature), as both data would constrain the function space stronger than each of the variables individually; ii) real world settings are usually trade-off problems in the calibration of multiple variables.

While I can accept the reply to argument ii), I think the reply to argument i) misses the main issue. The authors argue that the model is physically based and should therefore be capable of reproducing observations of physical variables. Further, as the model is physically based it is capable of generating realistic data that can be used as observation data in a test case. The authors also refer to Baracchini et al. (2020) as this study supports the use of velocity data and temperature data for a 3D lake hydrodynamic model calibration.

The theoretical argument may be reasonable, but it misses the fact the measurement uncertainty in real observation data can be substantial. The study of Baracchini et al. (2020) used measured observation data for velocity and temperature. Baracchini et al. (2020) mention that measured flow velocities are often close to their measurement accuracy in one of the case studies while the difference between the variance in temperature measurements and the measurement accuracy is substantially lower. A synthetic example that uses simulated data as observation data cannot account for this property of the observation data. To make the synthetic study more comparable to a real case setting overlaying the synthetic data with an error model may be a probable solution. Although I fully understand that this is likely infeasible to be still considered in this study this aspect should be addressed. I think the distributions of the observation data and their uncertainties can strongly influence the performance of the simultaneous calibration.

Response: We sincerely thank the reviewer for elaborating their argument pertaining to the design and use of synthetic data in this study. We do agree with reviewer that our synthetic example did not account for uncertainty of observation data, and we agree that this point warrants a discussion in the manuscript.

The reviewer also mentioned the discussion of Baracchini et al. (2020) in relation to accounting for the measurement uncertainty of temperature and velocity. We believe that their discussion can be elaborated further, and in-fact, supports the use of DYNO. In Baracchini et al. (2020)'s study, the cost function is the square of temperature (or velocity) difference between computed and measured value divided by the observational uncertainty. The observational uncertainty in Baracchini et al. (2020)'s study is defined as the maximum value of two elements: 1) the instrument precision, and 2) the temporal dynamic variability (i.e., standard deviation of observations) at the measurement location over a short-term period (e.g., ±6h to ±12h). The first element is a fixed value in their case (0.002°C for temperature and 0.8cm s$^{-1}$ for velocity). . Baracchini et al. (2020) mentioned that the computed and measured velocity value is close to velocity measurement accuracy of 0.8cm s$^{-1}$ (Teledyne RDI Workhorse Sentinel) while

temperature computed and measured value is orders of magnitude larger relative to temperature measurement accuracy in their study. Hence compared with temperature observations, velocity has much less impact on the cost function than temperature and therefore it might have an issue to calibrating both temperature and velocity simultaneously. In our opinion, Barachhini et al. (2020)'s discussion and findings add more value to our study. The issue of velocity having much smaller impact on the cost function than temperature is true for traditional approach of using a fixed weight in adding the cost function of temperature and velocity together as the objective function. However, the proposed new objective function in our study handles the error of each variable separately (the error functions may also account for measurement error) and dynamically normalized their error so to balance their impact on the objective function (and hence their impact on calibration). The issues mentioned by Barachhini et al. (2020) related to the difference in terms of the magnitude of temperature and velocity's cost function value is exactly what our new proposed objective function (DYNO) attempts to rectify. We thank the reviewer for bringing up this issue since it reminded us to point out in the manuscript the advantage of the new objective function DYNO to incorporate the error in each variable.

However, as mentioned earlier, we do acknowledge that measurement errors are important and should be incorporated in future studies to understand effectiveness of DYNO in multi-constituent calibration settings.

In Section 3.6 of our revised manuscript we have added following discussion: "*Moreover, the synthetic observation data used in our analysis did not account for the measurement uncertainty of observation data. Further investigations related to the impact of measurement errors on the calibration setup proposed here will also be beneficial. It is important to note that the measurement uncertainty and distribution of different variables could be different (and thus, our new objective function formulation DYNO could be very useful in balancing the calibration process in such a scenario). For example, Baracchini et al. (2020) reported that the measured and computed velocity value (in the magnitude of 1 cm s$^{-1}$ for velocity in hypolimnion layer) is close to velocity measurement uncertainty 0.8cm s$^{-1}$ (the velocity measurement instrument precision) while the computed and measured temperature value is an order of magnitudes larger relative to temperature measurement uncertainty in their study. The difference in terms of measurement accuracy and measurement value could lead to a different magnitude of error function value for each variable (temperature or velocity). (In their study, the error function is the square of temperature (or velocity) difference between computed and measured value divided by the observational uncertainty). Baracchini et al. (2022) pointed out that such discrepancy hinders the use of different kinds of data (e.g., temperature and velocity) simultaneously because the impact of velocity on the cost function is almost negligible compared with temperature observations. Hence, they carried out a separate discussion for both types of observation data. Their argument is true if the calibration objective function is a sum of temperature or velocity's error function with a fixed weight. In this case, the difference of the error function value's magnitude might lead to a biased calibration to the variable that has a larger impact on the error function.*

*However, our proposed new objective function DYNO dynamically normalizes the error function value of each variable using the maximum and minimum value of each variable's error*

*function value obtained during the calibration and hence balances the impact of each variable on the objection function. Hence DYNO is designed to work well in scenarios where the error function values of each variable are significantly different due to differences in measurement uncertainty and the distribution of each variable's observations."*

Baracchini, T., Hummel, S., Verlaan, M., Cimatoribus, A., Wüest, A., and Bouffard, D.: An automated calibration framework and open source tools for 3D lake hydrodynamic models, Environmental Modelling & Software, 134, 104787, 2020.

Line-by-line comments

p.4 L141 - L143 Please rephrase the sentence. The formulation (e.g. don't) is highly informal.

Response: We changed the sentence to *"Real observations for velocity are not available in our case as well."* (In L141 of the revised manuscript)

p.8 L256 Remove assume.

Response: We have removed "assume" (In L256 of the revised manuscript)

p.9 L266 This section gives...

Response: We have added the missing "s" after "give" (In L266 of the revised manuscript)

p.13 L382 Remove about. It is exactly 8 times 24.

Response: We have removed "about" (In L382 of the revised manuscript)

p.14 L427 Please revise or remove this sentence. It sounds unnecessary and very vague to me.

Response: We have removed the sentence in Line 426-427 of the manuscript in last submission.

p.14 L443 Please rephrase 'is set be the middle'.

Response: We have rephrased the sentence to be: "The parameter value of the uncalibrated solution (in Table S1) uses the mean of the calibration range in Table 2" (In L442 of the revised manuscript)

p.15 L447 ...in the other two scenarios...

Response: We have added the missing "the" in "… in the other two scenarios …". (In L446 of the revised manuscript)

p.15 L450 - L452 Please revise the two sentences. The wording sounds odd. Avoid using vague wording such as 'good' and 'bad'.

Response: Thanks for reviewers' suggestion. We have revised this two sentences to be: *"In contrast, calibrated values proposed by the Cali-Tem scenario are the worst (i.e., the parameter values are farthest from the true solution, relative to other scenarios) for five parameters (i.e., $v_H^{back}$, $D_H^{back}$ $v_V^{back}$, $D_V^{back}$, $c_e$) and calibrated parameter values for the Cali-Vel scenarios are worst for $L_{oz}$, $H_{Secchi}$, $c_H$. This indicates that calibrating to both temperature and velocity can*

*help to prevent the value of the 9 calibration parameters from being very far from the corresponding value for the True solution.*" (In L450-L454 of the revised manuscript)

p.23 L671 - L675 Please revise this section. The formulation (e.g. don't) is highly informal. Further, either it is a physical law or some empirical observation by a human. I would avoid this formulation.

Response: We have revised this section to be: "*Our analysis is based on synthetic observation data from the physical model since we do not have real velocity measurements. These physical models are based on physics laws. The analysis from modelling can provide some implications for the real-world situation. Hence, it is worthwhile to repeat the analysis based on real data if there are real velocity measurements available in the future.*"